# *PPM1D* mutations are oncogenic drivers of de novo diffuse midline glioma formation

Prasidda Khadka [1,2,3,19], Zachary J. Reitman [4,5,6,19], Sophie Lu[7], Graham Buchan[7], Gabrielle Gionet[7], Frank Dubois[1,2], Diana M. Carvalho [8], Juliann Shih [2], Shu Zhang[2], Noah F. Greenwald[1], Travis Zack[1], Ofer Shapira[1], Kristine Pelton[9], Rachel Hartley[10], Heather Bear[15], Yohanna Georgis[7], Spandana Jarmale[7], Randy Melanson[2], Kevin Bonanno[2], Kathleen Schoolcraft[9], Peter G. Miller[2,11], Alexandra L. Condurat[7], Elizabeth M. Gonzalez[2,7], Kenin Qian[7], Eric Morin [7], Jaldeep Langhnoja[10], Leslie E. Lupien[7], Veronica Rendo[1], Jeromy Digiacomo[7], Dayle Wang [7], Kevin Zhou [7], Rushil Kumbhani[7], Maria E. Guerra Garcia[5], Claire E. Sinai[9], Sarah Becker[9], Rachel Schneider[9], Jayne Vogelzang[9], Karsten Krug[2], Amy Goodale[2], Tanaz Abid[2], Zohra Kalani[2], Federica Piccioni [2], Rameen Beroukhim [1,2], Nicole S. Persky[2], David E. Root [2], Angel M. Carcaboso [12], Benjamin L. Ebert [2,11,13], Christine Fuller[14], Ozgun Babur[16], Mark W. Kieran[7,18], Chris Jones[8], Hasmik Keshishian[2], Keith L. Ligon[9], Steven A. Carr[2], Timothy N. Phoenix [10,15✉] & Pratiti Bandopadhayay[2,7,17✉]

The role of *PPM1D* mutations in de novo gliomagenesis has not been systematically explored. Here we analyze whole genome sequences of 170 pediatric high-grade gliomas and find that truncating mutations in *PPM1D* that increase the stability of its phosphatase are clonal driver events in 11% of Diffuse Midline Gliomas (DMGs) and are enriched in primary pontine tumors. Through the development of DMG mouse models, we show that *PPM1D* mutations potentiate gliomagenesis and that PPM1D phosphatase activity is required for in vivo oncogenesis. Finally, we apply integrative phosphoproteomic and functional genomics assays and find that oncogenic effects of *PPM1D* truncation converge on regulators of cell cycle, DNA damage response, and p53 pathways, revealing therapeutic vulnerabilities including MDM2 inhibition.

[1] Department of Cancer Biology, Dana Farber Cancer Institute, Boston, MA 02215, USA. [2] Broad Institute of MIT and Harvard, Cambridge, MA 02142, USA. [3] Harvard Biological and Biomedical Sciences PhD Program, Harvard University, Cambridge, MA 02138, USA. [4] Department of Radiation Oncology, Duke University, Durham, NC 27710, USA. [5] Duke Cancer Institute, Duke University, Durham, NC 27710, USA. [6] The Preston Robert Tisch Brain Tumor Center at Duke, Duke University, Durham, NC 27710, USA. [7] Dana-Farber/Boston Children's Cancer and Blood Disorders Center, Boston, MA 02215, USA. [8] Division of Molecular Pathology, Institute of Cancer Research, London, UK. [9] Department of Oncologic Pathology, Dana Farber Cancer Institute, Boston, MA 02215, USA. [10] Division of Pharmaceutical Sciences, James L. Winkle College of Pharmacy, University of Cincinnati, Cincinnati, OH 45267, USA. [11] Department of Medical Oncology, Dana-Farber Cancer Institute, Boston, MA 02215, USA. [12] Department of Pediatric Hematology and Oncology, Hospital Sant Joan de Deu, Institut de Recerca Sant Joan de Deu, Barcelona 08950, Spain. [13] Howard Hughes Medical Institute, Chevy Chase, MD 20815, USA. [14] Department of Pathology, Cincinnati Children's Hospital Medical Center, Cincinnati, OH 45267, USA. [15] Research in Patient Services, Cincinnati Children's Hospital Medical Center, Cincinnati, OH 45267, USA. [16] College of Science and Mathematics, University of Massachusetts Boston, Boston, MA 02125, USA. [17] Department of Pediatrics, Harvard Medical School, Boston, MA 02215, USA. [18] Present address: Bristol Myers Squibb, Boston, Devens, MA 01434, USA. [19] These authors contributed equally: Prasidda Khadka, Zachary J. Reitman. ✉email: phoenity@ucmail.uc.edu; Pratiti_bandopadhayay@dfci.harvard.edu

D iffuse midline gliomas H3K27-altered (DMG) are universally fatal pediatric brain tumors. In addition to the characteristic histone K27M mutations[1,2], DMGs often harbor alterations in the p53 pathway, including mutations in *TP53* itself, as well as *PPM1D*, a regulator of p53 activity[3–6]. *PPM1D*, also known as *WIP1*, is a PP2C family phosphatase that regulates known members of the DNA damage response (DDR) pathways, most notably p53, as well as other targets such as γ-H2AX, CHK1, ATM, and ATR[7–9]. *PPM1D* truncating mutations that increase the stability of the phosphatase have also recently been found in clonal hematopoiesis of indeterminate potential (CHIP), where it drives selective outgrowth of the mutant clones in response to cytotoxic chemotherapy[10,11].

Previous work in gliomas has largely focused on the role of *PPM1D* mutation as a driver of radiation resistance and/or evaluated therapeutic vulnerabilities associated with the mutation[12–14]. However, to fully characterize *PPM1D* as a therapeutic target, a number of questions remain, including its role in DMG oncogenesis, its necessity for the proliferation and maintenance of DMG cells, and finally, its mechanisms of action as an oncogene.

In this work, we apply an integrative functional genomic and proteomic approach to systematically examine the role of mutant *PPM1D* in enhancing DMG oncogenesis and explore its potential as a therapeutic dependency. To this end, we use murine models to demonstrate that endogenous truncation of murine *Ppm1d* together with histone and *Pdgfra* mutations is sufficient to drive de novo brainstem glioma formation, as is exogenous expression of truncated human *PPM1D* (*PPM1D*tr). Moreover, we find *PPM1D* to be necessary for the proliferation of *PPM1D*-mutant DMGs, and the mechanisms through which *PPM1D* confers oncogenicity to be primarily mediated through p53, cell cycle, and DDR pathways, creating cellular vulnerabilities that can be therapeutically targeted.

## Results

### Truncating mutations and amplifications of *PPM1D* are common in histone-mutant midline gliomas and other cancer types

*PPM1D* mutations have previously been reported in gliomas and have been shown to confer resistance to radiation[14]. However, its role as a de novo oncogene to enhance glioma formation has not been explored. To further evaluate the role of *PPM1D* mutations in gliomagenesis, we first performed a comprehensive analysis of whole-genome sequences (WGS) of 131 pre-treatment pediatric high-grade gliomas (pHGGs). This consisted of 76 DMGs, 58 of which were prototypical diffuse intrinsic pontine gliomas (DIPG) located in the brainstem, and 55 non-midline pHGGs. We also contrasted the whole-genome sequences of these pretreatment tumors to 39 post-treatment pHGGs (34 DMGs including 33 DIPGs and 5 non-midline HGG).

Analysis of recurrent single nucleotide variants and copy number alterations confirmed the presence of driver alterations that have been previously described in pHGGs (Supplementary Note 1). Consistent with previous reports[3–6], we also observed recurrent *PPM1D* truncating mutations in a subset of pHGG. In our cohort of 170 WGS, we observed *PPM1D* mutations in 8% of all gliomas (14/170) and 11% of DMG (12/110), with enrichment in DMGs (11/91, Fisher's exact test, $P = 0.056$) (Fig. 1A), and confirmed their anti-correlation with *TP53* mutations (Fisher's exact test, $P < 0.0001$) (Supplementary Fig. 1A). All *PPM1D* mutations were nonsense or frameshift mutations in exons 5 or 6 (Fig. 1B), resulting in a truncated form of the protein phosphatase WIP1 which has previously been associated with a gain-of-function phenotype[11,15].

The spectrum of cancers associated with *PPM1D* mutations is distinct in children compared to adults. *PPM1D* mutations have

previously been described in adult myeloid neoplasms including CHIP, myelodysplastic syndrome (MDS), and acute myeloid leukemia (AML)[10,11], but their incidence across other solid cancers is unknown. To assess this, we extended our analysis of *PPM1D* mutations to include other pediatric and adult cancers (Fig. 1C and Supplementary Note 1). Across adult cancers, we observed *PPM1D* mutations to be recurrent in 3% of endometrial cancers (Fig. 1C and Supplementary Fig. 1B). In contrast, *PPM1D* mutations appear to occur predominantly in gliomas in children. Among 41 pediatric cancers encompassing 13 histological subtypes, we found *PPM1D* truncating mutations in only 0.2% of all tumors, with gliomas being the top tumor type (1.37%) (Fig. 1C). Within our DMG WGS cohort, we did not observe differences in the frequency of *PPM1D* mutations between pre- and post-treatment glioma samples (Fisher's exact test, $P = 1$), suggesting a role in enhancing de novo glioma formation. Taken together, these data support truncating *PPM1D* mutations as being contributors of de novo DMG gliomagenesis.

### Endogenous truncation of *Ppm1d*[exon 6] is sufficient to enhance formation of brainstem gliomas in vivo and confers positive selection in vitro

The oncogenic properties of *PPM1D* mutations have not been previously studied in the primary neural context, particularly in primary neural stem and progenitor cells. To address this, we leveraged In-Utero Electroporation (IUE) to induce C′ terminal truncation of endogenous *Ppm1d* to evaluate whether this was sufficient to induce glioma formation.

We performed brainstem targeted IUE of sg*Ppm1d*[exon6] or sg*LacZ* control to evaluate the effects of *PPM1D*tr on gliomagenesis. We observed no differences in mice following C′ terminal truncation compared to sg*LacZ* controls, suggesting that *PPM1D*tr is insufficient to induce glioma formation as a sole driver alteration (Supplementary Fig. 1C). However, *PPM1D* mutations are never observed as sole drivers in DMGs, co-occurring with other alterations including mutated histones and those that result in aberrant growth factor activation. We therefore reasoned that *PPM1D* mutations may cooperate with other oncogenes to enhance gliomagenesis. To evaluate this, we concurrently electroporated sg*Ppm1d*[exon6] with plasmids encoding known drivers of DMGs, including *H3.3*[K27M] and *Pdgfra*[D842V] in fetal brainstems of mice.

Control IUE (guides targeting LacZ with concurrent *H3.3*[K27M] and *Pdgfra*[D842V] PiggyBac plasmids) conditions resulted in a partially penetrant phenotype, with only 50% (9/18) of mice developing neurological symptoms related to tumor, with a median survival of 85 days postnatal (Fig. 1D). In contrast, C-terminal truncation of *Ppm1d* by sg*Ppm1d*[exon6] was sufficient to generate fully penetrant brainstem gliomas, with all mice developing neurological symptoms (17/17) with a median survival of 44 days, representing a significantly shorter latency ($P < 0.0001$ log rank Mantel–Cox test) (Fig. 1D). Additional sg*Ppm1d*[exon6] IUE combinations tested recapitulated prior findings in DNp53 glioma models[16], where expression of *H3.3*[K27M] and sg*Ppm1d*[exon6] was not sufficient to induce gliomas (0/7), while *Pdgfra*[D842V] and sg*Ppm1d*[exon6] drove efficient tumorigenesis (4/7) with an extended latency (median survival of 83 days postnatal, $P < 0.0001$ log rank Mantel–Cox test) (Supplementary Fig. 1C).

sg*Ppm1d*[exon6] IUE brainstem tumors exhibited features that are also observed in human DMGs. GFP-positive sg*Ppm1d*[exon6] IUE brainstem tumors harbored truncations in exon 6 of *Ppm1d* (Supplementary Fig. 1D–F) and displayed histopathological traits of high-grade glioma (Fig. 1E, F). IUE generated murine gliomas showed characteristic histological features seen in human diffuse midline gliomas of the pons (Fig. 1E, F). Tumors exhibited diffuse single-cell infiltration of the brainstem parenchyma with moderate

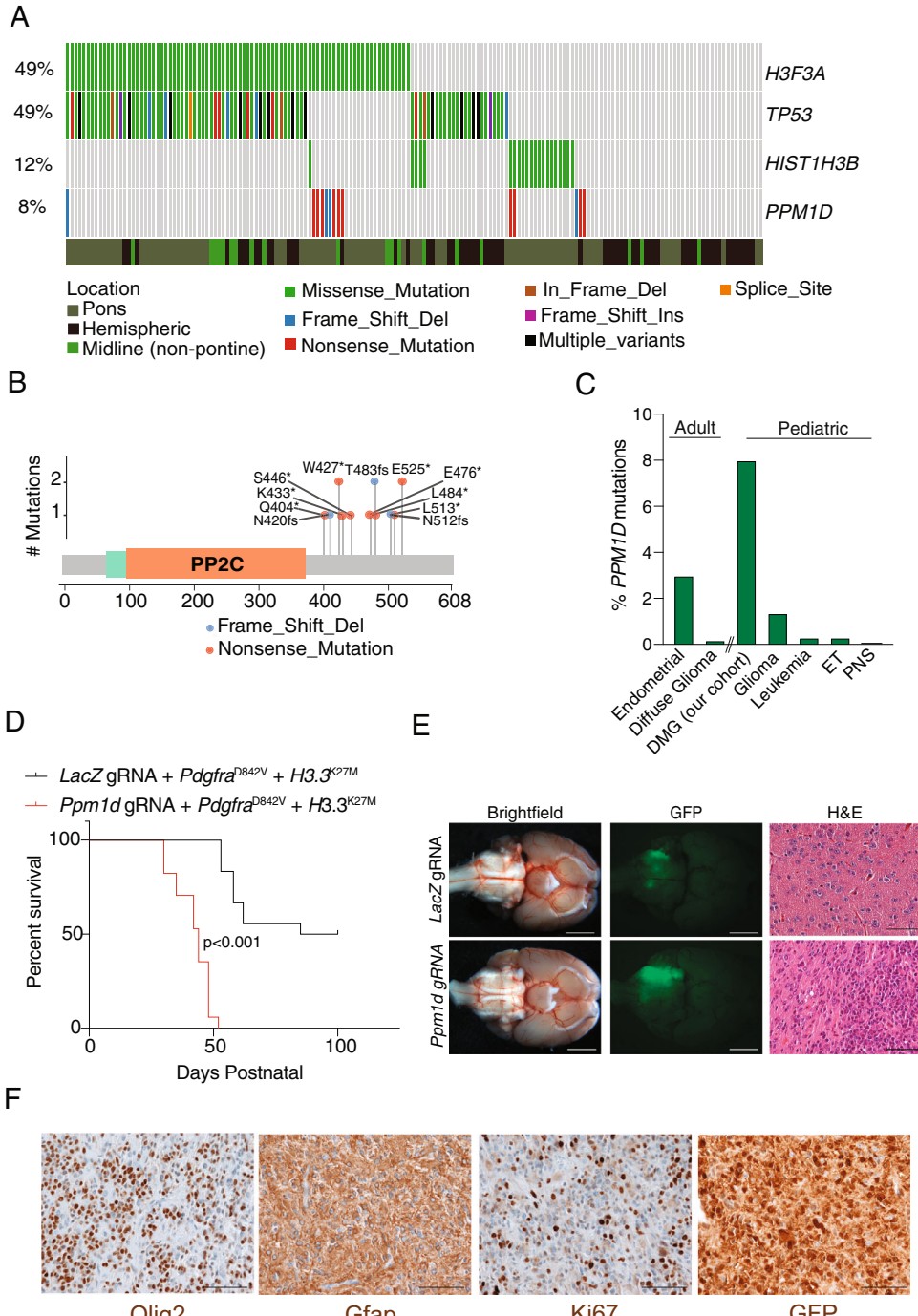

**Fig. 1 PPM1D mutations are oncogenic drivers of DMGs. A** Comutation plot showing alterations in histones (*H3F3A* and *HIST1H3B*), *TP53*, and *PPM1D* in 170 midline and non-midline gliomas. Variants observed are depicted below. **B** Lollipop plot of recurrent C-terminus *PPM1D* truncating mutations observed in DMGs. The position of nucleotide variants are shown. *Depicts nonsense or truncating alterations. **C** Percentage of samples with *PPM1D* truncating mutations across different adult and pediatric tumors including DMG dataset from our study. **D** Kaplan–Meier survival curves for *H3.3*[K27M]+*Pdgfra*[D842V] IUE DMG mouse models with *LacZ* gRNA (*n* = 18) or *Ppm1d* gRNA (*n* = 17). *P* < 0.0001 between *LacZ* gRNA vs *Ppm1d* gRNA conditions calculated using log-rank Mantel–Cox test. **E** Brightfield and GFP images of *LacZ* and *Ppm1d* gRNA IUE DMGs showing GFP-positive tumor regions, and H&E-stained images depicting high-grade glioma histology. Scale bar denotes 2.5 mm (brightfield and GFP) and 50 μm (H&E). Similar staining was performed in a minimum of three independent samples. **F** *Ppm1d* gRNA IUE DMG sections stained with Olig2, Gfap, Ki67, or GFP. Scale bar denotes 50 μm. Similar staining was performed in a minimum of three independent samples. Source data are provided as a Source Data file.

atypia and mitoses consistent with a high-grade glioma and equivalent to WHO 2016 Grade 3 or 4. Immunohistochemical analysis (Fig. 1F) showed the expression of GFP was retained in tumor cells uniformly and at high levels and cells retained defining lineage markers of DMG including diffuse Olig2 and Gfap

labeling. The proliferation rate by Ki67 staining showed more than 50% of glioma cells positive, similar to human DMGs.

We next compared mouse and human DMG transcriptomes. We leveraged RNA-sequencing of *sgPpm1d*[exon6] IUE brainstem tumors to identify highly expressed genes and compared this

profile with that of *PPM1D*-mutant human tumors. 209/14481 genes were found to be highly expressed (Z-score > 2) in the sg*Ppm1d*$^{exon6}$ tumors (Fig. 2A and Supplementary Data 1). 152 of these 209 genes were also highly expressed (Z-score > 2) in *PPM1D*-mutant human DMGs (Fig. 2B), representing a significant overlap between the two conditions ($p < 0.0001$). Taken together with the histopathological comparisons of these tumors, *Ppm1d*-mutant IUE brainstem tumors are representative of the tumors observed in human DMG patients.

C' terminal truncation of *Ppm1d* was also sufficient to exert a positive growth advantage in mouse neural stem cells (NSCs) obtained from E14.5 embryos following IUE to induce *Ppm1d* truncation (Supplementary Fig. 2A, B) when cultured ex vivo. In the absence of selective pressures, the use of CRISPR-Cas9 is expected to induce out-of-frame indels in a proportion of edits. As these indels would lead to truncation of *Ppm1d*, we would expect them to confer positive selection pressures. To assess this, we evaluated the change in proportions of truncating *Ppm1d* events to in-frame INDELS between an early time point (within one month of transfection) and a late time point (following culture for another two months). Indeed, we observed an enrichment of *Ppm1d* truncating INDELs across time, indicating a positive selection advantage. The cells generated by endogenously truncating *Ppm1d* at exon 6 (sg*Ppm1d*) showed significantly more frameshift modifications compared to cells with non-targeting guides against *LacZ* (sg*LacZ*) at an early timepoint after establishing the cell line (91.1% ± 0.4% vs 20.9% ± 2.2%, $P < 0.05$ two-tailed unpaired *t*-test) (Supplementary Fig. 2C–E). By the late time point, the proportion of frameshift *Ppm1d* edits increased to encompass the majority of sequencing reads (99.7% ± 0.2%) in the cells that were transfected by sg*Ppm1d*, but not in the cells transfected with sg*LacZ* (vs 3.7% ± 3.2%) (Supplementary Fig. 2C–E), consistent with positive selection of *Ppm1d* truncating INDELs.

**Overexpression of *PPM1D*tr is sufficient to potentiate gliomagenesis and requires the *PPM1D* phosphatase domain.** C-terminal *PPM1D* truncating mutations remove a negative regulatory domain which has been shown to increase the stability of the phosphatase[15]. It is possible that *PPM1D* mutations are oncogenic specifically through expression of truncated *PPM1D*, or that simply an increase in wild-type phosphatase activity through over-expression can contribute to oncogenesis. We addressed this question by evaluating whether the exogenous expression of human truncated *PPM1D* (*PPM1D*tr) was also sufficient to promote glioma formation in vivo. Compared to *H3.3*$^{K27M}$ and *Pdgfra*$^{D842V}$ control IUE conditions, the addition of *PPM1D*tr resulted in enhanced development of fully penetrant gliomas with significantly shortened latency ($P < 0.005$, log-rank Mantle-Cox test; median survival 63 vs. 46 days postnatal) (Fig. 2C and D). *PPM1D*tr IUE DMG models displayed a similar latency to DMG models generated by IUE with the addition of a dominant-negative p53 construct (DN*p53*), which we have previously shown to efficiently cooperate with *H3.3*$^{K27M}$ and *Pdgfra*$^{D842V}$ overexpression[16] (median survival 46 vs. 44 days postnatal, ns, $P = 0.37$, log-rank Mantle-Cox test) (Supplementary Fig. 3A). IUE of full-length *PPM1D* (*PPM1D* FL) with *H3.3*$^{K27M}$ and *Pdgfra*$^{D842V}$ plasmids displayed a trend for shortened latency compared to control conditions but did not reach statistical significance (median survival 51 vs. 63 days postnatal, ns, $P = 0.14$, log-rank Mantle-Cox test) (Supplementary Fig. 3A). This may reflect known differences in *PPM1D* FL and *PPM1D*tr stability[11,15].

The PPM1D protein domains that contribute to its oncogenic effects to enhance glioma formation have not been evaluated. To test whether PPM1D phosphatase activity is required for accelerated glioma formation in our models, we performed IUE of *H3.3*$^{K27M}$ and *Pdgfra*$^{D842V}$ plasmids along with a mutant form of *PPM1D*tr (*PPM1D*tr-D314A), that inactivates the phosphatase activity[15]. In contrast to overexpression of *PPM1D*tr with a wild-type protein phosphatase domain, overexpression of *PPM1D*tr-D314A was not sufficient to enhance glioma formation relative to IUE of *H3.3*$^{K27M}$ and *Pdgfra*$^{D842V}$ together, and overall survival rates of mice transduced with the phosphatase dead *PPM1D*tr construct did not differ compared to controls (85 vs 63 days postnatal, ns, $P = 0.7$, Fig. 2C).

Tumors collected from each IUE condition displayed high-grade glioma histology, expressed Olig2, and contained Gfap-positive reactive astrocytes (Fig. 2E). Compared to control IUE DMG models, *PPM1D*tr tumors displayed a significant increase in Ki67-positive proliferating cells (30.59% ± 1.9% vs 12.42% ± 1.5%; $P < 0.0001$, two-tailed unpaired t-test) (Fig. 2E, F). Moreover, *PPM1D*tr-D314A tumors displayed a similar rate of proliferating Ki67-positive cells to that of control IUE conditions (12.6% ± 2.2% vs 12.42% ± 1.5%; ns, $P = 0.095$ two-tailed unpaired *t*-test) (Fig. 2E, F), suggesting that the *PPM1D* phosphatase activity is required to enhance the growth rate of IUE DMG mouse models.

From these experiments, we conclude that the endogenous truncation of the C' terminal regulatory region of *Ppm1d* by CRISPR-Cas9 or ectopic expression of truncated *PPM1D* are similarly sufficient as expression of dominant-negative *TP53* to potentiate the in vivo formation of brainstem gliomas in the setting of *H3.3*$^{K27M}$ and *Pdgfra*$^{D842V}$. Moreover, the PPM1D phosphatase is necessary to enhance DMG formation.

**Expression of *PPM1D* is necessary for proliferation of *PPM1D*-mutant DMG cells.** Our findings suggest that *PPM1D* mutations are a clonal driver event that enhances DMG formation. We, therefore, evaluated whether the expression of *PPM1D* was also necessary for the proliferation of an established *PPM1D*-mutant DMG cell line BT869 (Supplementary Fig. 4A, B and Supplementary Data 2). *PPM1D* mutant BT869 cells infected with two independent *PPM1D*-targeting guides exhibited significantly reduced rates of proliferation (mean fold changes from day 0 to day 17, 2.4 ± 0.2 and 1.5 ± 0.1 respectively) compared to those infected with non-targeting control guides (mean fold change 9.1 ± 0.9) ($P < 0.005$ for both *PPM1D*-targeting guides compared to the control guide) (Fig. 3A). The magnitude of this antiproliferative effect was similar to our positive control guides which targeted the essential gene *EXOSC8* (mean fold change for BT869 from day 0 to day 17, 2.1 ± 0.1, $P < 0.005$). In contrast, *PPM1D*-WT DMG lines SU-DIPG-IV and SU-DIPG-XIII were not dependent on *PPM1D* expression (SU-DIPG-IV: mean fold changes from day 0 to day 22 for sgRNA *PPM1D* #1 and #2 and sgRNA *LacZ* 77.5 ± 3.3, 82.8 ± 5.4, and 76.3 ± 18.6 respectively, $P = 0.9$ and $P = 0.6$ respectively. SU-DIPG-XIII: mean fold changes from day 0 to day 11 for sgRNA *PPM1D* #1 and #2 and sgRNA *LacZ* 17.6 ± 1.4, 16.4 ± 3.9, and 20.6 ± 1.5 respectively, $P = 0.2$ and $P = 0.4$ respectively) (Fig. 3B and Supplementary Fig. 4C).

To further complement our findings, we used a next-generation sequencing (NGS)-based competition assay to track the selection of *PPM1D*-disrupted alleles relative to unedited or in-frame modifications. In this assay, the relative abundance of loss-of-function CRISPR-Cas9 mediated edits can be used to determine whether the expression of *PPM1D* is necessary for proliferation. In the *PPM1D*-mutant line BT869, the proportion of loss-of-function *PPM1D* alterations decreased over time (33% ± 2.2% at early time point vs. 16% ± 2.4% at late time

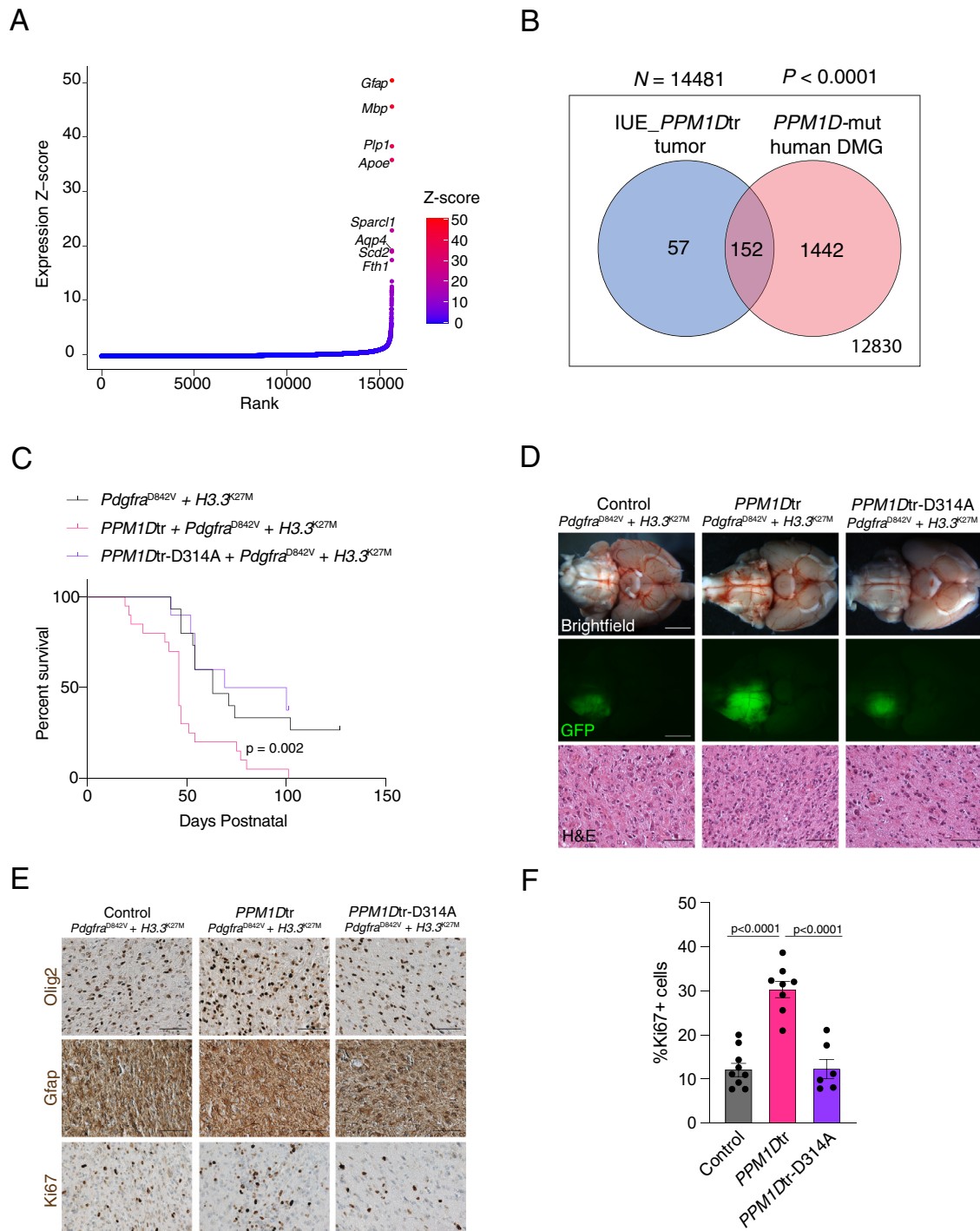

**Fig. 2 Expression of human *PPM1D*tr potentiates DMG formation in vivo and requires the *PPM1D* phosphatase domain. A** Plot showing ranked list of genes based on their expression Z-score in *Ppm1d* gRNA IUE DMG ($n = 3$ tumors). Top genes with expression Z-score >15 are depicted. **B** Venn diagram showing highly expressed genes (Z-score > 2) in *Ppm1d* gRNA IUE DMG ($n = 3$), *PPM1D*-mutant human DMGs ($n = 7$), and the overlap between two datasets. Total number of genes analyzed in both datasets is shown outside the box. $P < 6.27e-100$ calculated using hypergeometric distribution. **C** Kaplan–Meier survival curves for control ($n = 19$), *PPM1D*tr ($n = 20$), and *PPM1D*tr-D314A ($n = 10$) IUE DMG mouse models. $P = 0.002$ between control vs. *PPM1D*tr conditions calculated using log-rank Mantel–Cox test. **D** Brightfield and GFP images of control, *PPM1D*tr, and *PPM1D*tr-D314A IUE DMGs showing GFP-positive tumor regions, and H&E-stained images depicting high-grade glioma histology. Scale bar denotes 2.5 mm (brightfield and GFP) and 50 µm (H&E). Similar staining was performed in a minimum of three independent samples. **E** Control, *PPM1D*tr, and *PPM1D*tr-D314A IUE DMG sections stained with Olig2, Gfap, or Ki67. Scale bar denotes 50 µm. **F** Quantification of the percentage of Ki67-positive cells in Control ($n = 9$), *PPM1D*tr ($n = 8$), and *PPM1D*tr-D314A ($n = 6$) IUE DMG models. Data presented as mean ± SEM. $P < 0.0001$ for both control vs *PPM1D*tr and *PPM1D*tr vs *PPM1D*tr-D314A conditions calculated using two-tailed *t*-test. Source data are provided as a Source Data file.

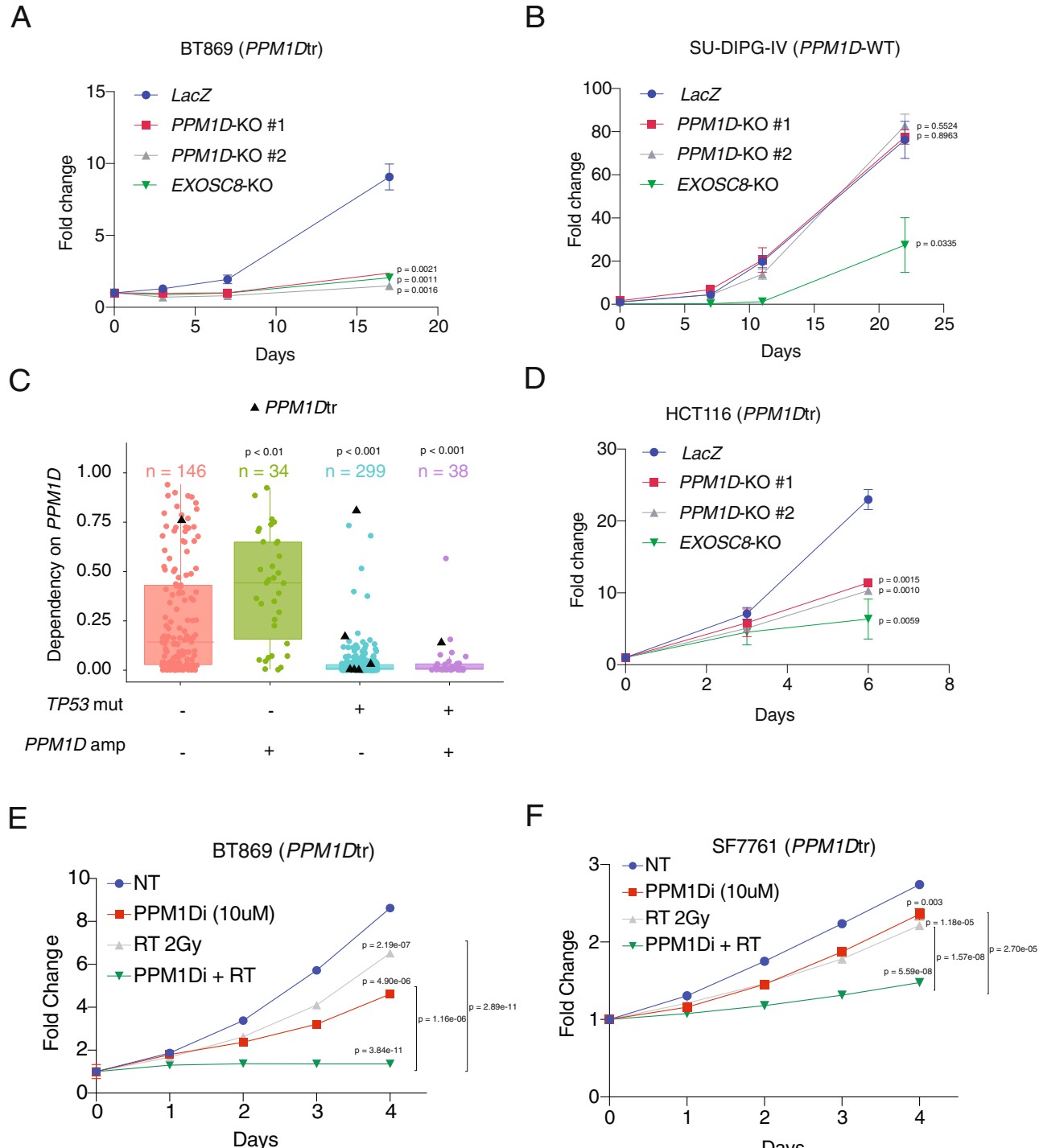

point) (Supplementary Fig. 4D), suggesting that *PPM1D*-mutant DMG cells require the expression of *PPM1D* for proliferation. However, we did not observe a similar decrease in loss-of-function alterations of *PPM1D* in *PPM1D*-WT cell lines SU-DIPG-IV and SU-DIPG-XIII (Supplementary Fig. 4E, F).

Our finding that *PPM1D* is necessary for *PPM1D*-mutant DMGs extended to other *PPM1D*-expressing cancers. We examined whether *PPM1D* dependency was linked to genetic activation of *PPM1D* using pooled CRISPR-cas9 assays across 558 cancer cell lines[17,18]. We hypothesized that *TP53*-WT cell lines would be more dependent on *PPM1D* than *TP53*-mutant cell lines because of PPM1D's role in opposing p53 function. As expected, lines with wild-type *TP53* were significantly more

dependent on *PPM1D* than *TP53*-mutant lines ($P < 0.0001$) (Fig. 3C). Among *TP53*-WT lines, *PPM1D* copy-number gain was associated with significantly higher probability of dependency on *PPM1D* ($P < 0.01$) (Fig. 3C). We conclude that *PPM1D* is required for the proliferation of p53 wild-type cell lines, particularly those that harbor *PPM1D* activating genetic alterations.

However, the cell lines in this dataset did not fully represent *PPM1D*-mutant DMGs as almost all non-DMG cell lines containing *PPM1D*tr mutations also harbored *TP53* mutations (7/8 lines) which may be acquired during cell passaging[19]. We therefore also validated this dependency in a *PPM1D*tr colon cancer cell line (HCT116), which does not harbor a *TP53*

**Fig. 3 PPM1D is a dependency in PPM1Dtr but not PPM1D-WT cell lines. A** Fold change in proliferation compared to day 0 of *PPM1D*-mutant patient-derived cell line (PDCL) BT869 after transfection with *PPM1D*-KO sgRNAs, nontargeting (*LacZ*) sgRNA, or lethal (*EXOSC8*) sgRNA. Growth curves show mean ± SEM from three replicates and are representative of three independent experiments. $P = 0.0021$, 0.0011, and 0.0016 for *PPM1D*-KO #1, *PPM1D*-KO #2, and *EXOSC8* respectively calculated using two-tailed *t*-test. **B** Growth of *PPM1D*-WT PDCL SU-DIPG-IV after transfection with indicated sgRNAs. Growth curves show mean ± SEM from three replicates and are representative of three independent experiments. $P = 0.8963$, 0.5524, and 0.0335 for *PPM1D*-KO #1, *PPM1D*-KO #2, and *EXOSC8* respectively calculated using two-tailed *t*-test. **C** Probability of *PPM1D* dependency as determined by pooled CRISPR-cas9 assays across 558 cancer cell lines, subgrouped according to their *TP53* mutation and *PPM1D* copy-number status. Bounds of the box represent the IQR, center represents the median, and the bounds of the whiskers represent 1.5 times IQR. $P < 0.0001$ for *TP53* WT/*PPM1D* non-amplified cells vs both *TP53*-mutant/*PPM1D*-amplified and *TP53*-mutant/*PPM1D* non-amplified cells and $P < 0.01$ for *TP53* WT/*PPM1D* non-amplified cells vs *TP53* WT/*PPM1D*-amplified cells using Kruskal–Wallis test. **D** Growth of *PPM1D*-mutant colon cell line HCT116 after transfection with indicated sgRNAs. Growth curves show mean ± SEM from three replicates and are representative of three independent experiments. $P = 0.0015$, 0.0010, and 0.0059 for *PPM1D*-KO #1, *PPM1D*-KO #2, and *EXOSC8* respectively calculated using two-tailed *t*-test. **E–F** Growth of *PPM1D*-mutant DMG cell lines BT869 (**E**) and SF7761 (**F**) after treatment with vehicle control (NT), 10 μM of GSK2830371 (PPM1Di), 2 Gy of ionizing radiation treatment (RT), or the combination of both (GSK + RT). Growth curves show mean ± SEM from at least three replicates and are representative of three independent experiments. BT869: $P = 4.90083E{-}06$, $2.18655E{-}07$, $3.83657E{-}11$, $2.8877E{-}11$, and $1.1625E{-}06$ for NT vs PPM1Di, NT vs RT, NT vs PPM1Di + RT, RT vs PPM1Di + RT, and PPM1Di vs PPM1D + RT respectively calculated using two-tailed *t*-test; SF7761: $P = 0.003367278$, $1.17616E{-}05$, $5.59401E{-}08$, $1.5694E{-}08$, and $2.698E{-}05$ for NT vs PPM1Di, NT vs RT, NT vs PPM1Di + RT, RT vs PPM1Di + RT, and PPM1Di vs PPM1D + RT respectively calculated using two-tailed *t*-test. Source data are provided as a Source Data file.

mutation. Similar to our observations in *PPM1D*-mutant DMGs, disruption of *PPM1D* using guides directed to the *PPM1D*tr phosphatase domain resulted in reduced growth compared to controls (mean fold changes from day 0 to day 6 for *PPM1D* sgRNA #1, *PPM1D* sgRNA #2, sgRNA *LacZ*, and sgRNA *EXOSC8* were 11.4 ± 0.5, 10.3 ± 0.4, 23 ± 1.4, 6.4 ± 2.8 respectively, $P < 0.005$ for both *PPM1D* guides and $P < 0.05$ for sg*EXOSC8*) (Fig. 3D). Taken together, these findings confirm *PPM1D* expression to be necessary for the proliferation of *PPM1D*-mutant cancer cells, nominating it as a potential therapeutic target across multiple lineages.

We also found small-molecule inhibition of *PPM1D* to further suppress the viability of *PPM1D*-mutant DMG cells in the presence of ionizing radiation (IR). GSK2830371 is a tool *PPM1D* inhibitor that has been shown to antagonize *PPM1D* function in cancer cell lines[20,21]. We treated *PPM1D*-mutant DMG cell lines BT869 and SF7661 with 10 μM of GSK2830371 (GSK), 2 Gy of IR, or the combination of both. Inhibition of *PPM1D* with GSK or treatment with IR by themselves decreased the viability of the cells compared to no treatment (NT) control (BT869: mean fold change from day 0 to 5: 4.6 ± 0.2 for NT, 6.5 ± 0.1 for GSK ($P < 0.0001$), and 8.6 ± 0.1 ($P < 0.0001$) for IR respectively) (Fig. 3E); SF7761 mean fold change from day 0 to 5: 2.4 ± 0.1 for NT, 2.2 ± 0.0 for GSK ($P < 0.0005$), and 2.7 ± 0.0 for IR ($P < 0.0001$) respectively (Fig. 3F). However, the combination of these two treatments further decreased the viability of these cells over time (BT869: mean fold change at day 5 1.4 ± 0.1 ($P < 0.0001$); SF7761 mean fold change at day 5 1.5 ± 0.0 ($P < 0.0001$)) (Fig. 3E, F), exhibiting a possible additive effect. These findings suggest that the induction of DNA-damage through IR further accentuates the necessity of *PPM1D* expression in *PPM1D*-mutant DMGs.

**PPM1Dtr suppresses apoptosis and potentiates progression of cells through the G1/S checkpoint following ionizing radiation treatment.** *PPM1D* has been shown to have multiple substrates that regulate cellular functions, including regulation of cell cycle progression, DDR, and the induction of apoptosis[8–11]. We, therefore, reasoned that *PPM1D*tr may act through these pathways to enhance the cellular proliferation of *PPM1D*-mutant DMGs. To evaluate the effects of *PPM1D*tr overexpression, we transduced in vitro primary mouse neural stem cell models (mNSCs) that ectopically express K27M mutant *H3F3A* (*H3K27M*) with *PPM1D*tr-, *PPM1D* FL-, or a green fluorescent protein (GFP) control-expressing vector. We were able to achieve

successful consistent overexpression of *PPM1D*tr but not *PPM1D* FL (Supplementary Fig. 5A), likely because full-length *PPM1D* is actively degraded as previously reported[11,15].

Expression of *PPM1D*tr was associated with attenuated apoptosis and more rapid cell cycling after IR treatment. We treated our mNSC models with IR and assessed the percentage of apoptotic and cycling cells using flow-cytometry analysis of Annexin V/Propidium Iodide (PI) and BrdU/7-AAD respectively (Supplementary Fig. 5B–G). Compared to GFP controls, *PPM1D*tr overexpression led to significantly lower rates of apoptosis at baseline (11.5% ± 1.3% and 6.1% ± 0.9% respectively; $P < 0.05$), and with even greater effects 24 h after 8 Gy of IR treatment (22.9% ± 2.3% and 13.3% ± 2.4%, respectively; $P < 0.05$) (Fig. 4A). *PPM1D* FL overexpression did not lead to a significant decrease in apoptosis at baseline, but decreased apoptosis 24 h after 8 Gy of IR treatment (12.8 ± 2.3; $P < 0.05$). At 24 h post-IR treatment, the majority of GFP expressing cells remained in G0/G1 (77.9% ± 7.1%), while only 6.3% ± 1.5% had re-entered cell-cycling and were in S-phase. However, cells expressing *PPM1D* FL and *PPM1D*tr exhibited a more rapid progression through the G1/S checkpoint, with 18% ± 1.4 and 17% ± 1.5% of cells respectively, observed to be in S phase ($P < 0.05$ and $P < 0.005$ respectively) (Fig. 4B). The suppression of apoptosis and G1/S cell cycle checkpoint by *PPM1D* FL suggests that even low levels of *PPM1D* overexpression might be enough to confer this difference in the context of DNA damage.

*PPM1D* truncations have been shown to stabilize PPM1D, leading to enhanced dephosphorylation of its substrates[11,15]. Moreover, we observed the PPM1D phosphatase to be necessary for *PPM1D*tr to enhance DMG formation. We, therefore, reasoned that *PPM1D* truncations associated with *PPM1D*-mutant gliomas may also exhibit the enhanced activity of the PPM1D phosphatase, targeting known substrates in the DDR and cell cycle pathways. Expression of *PPM1D*tr was associated with reduced phosphorylation of PPM1D substrates in the DDR pathway, in both primary mNSC models and patient-derived DMG cell lines. We treated mNSCs generated by endogenously truncating *Ppm1d* at exon 6 (sg*Ppm1d*) or non-targeting guides against *LacZ* (sg*LacZ*) with IR and assessed response to DNA damage using previously described markers γ-H2AX and p-p53 (Ser15) at both baseline and five-hours post-IR treatment (Supplementary Fig. 6A). We observed decreased phosphorylation of these markers in mNSCs with truncated *Ppm1d* in both conditions ($P < 0.05$) (Supplementary Fig. 6A, B). We also found similar decrease in phosphorylation of these markers in mNSCs

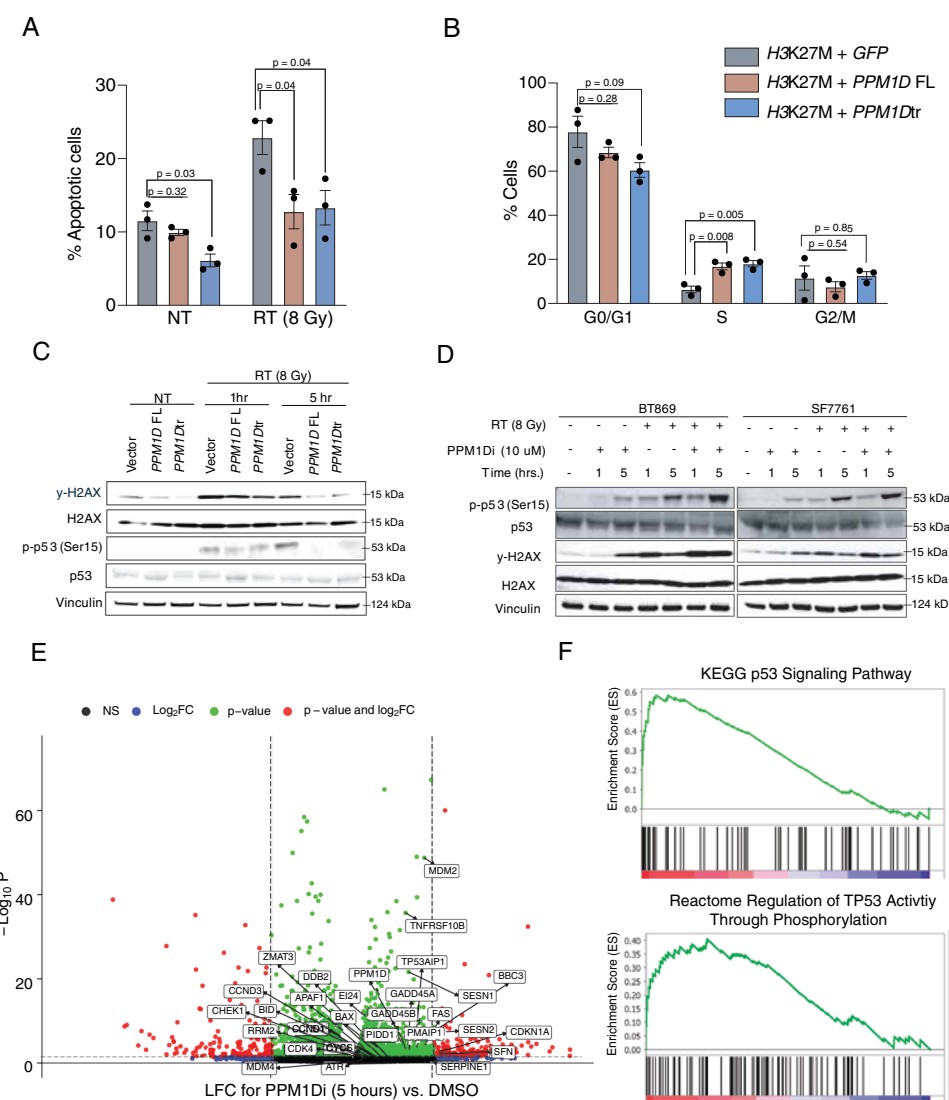

**Fig. 4 *PPM1D* suppresses apoptosis, DDR, and p53 pathways.** Apoptosis and cell cycle responses were assessed in isogenic mouse neural stem cells (mNSCs) overexpressing *H3F3A* K27M (*H3*K27M) mutation plus either eGFP (vector control), full-length *PPM1D* (*PPM1D* FL), or C-terminus truncated *PPM1D* (*PPM1D*tr) (**A**–**B**). **A** Cells were treated with 8 Gy of ionizing radiation (RT) or no treatment (NT) for 24 h, incubated with Annexin V-APC and Propidium Iodide (PI), and analyzed using flow cytometry. The total percentage of Annexin V positive cells representing both early and late apoptotic cells are shown. Data presented as mean ± SEM from three biological replicates. *P* = 0.027, 0.318, 0.044, 0.037 for *GFP* vs *PPM1D*tr (NT), *GFP* vs *PPM1D* FL (NT), *GFP* vs *PPM1D*tr (RT) and *GFP* vs *PPM1D*-FL (RT) conditions respectively calculated using two-tailed t-test. **B** Cells were treated with 8 Gy RT for 24 h and integration of BrdU was assessed to determine the percentage of cells in S-phase. Anti-BrdU APC and 7-AAD DNA staining were used to distinguish cells in each stage of the cell cycle. Data presented as mean ± SEM from three biological replicates. *P* = 0.276, 0.090 0.008, 0.005, 0.543, 0.848 for *GFP* vs *PPM1D* FL (G0/G1) *GFP* vs *PPM1D*tr (G0/1), *GFP* vs *PPM1D* FL (S) *GFP* vs *PPM1D*tr (S), *GFP* vs *PPM1D* FL (G2/M) *GFP* vs *PPM1D*tr (G2/M) conditions respectively calculated using two-tailed *t*-test. **C** Cells were treated with 8 Gy of RT and lysates were collected at baseline (NT), 1 and 5 h post-RT respectively, and probed with the indicated antibodies. Three independent experiments were performed with similar results. **D** *PPM1D*-mutant patient-derived DMG cell lines BT869 and SF7761 were treated with 10 μM of GSK2830371 (PPM1Di) and/or 8 Gy of RT for 1 and 5 h, lysed and probed with indicated antibodies. Three independent experiments were performed with similar results. **E** Volcano plot showing differentially expressed genes in BT869 and SF7761 DMG cells (*n* = 3 per cell line per condition) treated with 10 μM of GSK2830371 (PPM1Di) for 5 h compared to vehicle treatment. Genes in the p53 signaling pathway are labeled. **F** GSEA enrichment plots of two p53 related pathways significantly enriched (FDR < 0.25) after inhibition of PPM1D in BT869 and SF7761 cells. Source data are provided as a Source Data file.

overexpressing *PPM1D*tr compared to those overexpressing GFP (*P* < 0.05) (Fig. 4C and Supplementary Fig. 6C). Next, we leveraged the tool PPM1D inhibitor to evaluate whether suppression of PPM1D was sufficient to reverse this phenotype. We treated two patient-derived *PPM1D*-mutant DMG cell lines, BT869 and SF7761 with the PPM1D inhibitor GSK2830371 at 10 μM. Treatment with GSK2830371 by itself for 5 h increased the levels of γ-H2AX and p-p53 (Ser15) in *PPM1D*-mutant DMG lines (*P* < 0.05) and this effect was further enhanced by IR

treatment (Fig. 4D and Supplementary Fig. 6D). Taken together, these data indicate that *PPM1D*tr confers changes on cellular processes that converge on regulation of cell-cycle progression and DDR including induction of apoptosis.

**PPM1Dtr inhibition leads to increased transcription of *TP53* and DDR pathways in *PPM1D* mutant DMGs.** To evaluate the transcriptional changes conferred by *PPM1D*tr in DMGs, we

treated *PPM1D*-mutant DMGs BT869 and SF7761 with 10 μM of GSK2830371 and compared their gene-expression profiles to those of DMSO treated cells at 1, 5, and 24 h post treatment. We observed 114, 368, and 219 genes to be differentially expressed at 1, 5, and 24 h post treatment (LFC > 2, FDR < 0.25) in the GSK2830371 treated cells compared to vehicle controls (Fig. 4E, Supplementary Fig. 7A and Supplementary Data 3). The most significantly upregulated genes five hours post treatment included the chemokine *CXCL8*, *ZBTB32*, a member of the ZBTB family of transcription factors and *TRIML2*, which has been reported to enhance p53 SUMOylation[22], while the most downregulated genes included the monocarboxylate transporter *SLC16A3* and *MYOD1*, a bHLH transcription factor most well-known for its role in regulating myogenic differentiation[23] (Supplementary Data 3). At a pathway level, 353/1836, 176/1453, and 90/1858 pathways were significantly upregulated (FDR < 0.25) in the GSK2830371 treated condition relative to DMSO condition at 1, 5, and 24h time points respectively. Five hours post treatment, 8 of the 10 most significantly upregulated pathways included gene-sets associated with *TP53*-signaling and one with DDR. Moreover, of the remaining 166 pathways, another 10 were also related to *TP53* signaling, 12 with cell cycle regulation, and 9 with DDR (Fig. 4F, Supplementary Fig. 7B, C and Supplementary Data 4). Ten of these signaling pathways remained upregulated at the 24 h time point.

**PPM1Dtr dephosphorylates substrates in the DDR and cell cycle regulation pathways in *PPM1D*-mutant DMGs.** To further evaluate the magnitude of the effects of PPM1Dtr on proteins involved in the DNA damage and cell cycle responses relative to the entire phosphoproteome, we leveraged multiplexed quantitative mass-spectrometry approaches to systematically identify all substrates that are differentially altered in the presence of PPM1Dtr.

We first quantitatively identified proteins and phosphosites associated with the PPM1D phosphatase function in the *PPM1D*-mutant cell line BT869. Cells treated with 10 μM of the PPM1D inhibitor GSK2830371 for 5 h exhibited differential alteration of 184 phosphosites relative to DMSO controls (LFC > 1 and FDR < 0.01) (Fig. 5A and Supplementary Data 5). Some of these phosphosites included CHEK2 (S303, S478), TP53 (S15, S9 S15), TP53B1 (S1073, S1764), MDC1 (S955), SMC3 (S1067, S1065 S1067), MCM3 (S726, S580), and RAD18 (S142, S403) (Fig. 5A and Supplementary Data 5). Functional protein association of the top 50 differentially altered proteins using STRING[24] revealed that these targets were enriched for 89 biological processes (FDR < 0.05), including DNA repair, cell cycle, DNA damage checkpoint, DNA damage response by p53, G1/S transition of the mitotic cell cycle, and G2/M transition of the mitotic cell cycle (Fig. 5B, C and Supplementary Data 6).

However, these analyses are based on the identity of the protein and do not consider the status of specific phosphorylation sites, which confer functional significance. To address this, we also performed enrichment analysis of post-translational modification (PTM) sites using the recently developed PTM-SEA method[25]. We found 17 PTM pathways to be significantly altered (FDR < 0.05), including seven pathways related to cell cycle and CDK signaling (CDK1, CDK2, CK2A2/CSNK2A2, CK2A1/CSNK2A1, AurA/AURKA, and AurB/AURKB, TTK), four pathways related to DDR such as ATM, ATR, and CHEK2 signaling, and another two pathways associated with receptor tyrosine kinase signaling (Fig. 5D). We next applied a recently published method CausalPath[26] to integrate our phosphoproteomic data with literature knowledge to generate causal hypotheses from the data. This analysis also indicated several DNA repair genes (ATM,

ATR, MDC1, MRE11) to be activated upon PPM1D inhibition (Supplementary Fig. 8A). Some of the observed phosphorylations of TP53 and TRIM28 are identified to be a consequence of PPM1D inhibition (Supplementary Fig. 8A). We also observed some inhibitory sites on PPM1D (S54 and S85) to be down-regulated, possibly due to some negative feedback on PPM1D (Supplementary Fig. 8A). Furthermore, sequence motif analysis identified a conserved SQ motif among putative PPM1D-dependent phosphorylation events ($P = 1.6e{-}49$) (Fig. 5E), further supporting the role of PPM1D in response to DNA damage[27]. Consistent with the previous reports[11], we also observed an overrepresentation of glutamic acid at position +2 ($P = 4.2e{-}19$).

We also characterized the phosphoproteome of mNSCs transduced to express PPM1Dtr. We identified 1420 unique phosphosites to be differentially altered in PPM1Dtr cells compared to GFP cells (LFC > 1 and FDR < 0.01), including phosphosites in Tp53bp1 (S571, S1750), Mdm2 (T286 S288, S183), Smc3 (S1067, S1065 S1067), and Atrx (S822) (Supplementary Fig. 8B and Supplementary Data 7). We also found 18 PTM pathways to be significantly altered in PPM1Dtr cells compared to GFP (FDR < 0.25), including three pathways related to CDK signaling (CDK2, CDK4, and CDK5), consistent with our observations in the phosphoproteome of human DMG (Supplementary Fig. 8C). Taken together, these findings confirm that expression of PPM1Dtr is predominantly associated with altered phosphorylation in the regulators of TP53-related pathways, including cell cycle and DDR pathways.

**Cell cycle, *TP53* associated, and DDR pathways represent genetic dependencies in *PPM1D*tr-expressing mNSCs.** Our findings nominate *PPM1D*tr as being sufficient for enhancement of glioma formation and necessary for proliferation of *PPM1D*tr cells. Moreover, we find DDR, *TP53,* and cell cycle pathways to be associated with the functional effects of *PPM1D*tr. We, therefore, reasoned that these pathways would also be necessary for the proliferation of *PPM1D*tr-expressing cells.

To evaluate this, we performed a genome-scale CRISPR/Cas9 loss-of-function screen in mNSCs overexpressing *PPM1D*tr to identify genes whose ablation resulted in a decreased fitness of the cells. We found the ablation of 19 genes to result in a positive selection advantage (LFC > 1, FDR < 0.25), while ablation of another 1588 genes suppressed their growth (LFC < −2, FDR < 0.25) (Fig. 6A, Supplementary Data 8). However, 893 of these latter genes have been identified as pan-essential dependencies across all cell types, leaving 695 genes that are more likely to be *PPM1D*tr-associated dependencies. Gene-set enrichment analysis (GSEA) of the genes identified as not being pan-essential revealed 241 canonical pathway gene sets (FDR < 0.25) to be significantly enriched among this set. At least 38 of these gene sets included pathways representing cell cycle/cell cycle checkpoints, DDR, and the *TP53* pathway (Fig. 6B, Supplementary Data 9). We observed similar findings when the analyses were repeated with more stringent thresholds (Supplementary Data 9). In contrast, no gene-sets (0/19 significant gene sets, FDR < 0.25) in these pathways were enriched in dependencies identified in wild-type NSCs (Supplementary Data 10).

These results support the expression of cell cycle, DDR, and *TP53* pathway members as being necessary for the proliferation of *PPM1D*tr-expressing mNSCs. However, cancer cells frequently harbor other genetic alterations that can also influence vulnerabilities associated with oncogenes that are not in our primary mNSC models. We, therefore, sought to expand our analyses to include cancer cell lines, leveraging genome-scale CRISPR-Cas9 screens completed across 558 independent models[17]. We

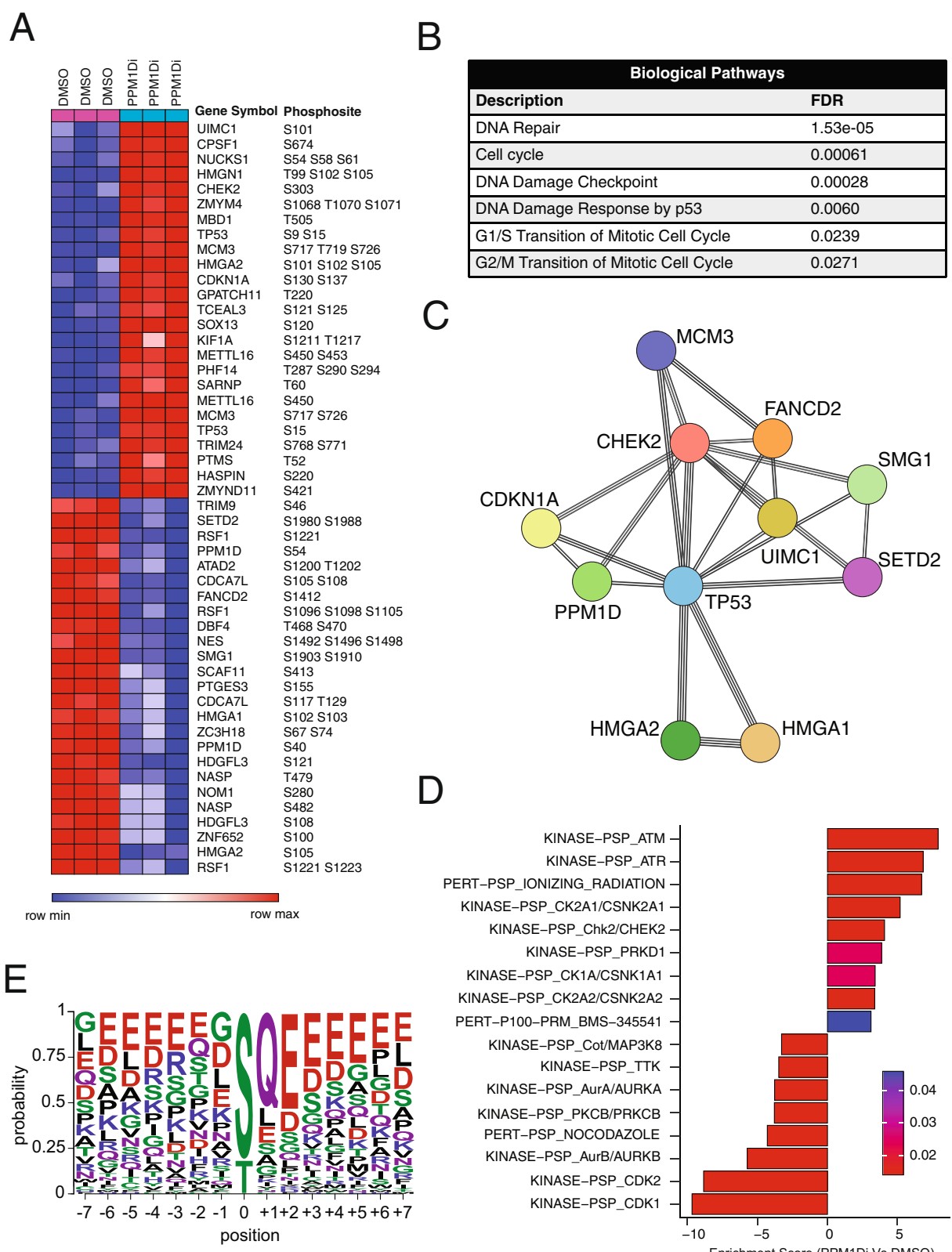

calculated pairwise Pearson correlations between *PPM1D* dependency and dependency on all other genes within the DepMap dataset (Fig. 6C) and found cell cycle checkpoint and *TP53*-associated DNA repair genes to be most strongly associated with *PPM1D* dependency. Negative regulators of *TP53* signaling and *PPM1D* substrates *MDM2* and *MDM4* were the top dependencies that correlated with *PPM1D* dependency (Fig. 6C and

Supplementary Fig. 9A). *PPM1D* expression also correlated with *MDM2* dependency across all cell lines (Supplementary Fig. 9B). However, cell cycle checkpoints *CHEK1*, *CHEK2*, *ATM*, and *ATR* were anti-correlated with *PPM1D* dependency. In contrast, dependency on *Atm* and *Chek2* were associated with *PPM1D*tr expression in our mNSC models. Unlike cancer cell lines, mNSCs have intact *TP53* and cell cycle checkpoints. We speculate that

**Fig. 5 Phosphoproteomic analysis of PPM1D substrates in *PPM1D*-mutant cell lines. A** Patient-derived *PPM1D*-mutant DMG cell line BT869 was treated with 10 μM of the PPM1D inhibitor GSK2830371 (or DMSO vehicle control) for 5 hours to suppress the PPM1D phosphatase following which phosphoproteomic profiling was performed. **A** Heatmap of top 50 differentially altered phosphosites (LFC > 1 and FDR < 0.01) between GSK2830371 treated (PPM1Di) and vehicle-treated (DMSO) samples is shown. Experiments were performed in triplicates. **B** Pathway enrichment analysis using STRING reveals significantly associated biological pathways among phosphosites shown in (**A**) and, (**C**) Associated core interactome following PPM1D suppression. **D** Significantly enriched or downregulated pathways (FDR < 0.05) revealed by PTM-SEA analysis of the phosphosites between the two conditions. Positive and negative enrichment scores correspond to biological pathways upregulated and downregulated respectively in GSK2830371 treated (PPM1Di) samples compared to vehicle-treated (DMSO) samples. **E** Motif analysis showing conserved amino acids flanking 97 confidently localized phosphorylation sites that are upregulated upon PPM1D inhibition in BT869 cells (LFC > 1, FDR < 0.01) and their associated probability of occurrence at that position.

other interventions that induce DNA damage may be required to unmask these dependencies in cancer cells that harbor other alterations beyond *PPM1D* mutations.

We validated *MDM2* to be a genetic dependency in *PPM1D*-mutant BT869 DMG cells. Suppression of *MDM2* expression by two different short hairpin RNA (shRNA) led to significantly reduced rates of proliferation (mean fold changes from hours 30 to hours 192 were $18.2 \pm 0.4$ and $13.4 \pm 0.3$ respectively) compared to those infected with non-targeting control shRNA against GFP (mean fold change $29 \pm 0.6$) ($P < 0.0001$ for both *MDM2*-targeting shRNA compared to the GFP shRNA) (Fig. 6E, F).

Consistent with the genetic data, *PPM1D*-dependent cancer cells also tended to be sensitive to pharmacologic inhibition of MDM2 using the MDM2 inhibitor Nutlin-3 (Pearson 0.38, $P < 10^{-21}$) (Fig. 6D).

This finding extended to *PPM1D*-mutant DMG cells. We evaluated the efficacy of the older generation MDM2 inhibitor Nutlin-3 and two newer generation MDM2 inhibitors, AMG232 and RG7388[28,29], across a panel of four patient-derived *PPM1D*-mutant and *PPM1D*-WT DMG cell lines. *PPM1D*-mutant lines (BT869 and SF7761) were more sensitive to all three drugs compared to the *PPM1D*-WT cells (SU-DIPG-XIII and SU-DIPG-XVII) ($P < 0.0005$ and $P < 0.05$ for AMG232 and RG7388 respectively) (Fig. 6G–I and Supplementary Data 11). We obtained a similar result in an analysis of AMG232 and RG7388 treatment across another independent cohort of five patient-derived DMG lines, including three *PPM1D*-mutant lines (HSJD-DIPG-007, HSJD-DIPG-008, and HSJD-DIPG-014A) and two *PPM1D*-WT TP53 mutant cell lines (QCTB-R059 and ICR-B184) ($P < 0.05$ and $P = 0.05$ for AMG232 and RG7388 respectively) (Supplementary Fig. 9C–D and Supplementary Data 11). We also evaluated the sensitivity of our endogenous *Ppm1d* truncated mouse NSC lines. Within these isogenic cell lines, *Ppm1d* truncated cells tended to be more sensitive to AMG232 treatment compared to WT (*sgLacZ*) control cells (Supplementary Fig. 9E).

Taken together, these results support the hypothesis that MDM2 inhibition is an actionable therapeutic vulnerability in *PPM1D*-mutant DMGs.

## Discussion

Our integrative analyses with an array of models and genomic, proteomic, and functional assays (Fig. 7), including the development of mouse models of *PPM1D*-mutant DMGs, nominate *PPM1D* as an oncogenic driver in de novo DMG. We confirm previous reports suggesting truncating mutations in exon 6 of *PPM1D* are clonal driver events in DMG[30,31] and show that they are sufficient to enhance de novo glioma formation. Moreover, we find expression of *PPM1D*tr to be necessary for proliferation of *PPM1D*-mutant DMG cells, and, applying orthogonal approaches, we find that these effects are largely mediated through *PPM1D*tr's role in regulating *TP53*, DDR and cell cycle pathways.

Using in utero electroporation to examine the role of mutant *PPM1D* in vivo, our data provide direct evidence that *PPM1D* truncations actively participate in de novo DMG development. Previous studies have leveraged flank implants of immortalized human astrocytes with endogenous *PPM1D* truncations, or orthotopic implants of murine glioma cells exogenously expressing *PPM1D*tr[13,14] but have not assessed the direct contribution of mutant *PPM1D* to the transformation of neural stem/progenitor cells into DMG. We demonstrate that when paired with expression of *Pdgfra* and histone K27M mutations, either truncation of endogenous *Ppm1d* or exogenous expression of *PPM1D*tr is sufficient to enhance gliomagenesis to a similar degree as p53 loss of function, and that the PPM1D phosphatase is necessary for this phenotype. While expression of *Pdgfra*, histone K27M or *Ppm1d* mutations alone do not result in high-grade glioma formation, the combination of *Pdgfra* and histone K27M mutations together results in a partially penetrant phenotype. This is likely due to the ability of histone K27M mutations to decrease/suppress *Cdkn2a* (p16) expression[16,32]. The presence of *PPM1D* or *TP53* mutations in histone K27M mutant DMGs suggests additional alterations in the cell cycle and DDR are necessary for DMG progression, and likely cooperate with decreased p16 expression.

We have generated mouse models of genetically engineered, *PPM1D*-driven DMG, including one that faithfully recapitulates endogenous *PPM1D* truncation as observed in human gliomas. One of the major limitations in developing new therapeutic approaches that target candidate genetic drivers is the availability of disease-relevant model systems. IUE DMG mouse models provide an in vivo platform to assess and optimize therapeutic approaches, including challenges associated with drug penetration across the blood-brain barrier[33,34] and resistance to radiotherapy, which is the only therapy that provides temporary relief for DMG patients[35]. With the development of *Ppm1d* mutant DMG mouse models, which represent to our knowledge the only DMG/DIPG mouse models that do not include *Trp53* or *Cdkn2a* loss of function[32,36], a direct assessment of p53 function in therapeutic response can be performed. Our data in *PPM1D*-mutant patient-derived DMG cultures nominates therapeutic targets (i.e. MDM2) that reactivate *TP53*. In addition, interactions of glioma cells with the tumor microenvironment can be fully evaluated since IUE mouse models develop as de novo tumors in immune-competent mice.

In this study, multiple assays indicated that *PPM1D*tr cooperatively drives DMG primarily by opposing the key functions of p53. First, genetic analyses found that *PPM1D*tr mutations are mutually exclusive with *TP53* mutations in DMG suggesting that these mutations have overlapping oncogenic functions. Second, in vivo and in vitro tumor initiation studies found *PPM1D*tr to be sufficient to replace dominant-negative *TP53* to enhance glioma formation in vivo and abrogate *TP53*-mediated G1/M cell cycle arrest and apoptosis in vitro. Third, genetic and phosphoproteomic assays of DMG model systems identified *TP53*, cell cycle and cell cycle checkpoints, and DDR pathways—all of which are

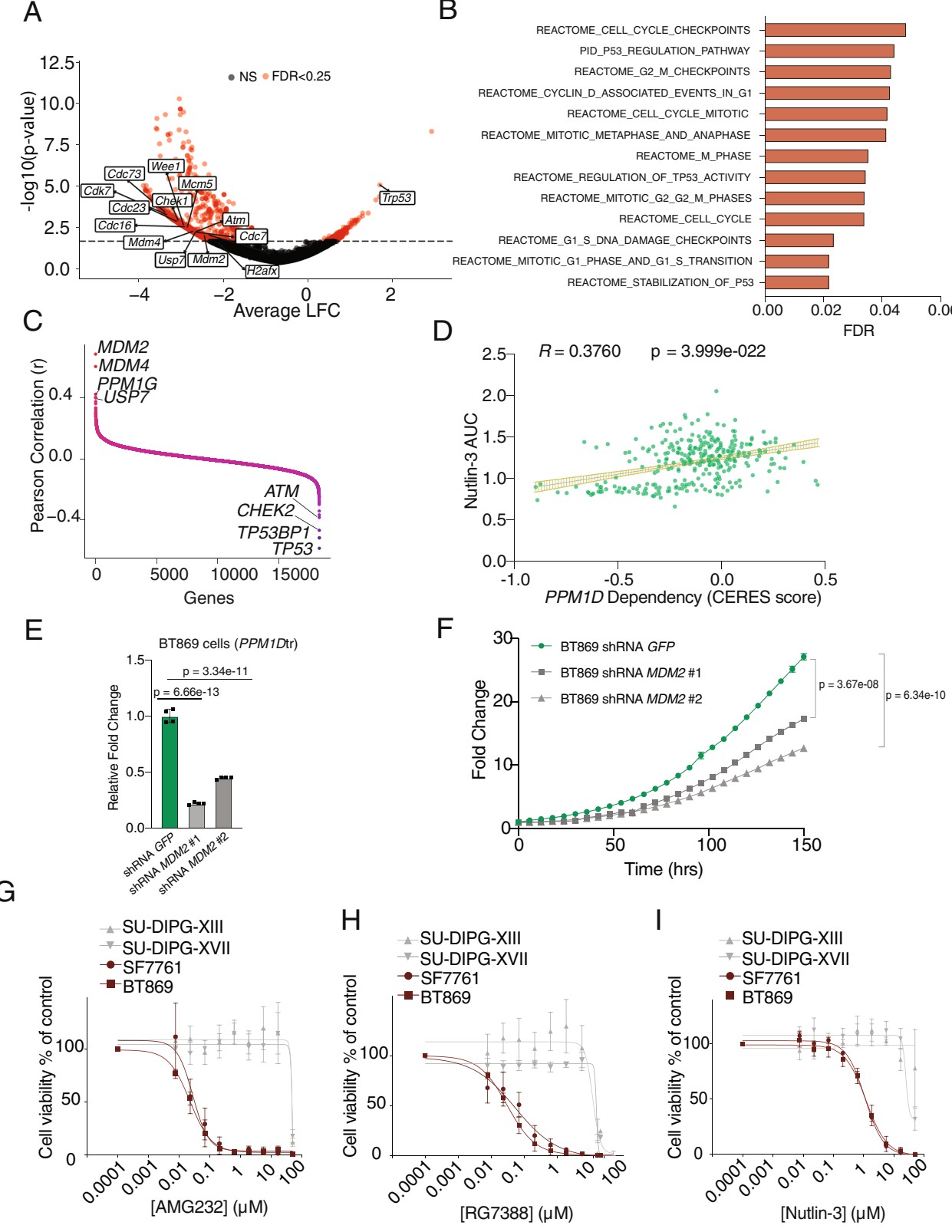

also functions of p53—as the primary pathways affected by perturbations of *PPM1D*tr. Similarly, prior studies found genetic exclusivity between *PPM1D* and *TP53* alterations[5,6] and identified *PPM1D*tr-associated chemotherapy/radiation resistance and clonal expansion in glial and hematopoietic compartments[10,11]. However, several clues point to possible additional functions of *PPM1D* truncation in glioma development. For instance,

*PPM1D*tr but not *TP53* mutation co-occurs with *PIK3CA* mutations in DMG, suggesting a possible interaction with the MAPK and mTOR pathways and its potential role in tumorigenesis. Future studies will seek to fully dissect these possible mechanistic interactions.

The current results nominate *PPM1D* as a therapeutic target for DMGs, both in solo *PPM1D*-targeting therapy and by

**Fig. 6 MDM2 inhibition is a dependency in *PPM1D*-mutant DMGs. A** Volcano plot of genetic dependencies associated with mNSC overexpressing *PPM1D*tr as revealed by genome-wide loss of function CRISPR-cas9 screen. Average LFC of normalized reads for each gene across three replicate experiments, and associated p-values are shown. A negative LFC represents depletion of guides across the assay. Genes that reach the FDR cutoff of 0.25 are labeled in red. **B** Gene-sets significantly enriched (FDR < 0.05) within the dependencies associated with *PPM1D*tr-expressing mNSCs following removal of pan-essential genes. **C** Genes ranked by Pearson correlation of *PPM1D* dependency (CERES score) against dependency of all other genes (CERES score) across 738 cancer cell lines. **D** Correlation between *PPM1D* dependency (CERES score) and response to Nutlin-3 treatment (AUC) across 309 cancer cell lines. Pearson correlation coefficient and associated two-tailed p-value of the coefficient are shown. Error bars represent 95% confidence interval. **E** RT-qPCR quantification of *MDM2* expression in BT869 cell line infected with shRNA targeting GFP or MDM2 (two independent shRNA). Results show mean ± SEM for four replicates and are representative of three independent experiments. $P = 6.66e{-}13$ and $3.34e{-}11$ for *GFP* vs shRNA *MDM2*#1 and *GFP* vs shRNA *MDM2* #2 respectively using two-tailed *t*-test. **F** Growth of BT869 cells with *MDM2* knockdown from (**E**). Growth curves are mean ± SEM for at least three replicates and are representative of three independent experiments. $P = 3.67e{-}08$ and $6.34e{-}10$ for *GFP* vs shRNA *MDM2*#1 and *GFP* vs shRNA *MDM2* #2 at 156 h respectively using two-tailed *t*-test. **G**–**I** Drug response curves for a panel of two *PPM1D*-mutant and two *PPM1D*-WT DMG cell lines treated with different concentrations of *MDM2* inhibitors AMG232 (**G**), RG7388 (**H**), and Nutlin-3 (**I**) as indicated. Data presented as mean ± SEM from three independent experiments. Source data are provided as a Source Data file.

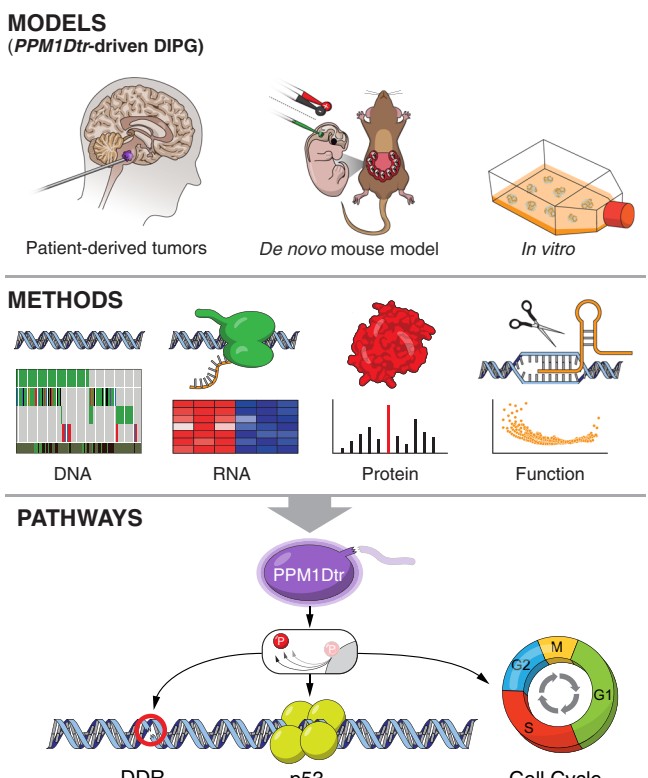

**Fig. 7 Integrative analysis of *PPM1D*-mutant DMG models.** Schematic depicting the overall method and findings of the study. Top panel: an array of models used in the study including biopsy of DMG tumors, de novo mouse models of *PPM1D* driven DMG using in utero electroporation, and in vitro models including patient-derived cell lines. Middle panel: integrative approach employing whole-genome sequencing of patient tumors, gene expression analysis, proteomics, and phosphoproteomics screens, and CRISPR screens used to study the function and mechanism of *PPM1D* mutation in DMG. Bottom panel: convergence of the oncogenic function of PPM1Dtr on p53, cell cycle and DDR pathways using the integrative approach in the middle panel.

enhancing DDR activators such as γ-H2AX and p-p53 (Ser15) when combined with DNA-damaging agents including radiation and chemotherapy. These findings are consistent with previous reports that *PPM1D* is a dependency when suppressed by shRNA or tool PPM1D inhibitors[12,14]. While there are currently no clinically relevant PPM1D inhibitors, tool compounds that inhibit its phosphatase domain have been developed, providing optimism for future efforts to generate compounds with favorable

pharmacokinetic and brain penetrant properties[37,38]. Notably, *TP53*-mutant/*PPM1D*-WT DMGs would not be predicted to benefit from PPM1D inhibition since PPM1D acts upstream of p53. These findings imply that *PPM1D* and *TP53* mutation status should be considered as biomarkers for response in investigations of *PPM1D*-directed therapies in DMG.

Our data also identify nodes in the *PPM1D* genetic network as therapeutically actionable dependencies in *PPM1D*tr DMGs. While our study implies that *PPM1D*tr and *TP53* loss of function have overlapping oncogenic functions, tumors with either alteration are likely to harbor different genetic dependencies. For instance, the protein-ubiquitin ligase MDM2 which targets p53 for degradation is well-known to be a dependency in *TP53* wild-type cancers, but inhibition of MDM2 is ineffective in cancers that already harbor loss of p53 function[28,29,39]. Importantly, emerging data suggest that p53 wild-type cancers that are driven by *PPM1D* gain of function are still susceptible to MDM2 inhibition[40]. Consistent with this observation, our current studies identify *MDM2* dependency as a potentially actionable feature of PPM1D-truncated DMGs, leading to genotyping for *PPM1D* as an inclusion criterion for a clinical trial of an MDM2 inhibitor in pediatric cancers (NCT03654716). However, multiple additional therapeutic avenues remain largely unexplored. For example, future studies should investigate additional nodes of the *PPM1D* molecular genetic network that could be therapeutically targeted both individually and in combination in *PPM1D*tr DMG.

## Methods

**Ethics statement.** The WGS and RNA-seq data of pHGGs were obtained from Dubois et al. (unpublished) study. New data were generated from tumors collected with informed consent under the Dana-Farber Cancer Institute Tissue Banking Protocol (DFCI 10417) or from tumors collected for research studies on the DIPG BATs clinical trial (NCT01182350). Ethics approval was granted by the relevant human IRB of Dana-Farber Cancer Institute (DFCI) and collaborating institutions. All patients provided informed consent prior to collection of samples or were analyzed as de-identified samples with specific IRB waiver of informed consent.

**Cell line authentication and mycoplasma testing.** SU-DIPG-IV, SU-DIPG-XIII, SU-DIPG-XVII were obtained from Dr. Michelle Monje at Stanford University, and BT869 cell lines were obtained from the DFCI Center for Patient-Derived Models (CPDM). HCT116 and HEK293T were obtained from Broad Institute's Cancer Cell Line Factory (CCLF). SF7761 cell line was purchased from Sigma Aldrich (SCC126). HSJD-DIPG-007, HSJD-DIPG-008, HSJD-DIPG-014A, ICR-B184, and QCTB-R059 cells were obtained from Drs. Chris Jones and Angel Carcaboso. mNSCs used for overexpression studies were obtained from Dr. Charles Stiles at DFCI and were generated as previously described[41,42], Cells were routinely fingerprinted or sequenced for identity validation and tested (at least every 3 months) for mycoplasma infection using the MycoAlert Mycoplasma Detection Kit (Lonza, LT07-318), according to the manufacturer's instructions.

*Whole-genome sequencing and processing.* We analyzed recently generated WGS from pediatric high-grade gliomas (pHGG), including DMG, and their matched normal[43]. Additional WGS data analyzed were downloaded from previously

published studies[3,4,44,45]. This dataset consisted of 131 pre-treatment pHGGs (76 DMGs and 55 non-midline pHGGs) and 39 post-treatment pHGGs (34 DMGs and 5 non-midline pHGGs). Among patients with available demographic information, there were equal proportions of males and females and the age ranged from 0.1 years to 31 years with mean age of 9.49 years. For whole-genome sequencing, genomic DNA extracted from the samples were fragmented and libraries were prepared. Alignment of reads to the human reference genome GRCh37 (hg19) was performed using the Burrows-Wheeler Aligner (BWA)[46]; duplicates were marked using SAMtools and Picard. Recalibration of base quality score to control for biases was performed using the Genome Analysis Toolkit (GATK)[47].

**Mutation and copy number analysis**. Single nucleotide variants (SNVs) and short indels were called using GATK4-MuTect2 pipeline and visual inspections were performed in the Integrative Genomics Viewer (IGV)[48]. MutSig2CV was applied to detect significantly recurrent mutations[49]. Copy number alterations were evaluated using the GATK4 somatic CNV pipeline. Copy number calls were purity and ploidy normalized using ABSOLUTE[50]. GISTIC 2.0 was used to identify recurrent copy-number alterations[51,52]. All analyses were performed within Broad Institute's Fireclould platform. Maftools R package was used to generate mutation plots used in the figure[53].

**TCGA analysis**. Analysis of *PPM1D* and its associated alterations across several cancer types in the TCGA dataset was performed using the cBioPortal[54,55], TumorPortal[49], and Tumorscape[56] software. The data from these portals were accessed on August 2019 for adult tumors and January 2021 for pediatric tumors.

**Vector constructions**. *H3F3A* K27M, *PPM1D* full-length (*PPM1D* FL), *PPM1D* truncated (*PPM1D*tr, AA 1–426), *PPM1D* truncated phosphatase-dead (*PPM1D*tr-D314A) constructs were synthesized as Gateway-compatible entry clones with GenScript. These gene sequences in the Gateway entry vectors were then cloned into the pLX313 (*H3F3A* K27M) and pLX311 (*PPM1D* FL, *PPM1D*tr, and *PPM1D*tr-D314A) destination vectors using Gateway LR Clonase II reaction (Life Technologies, 11791-020). pLX311-*GFP* and pLX313-*GFP* vectors were purchased from the Broad Genetics Perturbation Platform (GPP). For the *PPM1D* constructs, V5 was N-terminally tagged in the entry clone. For the remaining vectors, V5 in the destination vector was cloned to be in frame with the construct.

**In utero electroporation DMG mouse models**. All mouse work was done according to institutional and IACUC review boards (University of Cincinnati). Time pregnant (e13–14) CD1-ICR mice (Charles Rivers) were used for all experiments. Mice were socially housed in PIV cages on a 14/10-h dark/light cycle. Temperatures were maintained between 68–75 °F, with 45–65% relative humidity. In utero electroporation was performed as previously described[16]. Briefly, ~1 μL of concentrated DNA mixtures (1 μg/μL per plasmid) containing 0.05% Fast Green (Sigma) was injected into the fourth ventricle of embryos using a pulled glass capillary pipette. Injected embryos were electroporated by applying 5 square pulses (45 V, 50 ms pulses with 950 ms intervals; BTX ECM830) with the positive electrode of a 3 mm tweezer electrodes directed towards the lower rhombic lip of the fourth ventricle. Embryos were returned to the abdominal cavity, incision sutured, and the female monitored until fully recovered. Electroporated pups were identified by bioluminescent imaging (IVIS Xenogen) and monitored for neurological symptoms related to tumor burden. For histopathology, 10% formalin-fixed brains were transferred to 70% ethanol before paraffin embedding. 5 μm thick sections were prepared on a microtome (Lecia) and processed for haematoxolin-eosin (H&E) staining. Immunohistochemistry for noted antigens were performed at the CCHMC pathology core using the automated Ventana Symphony and BenchMark stainers. Antibodies include Olig2 (Millipore, Ab9610) (1:2000), Gfap (Cell Signaling, 12389) (1:1000), GFP (Thermo Fisher, A11122) (1:1000), and Ki67 (Cell Signaling, 9129) (1:2000) at indicated dilutions.

PiggyBac transposon donor plasmids encoding V5-tagged *PPM1D*tr, *PPM1D*tr-D314A, and *PPM1D* FL were generated by InFusion cloning into EcoRI digested PB-CAG-Ires-Luciferase (Takara, USA). *H3F3A* K27M, *Pdgfra*D842V, and *DNp53* PiggyBac donor plasmids were described previously[16].

sgRNA sequences were designed using CHOPCHOP[57] and cloned into the all-in-one CRISPR-Cas9 plasmid pX458 (Addgene, 48138) using previously described protocols[58]. 20nt-NGG gRNA-PAM sequences include: *Ppm1d* gRNA#1: CGTCG GTGCTTCTTCATAAG-GGG, *Ppm1d* gRNA#2: TTCGACTTAAGCCATTTCGT-CGG, and *LacZ* gRNA: TGCGAATACGCCCACGCGAT-CGG. Cutting efficiency of *Ppm1d* exon6 DNA was validated using the guide-it sgRNA screening system according to the manufactures instructions (Takara, USA). The following primers were used to PCR amplify fragments of *Ppm1d* exon 6 targeted by gRNAs: Fwd-TCCAGTAGTGATGACCTCA, Rvs- TCAACATTAGCCCTGCTGTCA CA. For TIDE analysis, *Ppm1d* exon6 sequence was amplified using the aforementioned primers using gDNA template from cells or tumor samples electroporated with *Ppm1d* gRNA#1 or *LacZ* gRNA. PCRs were submitted to the CCHMC DNA sequencing and genotyping core for sanger sequencing. Trace files were analyzed by TIDE[59] to determine the quantitative spectrum of indels around the *Ppm1d* cut site. For next-generation sequencing, the sequence surrounding the *Ppm1d* exon6 cut site was PCR amplified using the following primers: Fwd- GC

TCAAGAAGTTGAAAGAACCC, Rvs-CATCTCAGCACACACACACT. PCR products were submitted to the DFCI MBCF for NGS sequencing and the results were analyzed using the CRISPResso2 tool[60].

**Amplicon sequencing for CRISPR editing characterization**. DNA was extracted using DNA Mini Kit (Qiagen, 51306) and the target region in mouse *Ppm1d* was amplified by PCR (EMD Millipore) using the following primers: *Ppm1d*-F GCTCAAGAAGTTGAAAGAACCC; *Ppm1d*-R CATCTCAGCACACACACACT. The successful amplification of this region was validated by running the purified PCR product in 1.5% agarose gel. After validation, amplicons were submitted for NGS sequencing to the DFCI MBCF. The sequencing FastQ files were analyzed and the percent of modified reads (both in-frame and frameshift) were calculated using the CRISPResso2 software[60].

**Bulk RNA sequencing**. RNA extraction was performed using the RNAeasy kit (Qiagen, 74104). Once the samples passed initial QC analysis for quantity and quality, library prep was performed using Roche Kapa mRNA Hyper Prep and sequenced using the Illumina NovaSeq 6000 platform at DFCI MBCF. The sequencing FastQ files were aligned and counted using STAR[61]. The raw count matrix was normalized and Z-scored in IUE mouse tumors and human DMG for comparison. For *PPM1D*-mutant cell lines BT869 and SF7761, differential expression parameters were calculated using the DESeq2 pipeline[62]. Data from both cell lines were averaged for this analysis and genes with a total combined count of <10 were ignored. The normalized counts produced by DESeq2 were also used for pathway enrichment analysis using GSEA[63]. Calculation of significance of overlap of highly expressed genes between the human and mouse DMGs was calculated using the web tool available at http://nemates.org/MA/progs/overlap_stats.html.

**Culturing of mNSCs**. mNSCs were maintained in ultra-low attachment flasks in culture medium with 1:1 ratio of DMEM/F-12 (Invitrogen, 11330-032) and Neurobasal A (Invitrogen, 10888-022) consisting of 1% each of HEPES Buffer Solution 1 M, sodium pyruvate solution 100 nM, non-essential amino acids solution 10 mM, Glutamax-I Supplement and Penicillin/Streptomycin solution. The media was supplemented with B27 Minus Vitamin A (Invitrogen, 12587-010), epidermal growth factor (EGF; StemCell Tech. Inc., 78006), fibroblast growth factor (FGF; GF003, StemCell Tech. Inc., 78003) and heparin solution, 0.2% (StemCell Tech. Inc., 07980). Cells were dissociated using Accutase (StemCell Tech. Inc., 07922) and passaged every 3 or 4 days.

**Stable expression of H3F3AK27M, *PPM1D* FL, and *PPM1D*tr in mNSCs**. HEK293T cells were transfected with 10 μg of lentiviral expression vectors with packaging plasmids encoding PSPAX2 and VSVG using Lipofectamine 3000 (L3000075). Lentivirus-containing supernatant was collected 24 h after transfection, pooled, and concentrated using the Lenti X-Concentrator (Takara, 631231) according to the manufacturer's instructions. Target mNSCs were infected using a spin protocol (931 g for 120 min at 30 °C with no polybrene). Cells were selected using Blasticidin selection in pLX311 vector expressing cells (1 μg/mL) and hygromycin selection (200 μg/mL; Life Technologies, 7246) in pLX313 vector expressing cells 24 h after the infection.

**Culturing of patient-derived DMG and other cell lines**. All DMG cell lines used in the study (except for SF7761) were cultured in ultra-low attachment flasks in culture medium with 1:1 ratio of DMEM/F-12 (Invitrogen, 11330-032) and Neurobasal A (Invitrogen, 10888-022) and ten percent each of HEPES Buffer Solution 1 M (Thermo Fisher, 15630080), Sodium Pyruvate solution 100 nM (Life Technologies, 11360070), MEM non-essential amino acids solution 10 mM (Thermo Fisher, 11140050), Glutamax-I Supplement (Thermo Fisher, 35050061), and Penicillin/Streptomycin solution (Life Technologies, 15140122). The media was supplemented with B27 Minus Vitamin A (Invitrogen, 12587-010), epidermal growth factor (EGF; StemCell Tech. Inc., 78006), fibroblast growth factor (FGF; GF003, StemCell Tech. Inc., 78003) and heparin solution, 0.2% (StemCell Tech. Inc., 07980), as well as PDGF-AA (Shenandoah Biotech, 100-16) and PDGF-BB (Shenandoah Biotech, 100-18). SF7761 cell line was cultured in medium with Neurobasal A (Invitrogen, 10888-022) and N-2 Supplement (Invitrogen, 17502), further supplemented with B27 Minus Vitamin A (Invitrogen, 12587-010), epidermal growth factor (EGF; StemCell Tech. Inc., 78006), fibroblast growth factor (FGF; GF003, StemCell Tech. Inc., 78003) and heparin solution, 0.2% (StemCell Tech. Inc., 07980). Cells were dissociated using Accutase (StemCell Tech. Inc., 07922) and passaged every 3 or 4 days.

***MDM2* knockdown**. BT869 cells were infected with lentiviral plasmids expressing shRNA against human *MDM2* or GFP as a control. The following shRNA sequences were used: MDM2 #1, CTCTCGACTCAGAAGATTATA; MDM2 #2, ATCAACTTCTAGTAGCATTAT; GFP, GCAAGCTGACCCTGAAGTTCAT. One day after infection, cells were started selection in 1 μg/mL puromycin. Two days post selection, cells were plated into 96-well plates and the proliferation rates

were analyzed over time using spheroid assay in the Incucyte live cell imaging system. Knockdown of *MDM2* was validated using qRT-PCR.

**Quantitative reverse transcription PCR**. RNA was extracted using the RNeasy Mini kit (Qiagen) and subjected to on-column DNase treatment. cDNA was synthesized with the Superscript II Reverse Transcriptase kit (Life Technologies) with no–reverse-transcriptase samples serving as negative controls. Gene expression was quantified by Power Sybr Green Master Mix (Applied Biosystems). MDM2 expression values were normalized to Beta-Actin and the fold change was calculated by the DDCt method by comparing to the parental cells and scaling to the average of the vector control condition. Primers used in our studies are as follows: h-MDM2-F: GCAGTGAATCTACAGGGACGC; h-MDM2-R: ATCCT-GATCCAACCAATCACC; Beta-Actin-F: ACGTGGACATCCGCAAAGAC; Beta-Actin-R: CAA GAAAGGGTGTAACGCAACTA

**PPM1D knockout experiments**. SU-DIPG-IV, SU-DIPG-XIII, HCT116, and BT869 cells were infected with plasmids expressing cas9 and two independent guides against *PPM1D* exon 6, *LacZ*, or known an essential gene *EXOSC8*. The following gRNA sequences were used: lacZ, TGCGAATACGCCCACGCGAT; *PPM1D* #1 GCCAGTGTGGTCATCATTCG, *PPM1D* #2, GAACGAATCGAAG GACTTGG; *EXOSC8* CGGAATCTCGATGAACACAG. Cell lines were first transfected with LentiBlastCas9[64] and selected with Blasticidin (Thermo Fisher, A1113903) to generate stable pools of cells expressing Cas9. Cells were then transfected with LentiGuidePuro plasmids[64] as described above to express the indicated gRNAs. After two days, cells were introduced to 400 ng/μL puromycin selection. For growth assays, cells were seeded into 96-well plates in triplicate on day 4 after transfection. CellTiterGlo assays were performed on Day 0 and on at least 3 time points based on the growth dynamics of the cell line being assayed. For serial monitoring of *PPM1D* gene disruption by NGS, transfected cells were added to T75 flasks and serially passaged. gDNA was isolated and subjected to PCR to amplify a 150–300 bp region encompassing the guide site of interest. To confirm *EXOSC8* cutting, the following primers were used: AGCTGCAGAGT GTTTCTTTCA and AGAGCAAAGTAAATGAAAAGCCCAA. To confirm *PPM1D* #1 cutting, the following primers were used: GCGGAATGGCCAAA GACTAT and CACCAAGTCCTTCGATTCGT. For *PPM1D* #2, the following primers were used: CAACTGCCAGTGTGGTCATC and AAGGGACAGTAG TAGGTCAATTTCA. NGS was performed at the MGH CCIB to assess for the percentage of copies of *PPM1D* that were uncut, those that had *PPM1D* damaging edits, and those that had in-frame edits.

**Protein extraction and immunoblotting**. Cells were lysed in RIPA (radio-immunoprecipitation) buffer containing protease and phosphatase inhibitors on ice and sonicated for 10 one-second pulses. Lysate was centrifuged at 13,000 × *g* for 15 min and the supernatant was harvested. Supernatant was quantified with the Pierce 660 nm Protein Assay (Thermo Fisher, 22660) and mixed with 4X LDS Sample loading buffer (Invitrogen, NP0008) and heated at 70 °C for 10 min. Equal amounts of samples were loaded and separated by SDS-PAGE on 4–12% Bis-Tris gradient gels. Antibodies against the following proteins were used with indicated dilutions: Phospho-p53 (Ser15) (D4S1H) Rabbit mAb (Rodent Specific) (Cell Signaling, 12571) (1:200), p53 (1C12) Mouse mAB (Cell Signaling, 2524) (1:200), Phospho-Histone (H2A.X) (Ser139) (20E3) Rabbit mAB (1:1000), Rabbit Mono-clonal Histone H2A.X Antibody (Cell Signaling, 2595) (1:1000), p53 Antibody (DO-1) mouse monoclonal (Santa Cruz Biotechnologies, sc126) (1:200), Mono-clonal Anti-Vinculin antibody produced in mouse (Sigma, V9131) (1:1000), Anti-V5 Antibody (Thermo Fisher, R960-25) (1:1000), Phopho-p53 (Ser15) Antibody (Cell Signaling, 9284) (1:500). After transferring, the blots were blocked in Odyssey Blocking Buffer (Li-Cor, 927-40125) for one hour at room temperature. Blots were put in primary antibody at 4 °C overnight, and secondary antibody for 2 h at room temperature. Anti-mouse IgG, HRP-linked antibody (Cell Signaling, 7076) (1:10,000) and Anti-rabbit IgG, HRP-linked antibody (Cell Signaling, 7074) (1:10,000) were used as secondary antibodies against mouse and rabbit primary antibodies respectively. The blots were washed in 1X TBST three times for 15 min total between each step. To develop the blots, they were saturated in SuperSignal West Pico PLUS Chemiluminescent Substrate (Thermo Fisher, 34578). Visualization was performed using the Fujifilm LAS-3000 Imaging System.

**Flow cytometry for apoptosis and cell cycle**. mNSCs were treated with 8 Gy of ionizing radiation and the experiments were performed 24 h post treatment. Annexin V-APC (Thermo Fisher, A35110) and Propidium iodide (Thermo Fisher, NC9699940) were used to determine the proportion of early and late apoptotic cells as previously described[65]. The proportion of cells in different phases of the cell cycle was determined by flow cytometric assessment using the APC BRDU/7-AAD Flow Kit (BD biosciences, 556454) as per the manufacturer's instructions.

**GSK2830371 and ionizing radiation treatment of DMG cell lines**. *PPM1D*-mutant DMG cell lines BT869 and SF776 were treated with vehicle control (DMSO), 10 μM of *PPM1D* inhibitor GSK2830371 (MedChemExpress, 5140), 2 Gy ionizing radiation treatment, or the combination of both. Cells were seeded at a density of 1000 cells per well in 96-well ULA plates. Each biological replicate had

six technical replicates for each cell line. The proliferation rates were analyzed over time using spheroid assay in the Incucyte live cell imaging system. An unpaired t-test assuming equal variance was conducted to compare the fold change values between different conditions at day 5 of the experiment.

**Sample processing for phosphoproteomic analysis**. mNSC overexpressing GFP, *PPM1D* full-length or *PPM1D*tr were grown in ultra-low attachment chambers (Corning) in three technical replicates per condition. Similarly, patient-derived DMG cell line BT869 was treated with either DMSO or 10 μM of *PPM1D* inhibitor GSK2830371 in triplicates for 5 h. Both mNSC and BT869 cell pellets were pre-pared for MS analysis using an optimized workflow[66] employing a tandem mass tag (TMT) isobaric labeling strategy. Briefly, all cell pellets were lysed in an 8 M urea lysis buffer, followed by reduction, alkylation, and digestion using LysC followed by trypsin overnight. Samples were desalted using reverse phase C18 SepPak cartridges (Waters Corp.) and 400 μg aliquots were prepared for isobaric labeling. The TMT 10-plex included mNSC samples (three replicates for each experimental condition) along with a pooled reference sample that consisted of all the mNSC samples. The experiment using BT869 samples was a TMT 6-plex and contained three replicates for each experimental condition. Labeling was done following a reduced TMT protocol[67]. After confirming 98% or greater label incorporation, reactions were quenched; samples for each plex were mixed together, desalted, then fractionated and concatenated into 24 fractions. Five percent of each fraction was taken for global proteome analysis by liquid chromatography-mass spectrometry (LC-MS/MS). The remaining 95% of the 24 fractions was then concatenated down to 12 fractions that were enriched for phosphopeptides by immobilized metal (Fe3+) affinity chromatography (IMAC)[66].

**Mass spectrometry and data analysis**. The proteome and phosphoproteome fractions were analyzed as described previously[66] on either a Q Exactive Plus mass spectrometer (proteome) or a Q Exactive HF-X mass spectrometer (phosphoproteome) coupled with an Easy-nLC 1200 LC system (Thermo Fisher Scientific). Data were searched on the Spectrum Mill MS Proteomics Workbench (Broad Institute) using either a mouse or human database that contained 47069 and 69062 entrees downloaded from Uniprot.org on 12/28/2017 and 08/05/2020, respectively. The Spectrum Mill generated proteome level export from the proteome dataset filtered for proteins identified by two or more peptides and the phosphosite level export from the phosphoproteome dataset were used for further statistical analyses. Protein and phosphosite quantification were achieved by taking the ratio of TMT reporter ions for each sample over the TMT reporter ion for the pooled reference channel for the TMT10 mouse experiment or the median of all channels for the TMT6 human samples. TMT10 and TMT6 reporter ion intensities were corrected for isotopic impurities in the Spectrum Mill protein/peptide summary module using the afRICA correction method which implements determinant calculations according to Cramer's Rule[68] and correction factors obtained from the reagent manufacturer's certificate of analysis (https://www.thermofisher.com/order/catalog/product/90406) for lot numbers TE270709 and UC280588 respectively. After performing median-MAD normalization, a moderated two-sample t-test was applied to both the mouse and human datasets to compare GFP and *PPM1D*tr (mouse), or drug and DMSO (human) samples. Benjamini-Hochberg corrected p-value thresholds were used to assess significantly regulated proteins and phosphosites between experimental conditions. Enrichment of biological pathways among the proteins with top 50 significantly altered phosphosites (LFC < 1, FDR < 0.05) in the human data was performed using STRING[24].

**PTM signature enrichment analysis**. Enrichment analysis of phosphoproteome datasets was performed by Post Translational Modification-Signature Enrichment Analysis (PTM-SEA)[25] using results of moderated two-sample t-test. A table was generated for both mouse and human datasets that included all identified phosphosites with an assigned value that incorporated the FDR corrected p-value along with the direction of fold change. These phosphosite tables were searched against curated site-centric databases in PTMsigDB, version 1.9.0 which included 130 and 558 unique signatures for mouse and human, respectively.

*CausalPath analysis*. CausalPath[26] is an analysis tool for proteomic datasets that generates causal hypotheses by aligning the observed changes with literature knowledge. We applied CausalPath with 0.05 FDR threshold on phosphoproteomic differential abundances and generated 51 result relations. We inserted inhibition of PPM1D as a custom hypothesis and used 0.05 FDR threshold for network significance calculations.

**Sequence motif analysis**. Sequence motif logos depicting the conservation of amino acid residues around confidently localized phosphorylation sites (Spectrum Mill VML score > 1.1) were generated with the "motifStack" Bioconductor R-package. Putative PPM1D-dependent phosphorylation sites were defined by stringent criteria to mitigate the impact of potential secondary, non-PPM1D-dependent phosphorylation events (log FC > 1, adjusted p < 0.01) resulting in 97 upregulated phosphorylation sites upon PPM1D inhibition. Background frequencies of amino acids were calculated from the human proteome database (UniProt, downloaded 05 Aug 2020). The significance of the conserved occurrence

of amino acids in a sequence window of ±7 amino acids around phosphorylation sites was calculated by the Fisher exact test using the sequence windows of all detected and localized phosphorylation sites as background ($N = 23\,740$).

**Cas9 activity assay in mNSC overexpressing *PPM1D*tr**. Cas9 activity was assessed in mNSC overexpressing *PPM1D*tr in preparation for the genome-wide loss of function screen. $2 \times 10^6$ cells were seeded in one well of a 12-well plate, then spin-infected with pLX311-GFP virus. Cells were flow-sorted for GFP-positive cells and then spin-infected with the all-in-one vector of sgGFP and Cas9. Cells were selected for 48 h with puromycin (commencing 48 h post-infection). Cas9 activity was confirmed with flow analyzer, measuring percent decrease in GFP positive cell fraction as previously described[17].

**Determination of infection conditions for CRISPR pooled screens**. Optimal infection conditions were determined in order to achieve 30–50% infection efficiency, corresponding to a multiplicity of infection (MOI) of ~0.5–1, by infecting cells with different virus volumes. No-spin infections were performed in ultra-low attachment T25 flasks (Corning) with $6 \times 10^5$ cells/mL in conditioned media per flask. Each virus volume was seeded in duplicate flasks, and cells were then incubated at 37 °C. After 6 h, an additional 3 mL of '2X' media per flask was added to dilute the virus. Approximately 30 h after infection, one flask per virus volume was supplemented with 3 mL of '2X' media per flask; the other flask per virus volume was supplemented with 3 mL of '2X' media per flask and with a final concentration of 0.5 μg/mL puromycin. Cells were counted 3 days post selection to determine the infection efficiency, comparing survival with and without puromycin selection. Volumes of virus that yielded ~30–50% infection efficiency were used for screening.

**CRISPR pooled proliferation screens**. The lentiviral barcoded 'Brie' library used in the primary screen contains 79,633 sgRNAs, which includes an average of 4 guides per gene and 1000 non-targeting control guides. The 'Brie' library was cloned into the pXPR_BRD023 vector, which contains Cas9 under an EFS promotor. Genome-scale infections were performed in three replicates with the predetermined volume of virus in the same no-spin format as the viral titrations described above in ultra-low attachment 1-layer chambers (Corning). Infections were performed with enough cells per replicate in order to achieve a representation of at least 500 cells per sgRNA per replicate following puromycin selection (~$4 \times 10^7$ surviving cells per replicate). Approximately 6 h after infection, '2X' media was added to each chamber, and 30 h after infection, cells were selected with puromycin for three days to remove uninfected cells. After the selection was complete, ~$4 \times 10^7$ cells per replicate were passaged to maintain representation of the library. Cells were passaged every 3–4 days and harvested ~25 days after infection. Genomic DNA (gDNA) was isolated using Maxi kits according to the manufacturer's protocol (Machery-Nagel). PCR and sequencing were performed as previously described[69,70]. Samples were sequenced on a HiSeq2000 (Illumina). For analysis, the read counts were normalized to reads per million and then log$_2$ transformed. The log$_2$ fold-change of each sgRNA was determined relative to the plasmid DNA of the library for each biological replicate. The log$_2$ fold-change, $p$-value, and FDR adjusted $p$-value across the three biological replicates were calculated using the Broad Institute's Hypergeometric analysis. For the pathway enrichment analysis among the significantly enriched or depleted genes, GSEA pre-ranked[63,71] was used with all the gene sets in the c2.cp curated list in the MSigDB collections.

**MDM2 inhibition experiments**. MDM2i assays in the patient-derived cell lines (BT869, SF7761, SU-DIPG-XIII, and SU-DIPG-XVII) were performed by seeding 1000 cells per well in a 96-well white bottomed plate (Corning 3917) with DMSO controls. The MDM2i assays were performed using Nutlin-3 (Catalog No. HY-50696), Idasanutlin (RG7388 Catalog No. HY-15676), AMG232 (Catalog No. HY-12296). The drugs were plated in 1:3 serial dilutions, with 10 concentrations ranging from 0 μM to 50 μM with three biological replicates. Each biological replicate had three technical replicates for each cell line. Total number of viable cells were determined by trypan blue assays. After three days of incubation (72 h), luminescence measurements of ATP content using CellTiterGlo (Promega) were performed as a marker of cell viability. The measurements were normalized to the DMSO control with 0 μM drug.

Patient-derived cultures HSJD-DIPG-007 (*H3F3A* K27M, *PPM1D* P428fs), HSJD-DIPG-008 (*H3F3A* K27M, *PPM1D* 1286delG), HSJD-DIPG-014A (*H3F3A* K27M, *PPM1D* 1451dupT), ICR-B184 (*HIST1H3C* K27M, *PPM1D* wild-type), QCTB-R059 (*H3F3A* K27M, *PPM1D* wild-type) were grown as neurospheres in stem cell media consisting of Dulbecco's Modified Eagles Medium: Nutrient Mixture F12 (DMEM/F12), Neurobasal-A Medium, HEPES Buffer Solution 1 M, sodium pyruvate solution 100 mM, non-essential amino acids solution 10 mM, Glutamax-I Supplement and Antibiotic-Antimycotic solution (all Thermo Fisher, Loughborough, UK). The media was supplemented with B-27 Supplement Minus Vitamin A, (Thermo Fisher), 20 ng/mL Human-EGF, 20 ng/mL Human-FGF-basic-154, 20 ng/mL Human-PDGF-AA, 20 ng/mL Human-PDGF-BB (all Shenandoah Biotech) and 2 μg/mL Heparin Solution (0.2%, Stem Cell Technologies). Cell authenticity was verified using short tandem repeat (STR)

DNA fingerprinting. Cells were plated at a density of 1000–5000 cells/well on 96-well plates in a minimum of triplicates. After three days of incubation, the compound was added to each well in concentrations from 0.04–20000 nM and incubated at 37 °C, in 5% CO$_2$, 95% humidity for eight days (192 h).

For both sets of cell lines, cell viability was assessed by the 3D CellTiterGlo luminescent cell viability assay (Promega, G7573) and IC$_{50}$ values were calculated using GraphPad Prism version 8 as the concentration of compound required to reduce cell viability by 50%. Non-linear dose-response analysis with a four-parameter analysis (Inhibitor vs. Response–Variable Slope) was conducted using GraphPad Prism version 8 to fit a curve to the data. These IC$_{50}$ values for each cell line were then placed into one of two groups, wild-type or mutant, depending on *PPM1D* status. A parametric two-tailed t-test was run in Graphpad Prism version 8 to establish whether *PPM1D*-WT cell lines and *PPM1D*-mutant cell lines had significantly different IC$_{50}$ values.

**PPM1D and MDM2 cancer cell line dependency analysis**. The probability of dependency (PD) for all cell lines in the 18Q4 public release of the CRISPR-Avana pooled dependency screening database and associated mutation and copy number datasets were downloaded[17,18]. PD for *PPM1D* was displayed for cell lines based on *PPM1D* mutational status, *PPM1D* copy-number, and *TP53* mutational status. For pairwise gene-dependency, and *MDM2* and *PPM1D* dependency correlation analyses, CERES scores from the 20Q1 public release of the DepMap dataset were used[17]. For *PPM1D* expression level, 20Q2 release of the expression dataset was used[18]. AUC for Nutlin-3 was obtained from PRISM Repurposing Secondary Screen 20Q2 release dataset[72]. Figures were made in R and GraphPad Prism.

**Quantification and statistical analysis**. Graphs showing quantification of Ki67 IHC staining were calculated from at least six individual tumors per condition, with 10 randomly selected fields imaged per sample. Cell viability curves for the different inhibitors were calculated from at least three independent experiments with triplicates for each condition. Statistical significance was determined by Fisher's exact test, Student's $t$ tests, log rank Mantel–Cox test, Kruskal–Wallis test, or Mann-Whitney test as indicated in the figure legends using Microsoft Excel, Graphpad Prism or R.

**Reporting summary**. Further information on research design is available in the Nature Research Reporting Summary linked to this article.

## Data availability
WGS along with corresponding RNA-sequencing data used in this study have been deposited to dbGaP with restricted access under Accession Number phs002380.v1.p1. RNA-seq data from our mouse models and treatment of human DMG cells BT869 and SF7761 have been deposited to GEO under accession number GSE179813. The proteomic data has been deposited to MassIVE under dataset identifier MSV000088404. https://massive.ucsd.edu/ProteoSAFe/dataset.jsp?task=ce04ad6c48224d128213c42ebbe4c1b4. Source data are provided with this paper.

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

## Acknowledgements

We thank and acknowledge the patients and their families who made this research possible. We would like to thank Dr. Charles Stiles for sharing the mouse neural stem cells (mNSC) used for in vitro overexpression studies, the DFCI Center for Patient-Derived Models (CPDM) for sharing the patient-derived BT869 cell line, Dr. Michelle Monje for sharing the patient-derived cell lines SU-DIPG-IV, SU-DIPG-XIII, and SU-DIPG-XVII, Mr. Eric Smith for generating the schematic included in Fig. 7, and Dr. Cory Johannessen and members of the Phoenix, Beroukhim, and Bandopadhayay Laboratories for helpful discussion. The authors would like to thank and acknowledge the following funding sources: The Giving for Gabi Fund (P.B.), Abbie's Army (D.C., C.J.), Brain Research UK (D.C., C.J.), The V Foundation for Cancer Research (P.B.), Michael Mosier Defeat DIPG Foundation (Z.J.R., P.B.), The ChadTough Foundation (Z.J.R., P.B.), The SoSo Strong Foundation (Z.J.R.), The St. Baldrick's Foundation (Z.J.R., R.B., P.B.), Prayers from Maria Children's Cancer Foundation (P.B., T.N.P.), Pediatric Brain Tumor Foundation (P.B., Z.J.R., T.N.P., R.B.), Friends of DFCI (P.K., Z.J.R., P.B.), We Love You Connie Foundation (R.B., K.L.L., P.B.), National Cancer Institute (NCI) Clinical Proteomic Tumor Analysis Consortium grants NIH/NCI U24-CA210986 and NIH/NCI U01 CA214125 (S.A.C.), Department of Defense Grant #CA171185 (T.N.P.), The Matthew Larson Foundation (T.N.P.), NIH NCI R00-CA201592 (P.B.), NIH R37 5R37CA255245-02 (P.B.), NIH R01 CA188228 (R.B., K.L.L.), R01 CA215489 (R.B., K.L.L.), R01 CA219943 (R.B., K.L.L.), The Dana-Farber/Novartis Drug Discovery Program (R.B.), The Gray Matters Brain Cancer Foundation (R.B.), Ian's Friends Foundation (R.B.), The Bridge Project of MIT and Dana- Farber/Harvard Cancer Center (R.B.), The Fund for Innovation in Cancer Informatics (R.B.), and The Sontag Foundation (R.B.).

## Author contributions

P.K., Z.J.R., R.B., T.N.P., P.B. conceived the project, designed the experiments, and co-wrote the manuscript. P.K., Z.J.R., S.L., G.B., G.G., F.D., J.S., S.Z., N.F.G., T.Z., O.S., Y.G., S.J., P.G.M., A.L.C., E.G., K.Q., E.M., J.L., L.L., V.R., J.D., D.W., K.Z., R.K., M.E.G., B.L.E., O.B., M.W.K., and K.L.L. generated reagents, performed the experiments and computational analysis, and analyzed the data. D.M.C., A.M.C., and C.J. performed independent validation of MDM2 inhibition in patient-derived DMG cell lines. K.P., R.H., H.B., K.S., C.F., K.L.L., and T.N.P. performed in vivo mouse experiments and histopathological analysis. C.E.S., S.B., R.S., and J.V. assisted in the acquisition of patient samples and relevant clinical data. R.M., K.B., K.K., H.K., and S.A.C. generated and analyzed mass spectrometry-based proteomics and phosphoproteomic data. A.G., T.A., Z.K., F.P., N.S.P., and D.E.R. generated and analyzed CRISPR screen data in mouse neural stem cells. T.N.P. and P.B. contributed equally and supervised the overall study.

## Competing interests

R.B. and P.B. receive grant funding from the Novartis Institute of Biomedical Research for an unrelated project. P.B. has received grant funding from Deerfield and reports a consulting role for QED Therapeutics, R.B. reports consulting or advisory role for Novartis, Merck (I), Gilead Sciences (I), ViiV Healthcare (I); research funding from Novartis; patents, royalties, other intellectual property—Prognostic Marker for Endometrial Carcinoma (US patent application 13/911456, filed June 6, 2013), SF3B1 Suppression as a Therapy for Tumors Harboring SF3B1 Copy Loss (international application No. WO/2017/177191, PCT/US2017/026693, filed July 4, 2017), Compositions and Methods for Screening Pediatric Gliomas and Methods of Treatment Thereof (international application No. WO/2017/132574, PCT/US2017/015448, filed 1/27/2017). M.W.K. is now an employee of Bristol-Myers Squibb. F.P. is now an employee of Merck Research Laboratories. S.A.C. is a member of the scientific advisory boards of Kymera, PTM BioLabs and Seer and a scientific advisor to Pfizer and Biogen. D.E.R. receives research funding from Abbvie, Jannsen, Merck, and Vir through the Functional Genomics Consortium and serves on the board of directors of Addgene. B.L.E. has received research funding from Celgene and Deerfield. He has also received consulting fees from GRAIL, and he serves on the scientific advisory boards for and holds equity in Skyhawk Therapeutics and Exo Therapeutics. (I) denotes a competing interest involving a first degree relative of the author. All the remaining authors declare no competing interests.
