## [Peer Review File · Nature Communications]

PPM1D mutations are oncogenic drivers of de novo Diffuse Midline Glioma formationEditorial Note: Parts of this Peer Review File have been redacted as indicated to maintain the confidentiality of unpublished data.

Reviewers' comments:

Reviewer #1 (Remarks to the Author): Expert in glioma progression and mouse models

This manuscript expands on the role of phosphatase PPM1D as a driver of malignancy in pediatric DMG tumors. WGS data generated from 170 pediatric gliomas, including 76 untreated and 34 treated DMGs, 11% DMG presented gain of function nonsense/truncating mutations in exons 5 or 6, which were mutually exclusive with TP53 mutations. VAF and CCF analysis revealed these mutations tend to be clonal. In large TCGA dataset, PPM1D amplification was observed in 2% of cancers, which did not include gliomas. Using a new method for "Breakpoint Identification of Significant Cancer Undiscovered Targets (Shih et al., unpublished)", the authors conclude that PPM1D is within a region comprising 43 genes observed in the pan-cancer dataset, with partial somatic amplification that undergoes positive selection, including melanoma and glioma, implicating a role for low level amplification of PPM1D wt, in addition to PPM1D truncated variants, in DMG. Using CRISPR-Cas9 targeting exon 6 in mouse NSCs, the authors report a positive selection for truncated PPM1D in mouse NSCs between 1 and 3 months in culture. The authors had previously developed a DIPG mouse model using IUE targeting the 4th ventricle of e13.5 embryos, showing that ectopic expression of PdgfraD842V and DNP53 is tumorigenic whether in combination with H3.3 or H3.3K27M (doi:10.1093/neuonc/noz197). Using this model, the authors show that PPM1Dtr is not sufficient to induce tumors, but significantly potentiate the tumorigenicity of PdgfraD842V / H3.3K27M mutations, to the same magnitude as DNP53. The resulting tumors presented high grade glioma histopathological characteristics. PPM1Dtr-D314A phosphatase dead variant failed to increase penetrance and growth rate, while the full length PPM1D had a more modest effect than PPM1Dtr. Treatment with GSK2830371 Wip1/PPM1D inhibitor decreased viability of PPM1Dtr DMG cell lines BT869 and SF7761, and potentiated RT, accompanied by increased phosphorylation of its targets H2AX and p53. Inhibition of PPM1D in the two PPM1Dtr DMG cell lines lead to increased p53 activity, as inferred by transcriptome and phosphoproteomics. The proteomics analysis identified inhibitor-mediated alterations in canonical PPM1D phosphatase substrates, and other potential novel direct or indirect targets, with enrichment in cell cycle and DDR regulation pathways. The phosphoproteomic analysis was repeated to compare mNSC expressing PPM1Dtr or control, with similar results. A genome-scale CRISPR/Cas9 loss-of-function screen in mNSCs overexpressing PPM1Dtr identified genes required to maintain fitness in the cells carrying the mutant gene with some enrichment in cell cycle, DDR and TP53 pathways, which were further confirmed in database containing results from human cancer cell lines CRISPR-CAS9 screens. MDM2 knockdown in PPM1Dtr BT869 DMG cells impaired cell proliferation. PPM1Dtr DMG cells were sensitive to MDM2 antagonists, while PPM1D wt cells were not. The work presented is comprehensive and contributes to advances in the understanding of the role of PPM1D in wt TP53 DMGs, in particular the results using the recently developed mouse models. However, the following points need further clarification:

- "Taken together, these results support the hypothesis that MDM2 inhibition is an actionable therapeutic vulnerability in PPM1D-mutant DMGs." (line 490/91). This may well be the case, but the results presented in the manuscript cannot dissect whether the PPM1D-mutant DMGs are sensitive to MDM2 antagonists exclusively on account of the TP53 wt status. Results from comparison of the sensitivity of TP53 wt type lines that carry wt or mutant PPM1D would provide more direct evidence that the gain of function for the phosphatase indeed further sensitizes the TP53 wt cells to the treatment. The TP53 status for the PPM1D wt DMG lines employed in these experiments is not clear.
- The inclusion of references (if published) or unpublished information regarding clinical and genomic and molecular characteristics associated with the BT869, SF776, and the wt PPM1D patient derived cell lines are needed to contextualize the data presented.
- Figure 4: results from independent Western Blot experiment should be represented and ideally bands quantified following contemporary standards for reproducibility.
- As noted in the text (lane 342), Sup. Fig. 4A show considerably higher levels of PPM1Dtr relative to PPM1D FL, yet the effect of both variants on irradiated cells shown in Fig. 4 A,B are equivalent. The authors should comment on this discrepancy.

- Fig 1G, methods (line 696) “transcripts in these populations were determined using the normalized and scaled counts”, please provide the parameters used for normalization/scaling for the scRNAseq data.

- The transcriptome analysis of GSK2830371-treated cells vs. control is important to support the premise that PPM1D suppresses p53. This experiment is limited to 1 time point (5h) and n=2 cell lines (apparently with no replicates for each cell line), which severely diminishes the statistical power and confidence on the results. Please provide an experimental basis for the single timepoint chosen and sample size calculation for the statistics.

- The assertion in lane 183: “ PPM1D expression is enriched in progenitor cells in PPM1D-mutant DMGs”, should be qualified as referring to n=1 tumor analyzed.

Other points:

- Accession dates for the TCGA data from cBioPortal, TumorPortal, and Tumorscape should be provided.

- Supplementary Tables did not format correctly in the PDF, information could not be properly accessed.

- Sup.Fig 4b: time points in the legend do not seem to be matching those in the figure.

Reviewer #2 (Remarks to the Author): Expert in glioma molecular and functional assays

This manuscript confirms the presence of mutations of the PPM1D phosphatase from the whole genome sequencing of 131 pediatric glioma. As previously reported, PPM1D mutations are truncating events targeting the C-terminal region of PPM1D and result in gain-of-function oncogenic activation as shown by the ability of mutant PPM1D to cooperate to oncogenesis in association with other genetic alterations and promote more aggressive cancer phenotypes. Similarly, the authors also confirm that this effect requires the enzymatic phosphatase activity of PPM1D and PPM1D-driven oncogenesis converges on disabling the p53 pathway. The biological phenotype of PPM1D-mediated oncogenesis is associated with alterations of cell cycle regulation, the DNA damage response, radiation sensitivity and apoptosis.

Whereas the above conclusions are broadly supported by the results presented, none of them is novel as each of the functions associated with PPM1D in the manuscript was extensively and similarly shown by previous studies. Thus, the key weakness of the paper is that it lacks novelty. Additional challenges concern the rather superficial state of a number of analyses and the lack of rigorous support for some of the least general claims. For example, the claim that the PPM1D gene harbors broad and low-level copy number gains is of uncertain significance and remains rather superficial. Based on what is shown in the manuscript, it is difficult to ascertain whether such low-level copy number gains are associated with any functional consequences. Similarly, the claim that PPM1D expression is enriched in the cancer stem cell population of pediatric glioma is premature. The authors base this claim on the scRNAseq analysis of only one glioma. In the absence of solid validation in more representative cohorts of tumors analyzed by scRNAseq, such claim remains speculative.

Reviewer #3 (Remarks to the Author): Expert in glioma genomics

In this manuscript the authors have investigated pediatric high grade gliomas, in particular diffuse midline gliomas, and the role of PPM1D truncating mutations have in these tumors. There are several interesting aspects in this manuscript, especially regarding the new mouse models which clearly demonstrate the oncogenic role of truncated PPM1D. However, there are also many aspects in the paper presented as new findings which are in my opinion not so entirely new and have been published before, but these papers are unfortunately not (well) cited. For instance, in my opinion

the whole genomic section does not add much news to the paper and can be largely reduced as it is not new at all that PPM1D truncating mutations occur in DMGs and are mutually exclusive from TP53 mutations. This was for instance already presented in the large meta-analysis paper of MacKay et al. (2017) in which more than 1000 pediatric high grade gliomas have been analyzed. Also it is totally unclear to me why the authors have compared the PPM1D alterations in their glioma data set to the TCGA dataset which only includes adult tumors and not for instance to other pediatric cancers. Furthermore, identification of PPM1D as a therapeutic target in DMGs is also not entirely new and has been published before in for instance Akamandisa et al. (2019) or Wang et al. (2020) and the link with DDR pathways has been made as well in these papers. Both papers are cited (although the citation for the Wang paper is incomplete), but still the data are kind of presented in this manuscript as if they are entirely new.

Other comments:

Abstract:

Here we analyze whole genomes sequences of 170 pediatric high-grade gliomas and find that truncating mutations in PPM1D are clonal driver events in 11% of DMGs and are enriched in primary DIPG.

Abbreviations should not be used in the abstract and moreover DIPGs are DMGs and the term DIPG is no longer used in the updated WHO classification.

We show that PPM1D mutations potentiate gliomagenesis and that PPM1D activity is required for in vivo oncogenesis.

Unclear from this sentence in the abstract that mutations lead to truncated proteins and how that is related to PPM1D activity.

Together, these findings highlight PPM1D mutations to represent a targetable driver of pediatric gliomas.

PPM1D itself is not really targetable (only tool compounds available). Why not mentioning MDM2 here?

Introduction:

Line 82: PPM1D truncating mutations that increase the stability of the phosphatase have also recently been found in clonal hematopoiesis of indeterminate potential (CHIP),....

Unclear in the introduction that the mutations in DMGs are also truncating mutations

Line 91:maintenance of DMG cells, and finally, its mechanisms of action as an oncogene.

Unclear from the introduction that PPM1D should act as an oncogene.

Line 95:that exogenous expression of mutant human PPM1D or endogenous truncation of murine Ppm1d together.....

Suggest to add 'human' to the sentence

Line 107-110: This cohort consisted of 76 DMGs, including 58 diffuse intrinsic pontine gliomas (DIPG), which is the prototypical subtype of DMG, and 55 non midline HGGs. We contrasted these with whole genome sequences of 39 post-treatment gliomas (34 DMGs including 33 DIPGs, and 5 non-midline HGG).

Please use the correct nomenclature for these tumors according to the WHO classification, which no longer uses DIPG. Were these genomic data all new or have they (partly) been published before? There are some references to previous publications regarding these data in the methods but it is totally unclear which part is new and which are already existing data. Also, will all data be deposited in a database to make them accessible for other researchers? Accession numbers are lacking.

Also, what do these data add to the previous pan high grade glioma analyses published by Mackay et al. (2017)? Already in that paper it was clear that truncating PPM1D mutations are frequently found and mutually exclusive from TP53 mutations. This paper is also not referenced in this section.

Line 115-116:we did not observe differences in the frequency of PPM1D mutations between pre- and post-treatment glioma samples (Fisher's exact test, $P = 1$),.....

Were there any paired samples (primary vs treated) included in the cohort and if yes, how were the PPM1D mutations in those? Present in both or gained or lost in the treated samples?

Line 119-120: ..PPM1D mutations were mutually exclusive with TP53 mutations (Fisher's exact test, $P < 0.0001$) (Figure 1C),.....

That PPM1D mutations are mutually exclusive from TP53 mutations is already totally clear from Figure 1B and Figure 1C does not add anything to that. When checking the mutual exclusivity with other genes in Figure 1C, it would be helpful to add these then also to Figure 1A.

Figure 1A:

Annotation below figure for location is not correct as it shows partly location and partly tumor type instead of location.

Also, in order to make the figure more clear I would suggest to sort all samples by tumor type and not by mutation.

Figure 1B:

Y-axis with # mutations does not align with the lollipops in the figure.

Line 126-129:

PPM1D mutations have also been described in myeloid neoplasms including CHIP, myelodysplastic syndrome (MDS) and acute myeloid leukemia (AML) (Hsu et al., 2018; Kahn et al., 2018), but their incidence across other solid cancers is unknown. To assess this, we extended our analysis of PPM1D mutations to include other cancers (Supplementary Figure 1A) in the TCGA dataset.

The TCGA dataset only includes adult tumors. Why did the authors not include pediatric tumors here, which would have made more sense as a comparison?

Line 147-148: In addition to truncating PPM1D mutations, PPM1D has also been reported to be amplified in several cancer types, most notably breast, mesothelioma, and liver (TCGA; Supplementary Figure 1A).

Again, what about the pediatric cancers?

Line 148-150: Analysis of our copy-number data of pHGGs revealed broad low-level gains in the PPM1D containing region on 17q, but no focal amplification of PPM1D was detected (Supplementary Figure 1D).

Instead of showing all chromosomes in this Figure, it would have been better to only show chromosome 17 as from the current figure it is impossible to judge whether there are focal amplifications of PPM1D or not.

Line 150-155: Extending our analysis to other cancer types in the TCGA dataset, we observed PPM1D amplifications in 2% of 10,967 cancers comprising of 32 cancer types. These amplifications were statistically recurrent in several cancer types including bladder urothelial carcinoma (q-value = 0.0452), breast adenocarcinoma (q-value = 3.93×10^{-5}), epithelial cancers (q-value = 2.04×10^{-31}), liver hepatocellular carcinoma (q-value = 2.68×10^{-4}), colorectal cancers (q-value = 0.0111), colon adenocarcinoma (q-value = 0.0704) and melanomas (q-value = 0.244).

Not sure what this adds to the paper, especially as no pediatric cancers were included, which are known to harbor PPM1D amplifications as well.

[REDACTED]

Line 170-172: Taken together, these data support truncating PPM1D mutations as being contributors to DMG gliomagenesis. Furthermore, the rate of copy-number gains and amplifications of the region containing PPM1D across various cancer types suggest a possible dosage effect of wild-type PPM1D.

The last part has only been analyzed in adult cancers and not in the pediatric gliomas. Also, to make any conclusions about a possible dosage effect, one need to integrate transcriptome data as well then, again focused on the pediatric gliomas.

Line 182-183: These findings suggest that PPM1D expression is enriched in progenitor cells in PPM1D-mutant DMGs.

As PPM1D expression is not enriched in progenitor cells of tumors that are wild type for PPM1D, this would speak against a possible dosage effect of wildtype PPM1D in these tumors. How is the PPM1D expression in general in DMGs vs HGGs (and vs other brain tumors?), and in PPM1D mutant vs wildtype tumors?

Line 187-188: The spectrum of PPM1D alterations in DMGs nominate both full-length and truncated PPM1D as potential oncogenes in de novo gliomagenesis.

What is the evidence for the full length as there were no amplifications found?

Lines 218-220: We therefore reasoned that PPM1D mutations may cooperate with other oncogenes to enhance gliomagenesis. We evaluated this with IUE, electroporating sgPpm1d_{exon6} with concurrent H3.3K27M and PdgfraD842V PiggyBac plasmids.

This makes sense indeed, and from other publications it is indeed known that PDGFRA plays a role in DMGs, but the rationale why including PDGFRA mutant is lacking from their own genomic data as the authors did not include the analyses for PDGFRA mutations / amplifications etc. in their own cohort. This should have done as well then.

Line 220-222: Control IUE conditions resulted in a partially penetrant phenotype, with only 50% (9/18) of mice developing neurological symptoms related to tumor, with a median survival of 85 days postnatal.

Unclear from the text what these control conditions are then. Becomes only clear from the Figure.

Line 222-223: In contrast, C-terminal truncation of Ppm1d by sgPpm1d_{exon6} was sufficient to generate fully penetrant brainstem gliomas,.....

Yes, but only in combination with H3.3K27M and PdgfraD842V. However, it remains unclear whether these are both needed. Authors should have at least tested also the combination of sgPpm1d_{exon6} plus H3.3K27M, but preferably also the combination of sgPpm1d_{exon6} plus PdgfraD842V. Unclear why this was not done.

Also surprised to see why the mouse models were not further molecularly characterized, for instance using transcriptomics, to see how well they represent the human tumors. This would also be helpful with respect to the downstream analyses of PPM1D mutants as this was all done now in cell line models only.

Line 299-307: Our finding that PPM1D is necessary for PPM1D-mutant DMGs extended to other PPM1D-expressing cancers. We examined whether PPM1D dependency was linked to genetic activation of PPM1D using pooled CRISPR-cas9 assays across 558 cancer cell lines (Ghandi et al., 2019; Meyers et al., 2017). We hypothesized that TP53-WT cell lines would be more dependent on PPM1D than TP53-mutant cell lines because of PPM1D's role in opposing p53 function. As expected, lines with wild-type TP53 were significantly more dependent on PPM1D than TP53-mutant lines (Figure 3D). Among TP53-WT lines, PPM1D copy-number gain was associated with significantly higher probability of dependency on PPM1D (Figure 3D). We conclude that PPM1D is required for proliferation of p53 wild-type cell lines, particularly those that harbor PPM1D activating genetic alterations.

Figure 3D shows that TP53 wt cell lines depend on the expression of PPM1D, even when PPM1D is not amplified and these cell lines do also not have a PPM1D mutation as these mostly occur in the lines with TP53 mutations in this cell line panel. However, how does this match then with the results of the DMG lines that shows that cells do NOT depend on PPM1D when not mutated?

Figure 3F: is this not just an additive effect?

Using cell line models the authors further investigated the downstream effects of mutant PPM1D using transcriptomics and mass spec analyses to analyze the phosphoproteome. Several interesting candidates came out of these analyses, mostly concentrated around the TP53 and/or DDR pathways. It would have been helpful if some of these targets could have been validated (by mass spec or IHC) also in the mouse tumors (by comparing the ones with and without PPM1Dtr), but also in human DMGs that are mutated or not for PPM1D to see to which extent these identified targets indeed have a role in the tumors.

Discussion

Line 494-496: Our integrative analyses with an array of models and genomic, proteomic and functional assays (Figure 7), including the development of novel mouse models of PPM1D-mutant DMGs, nominate PPM1D as a tractable therapeutic target in DMG.

It was already known that PPM1D forms a therapeutic target in DMGs. Published in several papers, which are not cited here.

Line 496: We show that truncating mutations in exon 6 of PPM1D are clonal driver events in DMG....

Is also not new, and has been shown before in the paper by Nikbakht et al. (2015). Should have been cited then as well.

Line 497: Moreover, we find expression of PPM1Dtr to be necessary for proliferation of PPM1D-mutant DMG cells...

Again, already demonstrated before.....

Response to Reviewers

Reviewer #1 (Remarks to the Author): Expert in glioma progression and mouse models

This manuscript expands on the role of phosphatase PPM1D as a driver of malignancy in pediatric DMG tumors. WGS data generated from 170 pediatric gliomas, including 76 untreated and 34 treated DMGs, 11% DMG presented gain of function nonsense/truncating mutations in exons 5 or 6, which were mutually exclusive with TP53 mutations. VAF and CCF analysis revealed these mutations tend to be clonal. In large TCGA dataset, PPM1D amplification was observed in 2% of cancers, which did not include gliomas. Using a new method for “Breakpoint Identification of Significant Cancer Undiscovered Targets (Shih et al., unpublished)”, the authors conclude that PPM1D is within a region comprising 43 genes observed in the pan-cancer dataset, with partial somatic amplification that undergoes positive selection, including melanoma and glioma, implicating a role for low level amplification of PPM1D wt, in addition to PPM1D truncated variants, in DMG. Using CRISPR-Cas9 targeting exon 6 in mouse NSCs, the authors report a positive selection for truncated PPM1D in mouse NSCs between 1 and 3 months in culture. The authors had previously developed a DIPG mouse model using IUE targeting the 4th ventricle of e13.5 embryos, showing that ectopic expression of PdgfraD842V and DNp53 is tumorigenic whether in combination with H3.3 or H3.3K27M (doi:10.1093/neuonc/noz197). Using this model, the authors show that PPM1Dtr is not sufficient to induce tumors, but significantly potentiate the tumorigenicity of PdgfraD842V / H3.3K27M mutations, to the same magnitude as DNp53. The resulting tumors presented high grade glioma histopathological characteristics. PPM1Dtr-D314A phosphatase dead variant failed to increase penetrance and growth rate, while the full length PPM1D had a more modest effect than PPM1Dtr. Treatment with GSK2830371 Wip1/PPM1D inhibitor decreased viability of PPM1Dtr DMG cell lines BT869 and SF7761, and potentiated RT, accompanied by increased phosphorylation of its targets H2AX and p53. Inhibition of PPM1D in the two PPM1Dtr DMG cell lines lead to increased p53 activity, as inferred by transcriptome and phosphoproteomics. The proteomics analysis identified inhibitor-mediated alterations in canonical PPM1D phosphatase substrates, and other potential novel direct or indirect targets, with enrichment in cell cycle and DDR regulation pathways. The phosphoproteomic analysis was repeated to compare mNSC expressing PPM1Dtr or control, with similar results. A genome-scale CRISPR/Cas9 loss-of-function screen in mNSCs overexpressing PPM1Dtr identified genes required to maintain fitness in the cells carrying the mutant gene with some enrichment in cell cycle, DDR and TP53 pathways, which were further confirmed in database containing results from human cancer cell lines CRISPR-CAS9 screens. MDM2 knockdown in PPM1Dts BT869 DMG cells impaired cell proliferation. PPM1Dts DMG cells were sensitive to MDM2 antagonists, while PPM1D wt cells were not. The work presented is comprehensive and contributes to advances in the understanding of the role of PPM1D in wt TP53 DMGs, in particular the results using the recently developed mouse models. However, the following points need further clarification:

We thank the Reviewer for highlighting the major findings of our study and for commenting on the comprehensiveness of the work and how it contributes to the understanding of *PPM1D* mutation in DMGs, particularly leveraging the novel mouse models.

- *“Taken together, these results support the hypothesis that MDM2 inhibition is an actionable therapeutic vulnerability in PPM1D-mutant DMGs.” (line 490/91). This may well be the case, but the results presented in the manuscript cannot dissect whether the PPM1D-mutant DMGs are sensitive to MDM2 antagonists exclusively on account of the TP53 wt status. Results from comparison of the sensitivity of TP53 wt type lines that carry wt or mutant PPM1D would provide more direct evidence that the gain of function for the phosphatase indeed further sensitizes the TP53 wt cells to the treatment. The TP53 status for the PPM1D wt DMG lines employed in these experiments is not clear.*

We agree that it is difficult to dissect whether the *PPM1D*-mutant DMGs are sensitive to MDM2 inhibitors exclusively on account of their *TP53* wild-type status from the data presented. To address this, we now include experiments evaluating the sensitivity of *PPM1D*tr and WT lines derived from our in-utero electroporation murine models. We find that the *PPM1D*tr cells are more sensitive to the MDM2 inhibitor AMG232 at low concentrations. This data is included in Supplemental Figure 8E, and also in the results as below:

“We also evaluated sensitivity of our endogenous Ppm1d truncated mouse NSC lines. Within these isogenic cell lines, Ppm1d truncated cells tended to be more sensitive to AMG-232 treatment compared to WT (LacZ) control cells (Supplementary Figure 8E).”

- *The inclusion of references (if published) or unpublished information regarding clinical and genomic and molecular characteristics associated with the BT869, SF776, and the wt PPM1D patient derived cell lines are needed to contextualize the data presented.*

Thank you. We have included the references and the molecular characteristics of the cell lines used in the study (Supplementary Table 2) and cite this table in the Results section.

“Our findings suggest that PPM1D mutations are a clonal driver event that enhances DMG formation. We therefore evaluated whether the expression of PPM1D was also necessary for the proliferation of an established PPM1D-mutant DMG cell line BT869 (Supplementary Figures 4A-B and Supplementary Table 2).”

- *Figure 4: results from independent Western Blot experiment should be represented and ideally bands quantified following contemporary standards for reproducibility.*

Thank you for the suggestion which we agree makes it easier to interpret the Western immunoblots which had all been repeated in at least three independent experiments. We now include these values and corresponding p values throughout the Results section and have added these figures to Supplementary Figure 5.

“We treated mNSCs generated by endogenously truncating *Ppm1d* at exon 6 (*sgPpm1d*) or non-targeting guides against *LacZ* (*sgLacZ*) with radiation and assessed response to DNA damage using previously described markers γ -H2AX and p-p53 (Ser15) at both baseline and five-hours post-radiation (Supplementary Figure 5B-C). We observed decreased phosphorylation of these markers in mNSCs with truncated *Ppm1d* in both conditions ($P < 0.05$). We also found similar decrease in phosphorylation of these markers in mNSCs overexpressing *PPM1Dtr* compared to those overexpressing GFP ($P < 0.05$) (Figure 4C and Supplementary Figure 5D). Next, we leveraged the tool PPM1D inhibitor to evaluate whether suppression of PPM1D was sufficient to reverse this phenotype. We treated two patient-derived *PPM1D*-mutant DMG cell lines, BT869 and SF7761 with the PPM1D inhibitor GSK2830371 at 10 μ M. Treatment with GSK2830371 by itself for 5 hours increased the levels of γ -H2AX and p-p53 (Ser15) in *PPM1D*-mutant DMG lines ($P < 0.05$) and this effect was further enhanced by radiation (Figure 4D).”

• As noted in the text (lane 342), Sup. Fig. 4A show considerably higher levels of *PPM1Dtr* relative to *PPM1D FL*, yet the effect of both variants on irradiated cells shown in Fig. 4 A,B are equivalent. The authors should comment on this discrepancy.

We thank the Reviewer for this observation and agree that it is interesting. We have been unable to express full-length *PPM1D* to equivalent levels as *PPM1Dtr*, most likely because of the putative degron in C-terminal tail of *PPM1D*. The loss of this tail in *PPM1Dtr* leads to enhanced stability and expression as has been previously reported (Kahn et al., 2018; Kleiblova et al., 2013). However, despite this limitation, our results suggest that even low levels of *PPM1D* overexpression are enough to confer this difference in apoptosis and progression through cell cycle checkpoints, especially after treatment with ionizing radiation and induction of DNA damage.

We have commented on this in the Results section of the manuscript as follows:

“Expression of *PPM1Dtr* was associated with attenuated apoptosis and more rapid cell cycling after radiation. We treated our mNSC models with ionizing radiation and assessed the percentage of apoptotic and cycling cells using flow-cytometry analysis of Annexin V/Propidium Iodide (PI) and BrdU incorporation, respectively. Compared to GFP controls, *PPM1Dtr* overexpression led to significantly lower rates of apoptosis at baseline ($11.5\% \pm 1.3\%$ and $6.1\% \pm 0.9\%$ respectively; $P < 0.05$), and with even greater effects 24 hours after 8 Gy of irradiation ($22.9\% \pm 2.3\%$ and $13.3\% \pm 2.4\%$, respectively; $P < 0.05$) (Figure 4A). *PPM1D FL* overexpression did not lead to a significant decrease in apoptosis at baseline, but decreased apoptosis 24 hours after 8 Gy of irradiation (12.8 ± 2.3 ; $P < 0.05$). At 24 hours post-radiation, the majority of GFP expressing cells remained in G0/G1 ($77.9\% \pm 7.1\%$), while only $6.3\% \pm 1.5\%$ had re-entered cell-cycling and were in S-phase. However, cells expressing *PPM1D FL* and *PPM1Dtr* exhibited a more rapid progression through the G1/S checkpoint, with $18\% \pm 1.4\%$ and $17\% \pm 1.5\%$ of cells respectively, observed to be in S phase ($P < 0.05$ and $P < 0.005$ respectively) (Figure 4B). **The suppression of apoptosis and G1/S cell cycle checkpoint by**

PPM1D FL suggests that even low levels of PPM1D overexpression might be enough to confer this difference in the context of DNA damage.”

• Fig 1G, methods (line 696) “transcripts in these populations were determined using the normalized and scaled counts”, please provide the parameters used for normalization/scaling for the scRNAseq data.

We apologize for this omission. The scRNAseq analysis was performed using Seurat using standard parameters and methods. The normalization and scaling were performed using NormalizeData and ScaleData functions in Seurat respectively. According to the vignette, “Seurat implements a global normalization-scaling method called LogNormalize that normalizes the gene expression measurements for each cell by the total expression, multiplies this by a scale factor (10,000 by default), and log-transforms the result.” Similarly, scaling shifts the gene expression of each gene such that the mean expression and variance across cells are 0 and 1 respectively.

However, given the suggestions from Reviewers 2 and 3 regarding the single-cell data, we have now removed these analyses from the manuscript.

• *The transcriptome analysis of GSK2830371-treated cells vs. control is important to support the premise that PPM1D suppresses p53. This experiment is limited to 1 time point (5h) and n=2 cell lines (apparently with no replicates for each cell line), which severely diminishes the statistical power and confidence on the results. Please provide an experimental basis for the single timepoint chosen and sample size calculation for the statistics.*

Thank you for these comments. Our analyses had indeed incorporated RNA-sequencing data generated from three technical replicates per condition for each cell line. We apologize that this wasn't clearly stated in the legend and have amended the legends.

Our original RNA-sequencing of PPM1D-mutant DMGs cells following inhibition of the phosphatase was guided by our experimental analyses that suggested that sequelae of PPM1D inhibition was evident five hours post treatment. In response to the Reviewers suggestion, we have now expanded this to include a time course that includes timepoints at 1, 5 and 24 hours post treatment (or vehicle control), including multiple replicates as described above. We found 353, 176, and 90 pathways to be differentially expressed at each of these time points respectively. The effects of PPM1D suppression on TP53 signaling was most evident at the 5 (Figure 4E)- and 24-hour (Supplementary Figure 4D) time points. We include these additional data in the Results section as below:

“To evaluate the transcriptional changes conferred by *PPM1D*tr in DMGs, we treated *PPM1D*-mutant DMGs BT869 and SF7761 with 10 μ M of GSK2830371 and compared their gene-expression profiles to those of DMSO treated cells at 1, 5 and 24 hours post treatment. We observed 114, 368, and 219 genes to be differentially expressed at 1, 5, and 24 hours-post treatment (LFC > 2, FDR < 0.25) in the GSK2830371 treated cells compared to vehicle controls (Figure 4E, Supplementary Figure 6A and Supplementary Table 3). The most significantly upregulated genes five hours post treatment included the chemokine CXCL8, ZBTB32, a member of the ZBTB family of transcription factors and TRIML2, which has been reported to enhance p53 SUMOylation (Kung, Khaku, Jennis, Zhou, & Murphy, 2015), while the most downregulated genes included the monocarboxylate transporter SLC16A3 and MYOD1, a bHLH transcription factor most well-known for its role in regulating myogenic differentiation (Fong & Tapscott, 2013) (Supplementary Table 3). At a pathway level, 353/1836, 176/1453 and 90/1858 pathways were significantly upregulated (FDR < 0.25) in the GSK2830371 treated condition relative to DMSO condition at 1 hr, 5 hr, and 24 hr time points respectively. Five hours post treatment, 8 of the 10 most significantly upregulated pathways included gene-sets associated with *TP53*-signaling and one with DDR. Moreover, of the remaining 166 pathways, another 10 were also related to *TP53* signaling, 12 with cell cycle regulation, and 9 with DDR (Figure 4F, Supplementary Figure 6B-C and Supplementary Table 4). 10 of these signaling pathways remained upregulated at the 24-hour time point.”

- *The assertion in lane 183: “PPM1D expression is enriched in progenitor cells in PPM1D-mutant DMGs”, should be qualified as referring to n=1 tumor analyzed.*

Thank you for this comment. Given the concerns raised by the other Reviewers regarding the sample size, we have removed these data from the revised manuscript since it only pertains to one tumor.

Other points:

- *Accession dates for the TCGA data from cBioPortal, TumorPortal, and Tumorscape should be provided.*

Thank you. We have added these data to the Methods as below.

“The data from these portals were accessed on August 2019 for adult tumors and January 2021 for pediatric tumors.”

- *Supplementary Tables did not format correctly in the PDF, information could not be properly accessed.*

Thank you for pointing this out. We will work with the Editor to ensure that the tables can be accessed.

- *Sup.Fig 4b: time points in the legend do not seem to be matching those in the figure. We appreciate the Reviewer for highlighting this error which we have now corrected.*

Reviewer #2 (Remarks to the Author): Expert in glioma molecular and functional assays

This manuscript confirms the presence of mutations of the PPM1D phosphatase from the whole genome sequencing of 131 pediatric glioma. As previously reported, PPM1D mutations are truncating events targeting the C-terminal region of PPM1D and result in gain-of-function oncogenic activation as shown by the ability of mutant PPM1D to cooperate to oncogenesis in association with other genetic alterations and promote more aggressive cancer phenotypes. Similarly, the authors also confirm that this effect requires the enzymatic phosphatase activity of PPM1D and PPM1D-driven oncogenesis converges on disabling the p53 pathway. The biological phenotype of PPM1D-mediated oncogenesis is associated with alterations of cell cycle regulation, the DNA damage response, radiation sensitivity and apoptosis.

Whereas the above conclusions are broadly supported by the results presented, none of them is novel as each of the functions associated with PPM1D in the manuscript was extensively and similarly shown by previous studies. Thus, the key weakness of the paper is that it lacks novelty.

We thank the Reviewer for their comments. We agree that *PPM1D* mutations have previously been reported in DMGs and other cancers, particularly in post-treatment samples such as clonal hematopoiesis of indeterminate potential (CHIP). However, the role of *PPM1D* mutations in enhancing *de novo* glioma formation in the primary brainstem or neural context has not been previously shown.

To address the Reviewers concern regarding novelty, we reviewed all previously published papers focused on functional roles of *PPM1D* mutations in gliomas. We identified three such papers (Akamandisa, Nie, Nahta, Hambardzumyan, & Castellino, 2019; Fons et al., 2019; Wang et al., 2020).

Our manuscript adds the following novel findings/methods:

1. We have optimized in utero electroporation methods to induce *de novo*, genetically and anatomically relevant tumors in the brainstems of mice. In contrast, none of the previous publications had similar *in vivo* model systems. Akamandisa et al. used cell lines previously transformed with *PDGFB* and *INK4a/ARF* loss (without H3 K27M mutation that define DMGs), in which they then transduced *PPM1Dtr* in vitro, followed by intracranial injection. Wang et al. utilized patient derived cell lines while Fons et al. used immortalized astrocytes transduced with *PPM1Dtr* (without any of the contextual mutations, particularly the H3 K27M mutation), which were then implanted subcutaneously as flank injections.

Moreover, we have generated and characterized two distinct models – one model which leverages CRISPR-Cas9 technology to faithfully recapitulate endogenous truncation of *Ppm1d* as occurs in human tumors, and a second model which overexpresses truncated *PPM1D* - allowing us to evaluate the role

of the phosphatase domain *in vivo*. In the revised manuscript, we also now include transcriptomic characterization of these models to show relevance with the human disease.

2. We leveraged our primary *in vivo* models of DMG to show for the first time that expression of *PPM1Dtr* is sufficient to enhance brainstem glioma formation, co-operating with the DMG defining histone mutation and also mutant *PDGFRA* expression. Moreover, our models allowed us to evaluate the function of *PPM1Dtr* relative to loss of *TP53*, revealing that *PPM1Dtr* was indeed sufficient to replace loss of *TP53*, and requires PPM1D phosphatase activity.
3. Unbiased, integrative genome-wide genomic, proteomic and functional approaches to delineate the mechanisms through which *PPM1Dtr* exerts its oncogenic effects. While previous manuscripts have observed altered DNA-damage responses in models that overexpress *PPM1Dtr*, with focused validation of potential therapeutic implications, our manuscript intersects high-throughput methods performed across multiple models, and includes characterization of global transcriptomic, proteomic and synthetic dependencies that are induced by *PPM1Dtr*. By integrating these independent methods, we show that the major functional sequelae of *PPM1Dtr* centers on the combination of attenuated *TP53* signaling, DNA damage responses and cell-cycle progression.
4. Identification and validation of MDM2 as a potential therapeutic target for *PPM1D* mutant DMGs. This finding may have direct clinical implications as biopsy-driven clinical trials that incorporate targeted inhibitors are being designed for children with DMG. In fact, our data has already provided an impetus for *PPM1D* genotyping as an inclusion criterion for a clinical trial of an MDM2 inhibitor in pediatric cancers (NCT03654716).

To highlight these novel points more clearly, we have:

1. Changed the title of our manuscript to '***PPM1D* mutations are oncogenic drivers of *de novo* Diffuse Midline Glioma formation**'
2. Restructured our manuscript to highlight our novel models and our evaluation of *PPM1Dtr* as a driver of *de novo* gliomas, and the integrative approach that we applied to dissect the mechanisms of oncogenesis.
3. Added each of these points to the Discussion, as below.

“Using *in utero* electroporation to examine the role of mutant *PPM1D* *in vivo*, our data is the first direct evidence that *PPM1D* truncations actively participates in *de novo* DMG development. Previous studies have leveraged flank implants of immortalized human astrocytes with endogenous *PPM1D* truncations, or orthotopic implants of murine glioma cells exogenously expressing *PPM1Dtr* (Akamandisa et al., 2019; Fons et al., 2019) but have not assessed the direct

contribution of mutant *PPM1D* to the transformation of neural stem/progenitor cells into DMG.”

“We demonstrate that when paired with expression of *Pdgfra* and histone K27M mutations, either truncation of endogenous *Ppm1d* or exogenous expression of *PPM1Dtr* is sufficient to enhance gliomagenesis to a similar degree as p53 loss of function, and that the PPM1D phosphatase is necessary for this phenotype.”

“In this study, multiple assays indicated that *PPM1Dtr* cooperatively drives DMG primarily by opposing the key functions of p53. First, genetic analyses found that *PPM1Dtr* mutations are mutually exclusive with *TP53* mutations in DMG suggesting that these mutations have overlapping oncogenic functions. Second, *in vivo* and *in vitro* tumor initiation studies found *PPM1Dtr* to be sufficient to replace dominant negative *TP53* to enhance glioma formation *in vivo* and abrogate *TP53*-mediated G1/M cell cycle arrest and apoptosis *in vitro*. Third, genetic and phosphoproteomic assays of DMG model systems identified *TP53*, cell cycle and cell cycle checkpoints, and DDR pathways—all of which are also functions of p53—as the primary pathways affected by perturbations of *PPM1Dtr*. Similarly, prior studies found genetic exclusivity between *PPM1D* and *TP53* alterations (Mackay et al., 2017; Zhang et al., 2014) and identified *PPM1Dtr*-associated chemotherapy/radiation resistance and clonal expansion in glial and hematopoietic compartments (Hsu et al., 2018; Kahn et al., 2018). However, several clues point to possible additional functions of *PPM1D* truncation in glioma development. For instance, *PPM1Dtr* but not *TP53* mutation co-occurs with *PIK3CA* mutations in DMG, suggesting a possible interaction with the MAPK and mTOR pathways and its potential role in tumorigenesis. Future studies will seek to fully dissect these possible mechanistic interactions.”

“Our data also identify nodes in the *PPM1D* genetic network as therapeutically actionable dependencies in *PPM1Dtr* DMGs. While our study implies that *PPM1Dtr* and *TP53* loss of function have overlapping oncogenic functions, tumors with either alteration are likely to harbor different genetic dependencies. For instance, the protein-ubiquitin ligase MDM2 which targets p53 for degradation is well-known to be a dependency in *TP53* wild-type cancers, but inhibition of MDM2 is ineffective in cancers that already harbor loss of p53 function (Her et al., 2018; Howard et al., 2019; Verreault et al., 2016). Importantly, emerging data suggest that p53 wild-type cancers that are driven by *PPM1D* gain of function are still susceptible to MDM2 inhibition (Stolte et al., 2018). Consistent with this observation, our current studies identify *MDM2* dependency as a potentially actionable feature of *PPM1D*-truncated DMGs, leading to genotyping for *PPM1D* as an inclusion criterion for a clinical trial of an MDM2 inhibitor in pediatric cancers (NCT03654716). However, multiple additional therapeutic avenues remain largely unexplored. For example, future studies should investigate additional nodes of the *PPM1D* molecular genetic network that could be therapeutically targeted both individually and in combination in *PPM1Dtr* DMG.”

Additional challenges concern the rather superficial state of a number of analyses and the lack of rigorous support for some of the least general claims. For example, the claim that the PPM1D gene harbors broad and low-level copy number gains is of uncertain significance and remains rather superficial. Based on what is shown in the manuscript, it is difficult to ascertain whether such low-level copy number gains are associated with any functional consequences.

Thank you for this comment.

We agree with the Reviewer that it is challenging to functionally validate the role of *PPM1D* amplifications. We did attempt to evaluate the role of full length *PPM1D* (*PPM1D* FL), however had difficulty in consistently inducing its expression in our model systems, possibly due to the presence of C' terminal degrons that regulate its degradation and are predicted to be lost in *PPM1Dtr* (see response to Reviewer 1). However, despite this limitation, we did find some evidence of a functional role for *PPM1D* FL on progression through cell cycle and apoptosis.

These data are included in the Results:

“Expression of *PPM1Dtr* was associated with attenuated apoptosis and more rapid cell cycling after radiation. We treated our mNSC models with ionizing radiation and assessed the percentage of apoptotic and cycling cells using flow-cytometry analysis of Annexin V/Propidium Iodide (PI) and BrdU incorporation, respectively. Compared to GFP controls, *PPM1Dtr* overexpression led to significantly lower rates of apoptosis at baseline ($11.5\% \pm 1.3\%$ and $6.1\% \pm 0.9\%$ respectively; $P < 0.05$), and with even greater effects 24 hours after 8 Gy of irradiation ($22.9\% \pm 2.3\%$ and $13.3\% \pm 2.4\%$, respectively; $P < 0.05$) (Figure 4A). *PPM1D* FL overexpression did not lead to a significant decrease in apoptosis at baseline, but decreased apoptosis 24 hours after 8 Gy of irradiation (12.8 ± 2.3 ; $P < 0.05$). At 24 hours post-radiation, the majority of GFP expressing cells remained in G0/G1 ($77.9\% \pm 7.1\%$), while only $6.3\% \pm 1.5\%$ had re-entered cell-cycling and were in S-phase. However, cells expressing *PPM1D* FL and *PPM1Dtr* exhibited a more rapid progression through the G1/S checkpoint, with $18\% \pm 1.4\%$ and $17\% \pm 1.5\%$ of cells respectively, observed to be in S phase ($P < 0.05$ and $P < 0.005$ respectively) (Figure 4B). **The suppression of apoptosis and G1/S cell cycle checkpoint by *PPM1D* FL suggests that even low levels of *PM1D* overexpression might be enough to confer this difference in the context of DNA damage.**”

Finally, we extend our analysis of copy-number alterations that exert positive or negative fitness to Diffuse Midline Gliomas. This analysis also revealed amplification of *PPM1D* to be significantly associated with an increased positive selection advantage (p

= 0.007). This is included in Supplemental Figure 1K and in the Supplemental Note as below:

“Across all 10,967 cancers, the distribution of event lengths on 17q differed most from the background distribution in a region containing 43 genes including *PPM1D*, suggesting positive selective pressure emanating from this locus (pan-cancer: FDR < 10^{-6}) (Supplementary Figure 10B). Several individual cancer types including melanoma and adult glioma also showed this pattern (melanoma: FDR < 6×10^{-6} ; glioma: FDR < 0.05) (Supplementary Figure 10D). We expanded our analysis to include pHGGs and observed similar evidence of positive selection associated with the locus containing *PPM1D* (Supplementary Figure 10C).”

Similarly, the claim that PPM1D expression is enriched in the cancer stem cell population of pediatric glioma is premature. The authors base this claim on the scRNAseq analysis of only one glioma. In the absence of solid validation in more representative cohorts of tumors analyzed by scRNAseq, such claim remains speculative.

We agree with the Reviewer that it is difficult to make conclusions with one tumor. We have therefore removed this analysis from our manuscript

Reviewer #3 (Remarks to the Author): Expert in glioma genomics

In this manuscript the authors have investigated pediatric high-grade gliomas, in particular diffuse midline gliomas, and the role of PPM1D truncating mutations have in these tumors. There are several interesting aspects in this manuscript, especially regarding the new mouse models which clearly demonstrate the oncogenic role of truncated PPM1D.

We thank Reviewer 3 for these comments, particularly for pointing out the novelty of *in vivo* models generated in the study which represent the first *in vivo* models of *de novo* *PPM1D* mutant DMGs and which also conclusively establish the role of mutant *PPM1D* in enhancing DMG oncogenesis.

However, there are also many aspects in the paper presented as new findings which are in my opinion not so entirely new and have been published before, but these papers are unfortunately not (well) cited. For instance, in my opinion the whole genomic section does not add much news to the paper and can be largely reduced as it is not new at all that PPM1D truncating mutations occur in DMGs and are mutually exclusive from TP53 mutations. This was for instance already presented in the large meta-analysis paper of MacKay et al. (2017) in which more than 1000 pediatric high grade gliomas have been analyzed.

We agree with Reviewer 3's statement that *PPM1D* mutations have previously been reported and apologize for omissions in citations. As indicated in our response to Reviewer 2 above, we have performed a systematic analysis of previous publications

that focused on the evaluation of the functional role of *PPM1D* mutations in DMGs and have ensured that these are cited in our manuscript. We also have completed a systematic analysis of all papers reporting the genomic landscapes of DMGs that highlight mutations in these cancers. We have ensured that we now cite each of these in our manuscript.

We also agree with the Reviewer's comments regarding the genomic analyses which masked our more novel findings regarding the role of *PPM1D* mutations as oncogenes in DMGs. In response, we have de-emphasized the genomic analyses and restructured the paper and figures to highlight our novel findings more clearly. In addition, we have also changed the title of our manuscript to '***PPM1D* mutations are oncogenic drivers of *de novo* Diffuse Midline Glioma formation**' to emphasize our key findings.

In our response to Reviewer 2, we indicate the novel models that we have generated. In addition, our manuscript contributes the following novel data and insights:

1. Evaluation of *PPM1D* mutations in another 59 new pHGGs, including 48 which are pretreatment samples. These data further support *PPM1D* mutations as being a driver of *de novo* tumor formation, which differs from adult clonal hematopoiesis where *PPM1D* mutations are selected for with treatment.
2. Analysis of additional pre- and post-treatment *PPM1D* mutant DMGs that highlight the clonality of *PPM1D* mutations.
3. Systematic analysis of frequency of *PPM1D* mutations across all pediatric cancers
4. Association of copy number gains of a locus containing *PPM1D* with a positive fitness advantage in DMGs, and other cancers
5. Transcriptomic and phospho-proteomic analyses of patient-derived DMGs
6. Genome-scale genetic perturbation assays combined with an integrative analysis to elucidate the mechanisms through which *PPM1D*-mutant DMGs exert oncogenesis.
7. Inhibition of MDM2 as a potential therapeutic strategy for *PPM1D*-mutant DMGs.

*Also it is totally unclear to me why the authors have compared the *PPM1D* alterations in their glioma data set to the TCGA dataset which only includes adult tumors and not for instance to other pediatric cancers.*

We agree with the Reviewer that expanding this analysis to also include other pediatric cancers is relevant and interesting. We include these analyses in Figures 1C and the Results section. Intriguingly, while *PPM1D* mutations are observed across lineages in adult cancers (especially clonal hematopoiesis and endometrial cancers), they tend to be mostly observed in gliomas in pediatric cancers.

"The spectrum of cancers associated with *PPM1D* mutations is distinct in children compared to adults. *PPM1D* mutations have previously been described in adult myeloid neoplasms including CHIP, myelodysplastic syndrome (MDS) and acute myeloid leukemia (AML) (Hsu et al., 2018; Kahn et al., 2018), but their incidence across other

solid cancers is unknown. To assess this, we extended our analysis of *PPM1D* mutations to include other pediatric and adult cancers (Figure 1C and Supplementary Note). Across adult cancers, we observed *PPM1D* mutations to be recurrent in 3% of endometrial cancers (Figure 1C and Supplementary Figure 1B). In contrast, *PPM1D* mutations appear to occur predominantly in gliomas in children. Among 41 pediatric cancers encompassing 13 histological subtypes, we found *PPM1D* truncating mutations in only 0.2% of all tumors, with gliomas being the top tumor type (1.37%) (Figure 1C). Within our DMG WGS cohort, we did not observe differences in the frequency of *PPM1D* mutations between pre- and post-treatment glioma samples (Fisher's exact test, $P = 1$), suggesting a role in enhancing *de novo* glioma formation. Taken together, these data support truncating *PPM1D* mutations as being contributors of *de novo* DMG gliomagenesis."

Furthermore, identification of PPM1D as a therapeutic target in DMGs is also not entirely new and has been published before in for instance Akamandisa et al. (2019) or Wang et al. (2020) and the link with DDR pathways has been made as well in these papers. Both papers are cited (although the citation for the Wang paper is incomplete), but still the data are kind of presented in this manuscript as if they are entirely new.

As we have noted in our response above (and in our response to Reviewer 2), previous publications have reported *PPM1D* mutant models to be sensitive to *PPM1D* inhibitor GSK2830371 and shRNA knockdown. We have ensured that these manuscripts are cited. In our manuscript, we leverage CRISPR-cas9 technology (including sequencing-based competition assays) to demonstrate that *PPM1D* is a genetic dependency across a range of *PPM1D*-mutant patient derived models. All prior studies reporting therapeutic implications have focused on targeted approaches, usually DNA damage responses. In contrast, we apply an unbiased integrative approach to comprehensively detect all therapeutic vulnerabilities. By integrating these independent methods, we were able to show that the major functional sequelae of *PPM1D*tr centers on the combination of attenuated *TP53* signaling, DNA damage responses and cell-cycle progression, which also represent the major therapeutic vulnerabilities.

We highlight this in the Discussion section as below:

"In this study, multiple assays indicated that *PPM1D*tr cooperatively drives DMG primarily by opposing the key functions of p53. First, genetic analyses found that *PPM1D*tr mutations are mutually exclusive with *TP53* mutations in DMG suggesting that these mutations have overlapping oncogenic functions. Second, *in vivo* and *in vitro* tumor initiation studies found *PPM1D*tr to be sufficient to replace dominant negative *TP53* to enhance glioma formation *in vivo* and abrogate *TP53*-mediated G1/M cell cycle arrest and apoptosis *in vitro*. Third, genetic and phosphoproteomic assays of DMG model systems identified *TP53*, cell cycle and cell cycle checkpoints, and DDR pathways—all of which are also functions of p53—as the primary pathways affected by perturbations of *PPM1D*tr. Similarly, prior studies found genetic exclusivity between *PPM1D* and *TP53* alterations (Mackay et al., 2017; Zhang et al., 2014) and identified

*PPM1D*tr-associated chemotherapy/radiation resistance and clonal expansion in glial and hematopoietic compartments (Hsu et al., 2018; Kahn et al., 2018). However, several clues point to possible additional functions of *PPM1D* truncation in glioma development. For instance, *PPM1D*tr but not *TP53* mutation co-occurs with *PIK3CA* mutations in DMG, suggesting a possible interaction with the MAPK and mTOR pathways and its potential role in tumorigenesis. Future studies will seek to fully dissect these possible mechanistic interactions.”

Other comments:

Abstract:

Here we analyze whole genomes sequences of 170 pediatric high-grade gliomas and find that truncating mutations in PPM1D are clonal driver events in 11% of DMGs and are enriched in primary DIPG.

Abbreviations should not be used in the abstract and moreover DIPGs are DMGs and the term DIPG is no longer used in the updated WHO classification.

We agree, and we have used the terminology DMG throughout the manuscript. We do use the term DIPG to describe the subset of pontine tumors within the DMGs, as described below:

“To further evaluate the role of *PPM1D* mutations in gliomagenesis, we first performed a comprehensive analysis of whole genome sequences (WGS) of 131 pre-treatment pediatric gliomas. This consisted of 76 DMGs, 58 of which were prototypical diffuse intrinsic pontine gliomas (DIPG) located in the brainstem, and 55 non-midline pHGGS. We also contrasted the whole genome sequences of pretreatment tumors to 39 post-treatment gliomas (34 DMGs including 33 DIPGs, and 5 non-midline HGG.”

We show that PPM1D mutations potentiate gliomagenesis and that PPM1D activity is required for in vivo oncogenesis.

Unclear from this sentence in the abstract that mutations lead to truncated proteins and how that is related to PPM1D activity.

We have edited the abstract to clarify mutations are indeed truncating:

“The role of *PPM1D* mutations in *de novo* gliomagenesis has not been systematically explored. Here we analyze whole genomes sequences of 170 pediatric high-grade gliomas and find that truncating mutations in *PPM1D* that increase the stability of its phosphatase are clonal driver events in 11% of DMGs and are enriched in primary pontine tumors.”

Together, these findings highlight *PPM1D* mutations to represent a targetable driver of pediatric gliomas.

PPM1D itself is not really targetable (only tool compounds available). Why not mentioning MDM2 here?

Thank you for suggesting this. We have included our MDM2 results in the abstract:

“Finally, we applied integrative phosphoproteomic and functional genomics assays and found that the oncogenic effects of *PPM1D* truncation converge on regulators of cell cycle, DNA damage response, and p53 pathways, revealing therapeutic vulnerabilities including MDM2 inhibition.”

Introduction:

Line 82: PPM1D truncating mutations that increase the stability of the phosphatase have also recently been found in clonal hematopoiesis of indeterminate potential (CHIP),

Unclear in the introduction that the mutations in DMGs are also truncating mutations

We agree that it is important for this to be clear. The Introduction includes the following text to describe *PPM1D* mutations as being truncating:

“*PPM1D*, also known as WIP1, is a PP2C family phosphatase that regulates known members of the DNA damage response (DDR) pathways, most notably p53, as well as other targets such as γ -H2AX, CHK1, ATM, and ATR. (Cha et al., 2010; Lu, Nannenga, & Donehower, 2005; Lu, Nguyen, & Donehower, 2005). *PPM1D* truncating mutations that increase the stability of the phosphatase have also recently been found in clonal hematopoiesis of indeterminate potential (CHIP), where it drives selective outgrowth of the mutant clones in response to cytotoxic chemotherapy (Hsu et al., 2018; Kahn et al., 2018).”

Line 91:maintenance of DMG cells, and finally, its mechanisms of action as an oncogene.

Unclear from the introduction that PPM1D should act as an oncogene.

We agree with the Reviewer that this is an important point to address. Our introduction includes the following text to describe previous work that implicates *PPM1D* mutations as resistance drivers.

“Previous work in gliomas has largely focused on the role of *PPM1D* mutation as a driver of radiation resistance and/or evaluated therapeutic vulnerabilities associated with the mutation (Akamandisa et al., 2019; Fons et al., 2019; Wang et al., 2020).”

However, the finding that *PPM1D* mutations are present in treatment naïve DMGs raise questions about its role in enhancing tumor formation. We also highlight this in the Introduction as below:

“However, to fully characterize *PPM1D* as a therapeutic target, a number of questions remain, including its role in DMG oncogenesis, its necessity for the proliferation and maintenance of DMG cells, and finally, its mechanisms of action as an oncogene.”

Finally, we have also changed the title of our manuscript to “***PPM1D* mutations are oncogenic drivers of *de novo* Diffuse Midline Glioma formation**” to highlight the focus of our work in addressing this question.

Line 95:that exogenous expression of mutant human PPM1D or endogenous truncation of murine Ppm1d together.....

Suggest to add ‘human’ to the sentence

We have amended the sentence as suggested:

“To this end we used novel murine models to demonstrate that endogenous truncation of murine *Ppm1d* together with histone and *Pdgfra* mutations is sufficient to drive *de novo* brainstem glioma formation, as is exogenous expression of truncated human *PPM1D* (*PPM1Dtr*).”

Line 107-110: This cohort consisted of 76 DMGs, including 58 diffuse intrinsic pontine gliomas (DIPG), which is the prototypical subtype of DMG, and 55 non midline HGGs. We contrasted these with whole genome sequences of 39 post-treatment gliomas (34 DMGs including 33 DIPGs, and 5 non-midline HGG).

Please use the correct nomenclature for these tumors according to the WHO classification, which no longer uses DIPG.

We agree, and we have used the terminology DMG throughout the manuscript. We do use the term DIPG to describe the subset of pontine tumors within the DMGs, as described below:

“To further evaluate the role of *PPM1D* mutations in gliomagenesis, we first performed a comprehensive analysis of whole genome sequences (WGS) of 131 pre-treatment pediatric gliomas. This consisted of 76 DMGs, 58 of which were prototypical diffuse intrinsic pontine gliomas (DIPG) located in the brainstem, and 55 non-midline pHGGs. We also contrasted the whole genome sequences of pretreatment tumors to 39 post-treatment gliomas (34 DMGs including 33 DIPGs, and 5 non-midline HGG).”

Were these genomic data all new or have they (partly) been published before? There are some references to previous publications regarding these data in the methods but it is totally unclear which part is new and which are already existing data. Also, will all data be deposited in a database to make them accessible for other researchers? Accession numbers are lacking.

Our analyses of 179 pHGGs includes 59 currently unpublished pHGGs, including 48 which are pretreatment samples. We are in the process of depositing the WGS data with corresponding RNA-sequencing to dbGaP under Accession Number phs002380.v1.p1.

RNA-seq data from our mouse models are currently being uploaded to GEO. Once the upload is complete, they can be found in the following directory:
prasidda_MF6pAYAo/GEO_submission_May28

Proteomic data has been deposited to MassIVE. These data are accessible at <ftp://MSV000085700@massive.ucsd.edu> with username: MSV000085700 and password: truncating. This dataset will be made public upon acceptance of the manuscript.

Also, what do these data add to the previous pan high grade glioma analyses published by Mackay et al. (2017)? Already in that paper it was clear that truncating PPM1D mutations are frequently found and mutually exclusive from TP53 mutations. This paper is also not referenced in this section.

In accordance with the Reviewers comments, we have restructured our manuscript to highlight the major findings of our manuscript which are related to the role of *PPM1D* mutations as an oncogenic driver of *de novo* DMGs.

We have also moved the majority of our genomic analyses to a Supplemental Note. We retain a comut that shows mutual exclusivity of *PPM1D* and *TP53* alterations as these data provide validation of this finding in an independent dataset of pHGGs. We also expand our analyses to compare the rates of *PPM1D* mutations across different cancer types, including both adult and pediatric tumors. Moreover, we report *PPM1D* to be a focus of low amplitude arm level amplifications in pHGGs and across other cancers.

Line 115-116:we did not observe differences in the frequency of PPM1D mutations between pre- and post-treatment glioma samples (Fisher's exact test, $P = 1$),.....

Were there any paired samples (primary vs treated) included in the cohort and if yes, how were the PPM1D mutations in those? Present in both or gained or lost in the treated samples?

Our cohort originally included one pre- and post-treatment samples pair from a patient with a *PPM1D* mutant DMG, including multiple regional samples obtained at autopsy.

We now add an additional pair of pre- and post-treatment samples from a patient with a DMG. In both cases, the *PPM1D* mutations were detected in both biopsy and autopsy samples, including the paired multi-regional post-treatment samples (originally presented in Figure 1). We now include data from both of these tumors in Supplemental Figure 1E-F and Supplementary Table 1 and in the Supplemental Note:

“Second, whole genome sequencing of two *PPM1D*-mutant DMGs for which paired pre and post treatment samples were available revealed both biopsy and autopsy samples to harbor the same *PPM1D* mutation with CCFs of 1 (Supplementary Table 11). This included one *PPM1D*-mutant DMG for which we had a biopsy obtained at diagnosis and six matched multiregional samples obtained at autopsy from the same patient revealed the same *PPM1D* mutations in all samples (mean VAF = 0.64 ± 0.03 , median CCF = 1) (Supplementary Figure 1D-E). Taking these analyses together, we conclude that *PPM1D* mutations are early events in gliomagenesis.”

Line 119-120: ..PPM1D mutations were mutually exclusive with TP53 mutations (Fisher’s exact test, $P < 0.0001$) (Figure 1C),....

That PPM1D mutations are mutually exclusive from TP53 mutations is already totally clear from Figure 1B and Figure 1C does not add anything to that. When checking the mutual exclusivity with other genes in Figure 1C, it would be helpful to add these then also to Figure 1A.

Thank you for this suggestion. We have included these genes in a comut plot in Supplementary Figure 9A (corresponding to the genomic description in Supplementary Note).

We also agree that Figure 1A (comut plot) confirms the mutual exclusivity of *PPM1D* mutations with *TP53* mutations (while the original 1B depicted the location of *PPM1D* SNVs). However, the purpose of 1C (now Supplementary Figure 1A) is to comprehensively show all genes which are either significantly mutually exclusive or co-occurring with *PPM1D* mutations. In particular, this Figure shows that while *PPM1D* mutations are mutually exclusive with *TP53*, they co-occur with *PIK3CA* mutations.

Figure 1A:

Annotation below figure for location is not correct as it shows partly location and partly tumor type instead of location.

Thank you. We have corrected the comut plot to include location – pons, midline (non-pontine), and hemispheric.

Also, in order to make the figure more clear I would suggest to sort all samples by tumor type and not by mutation.

Thank you for this suggestion. We have included a comut plot in Figure 1 that is sorted by SNVs to allow for comparison of *PPM1D* mutant tumors to non-*PPM1D* mutant

tumors. In response to the Reviewer, we also include a comut plot sorted by location in Supplementary Figure 9A.

Figure 1B:

Y-axis with # mutations does not align with the lollipops in the figure.

We apologize for this error which we have now corrected.

Line 126-129:

PPM1D mutations have also been described in myeloid neoplasms including CHIP, myelodysplastic syndrome (MDS) and acute myeloid leukemia (AML) (Hsu et al., 2018; Kahn et al., 2018), but their incidence across other solid cancers is unknown. To assess this, we extended our analysis of PPM1D mutations to include other cancers (Supplementary Figure 1A) in the TCGA dataset.

The TCGA dataset only includes adult tumors. Why did the authors not include pediatric tumors here, which would have made more sense as a comparison?

Line 147-148: In addition to truncating PPM1D mutations, PPM1D has also been reported to be amplified in several cancer types, most notably breast, mesothelioma, and liver (TCGA; Supplementary Figure 1A). Again, what about the pediatric cancers?

We agree with the Reviewer that expanding this analysis to also include other pediatric cancers is relevant and interesting. We include these analyses in Figures 1C and the Results section. Intriguingly, while *PPM1D* mutations are observed across lineages in adult cancers (especially clonal hematopoiesis and endometrial cancers), *PPM1D* mutations tend to be restricted to gliomas in pediatrics.

“The spectrum of cancers associated with *PPM1D* mutations is distinct in children compared to adults. *PPM1D* mutations have previously been described in adult myeloid neoplasms including CHIP, myelodysplastic syndrome (MDS) and acute myeloid leukemia (AML) (Hsu et al., 2018; Kahn et al., 2018), but their incidence across other solid cancers is unknown. To assess this, we extended our analysis of *PPM1D* mutations to include other pediatric and adult cancers (Figure 1C and Supplementary Note). Across adult cancers, we observed *PPM1D* mutations to be recurrent in 3% of endometrial cancers (Figure 1C and Supplementary Figure 1B). In contrast, *PPM1D* mutations appear to occur predominantly in gliomas in children. Among 41 pediatric cancers encompassing 13 histological subtypes, we found *PPM1D* truncating mutations in only 0.2% of all tumors, with gliomas being the top tumor type (1.37%) (Figure 1C). Within our DMG WGS cohort, we did not observe differences in the frequency of *PPM1D* mutations between pre- and post-treatment glioma samples (Fisher’s exact test, $P = 1$), suggesting a role in enhancing *de novo* glioma formation. Taken together,

these data support truncating *PPM1D* mutations as being contributors of *de novo* DMG gliomagenesis.”

Line 148-150: Analysis of our copy-number data of pHGGs revealed broad low-level gains in the PPM1D containing region on 17q, but no focal amplification of PPM1D was detected (Supplementary Figure 1D).

Instead of showing all chromosomes in this Figure, it would have been better to only show chromosome 17 as from the current figure it is impossible to judge whether there are focal amplifications of PPM1D or not.

Thank you. We have included a figure showing copy-number profiles of chromosome 17q across samples in Supplementary Figure 10A, and also refer to it in Supplementary Note.

“In addition to truncating *PPM1D* mutations, analysis of our copy-number data of pHGGs revealed broad low-level gains in the *PPM1D* containing region on 17q, but no focal amplification of *PPM1D* was detected (Supplementary Figure 10A).”

Line 150-155: Extending our analysis to other cancers types in the TCGA dataset, we observed PPM1D amplifications in 2% of 10,967 cancers comprising of 32 cancer types. These amplifications were statistically recurrent in several cancer types including bladder urothelial carcinoma (q-value = 0.0452), breast adenocarcinoma (q-value = 3.93×10^{-5}), epithelial cancers (q-value = 2.04×10^{-31}), liver hepatocellular carcinoma (q-value = 2.68×10^{-4}), colorectal cancers (q-value = 0.0111), colon adenocarcinoma (q-value = 0.0704) and melanomas (q-value = 0.244).

Not sure what this adds to the paper, especially as no pediatric cancers were included, which are known to harbor PPM1D amplifications as well.

Our analyses reveal *PPM1D* mutations to be relevant across cancers, and therefore a potential therapeutic target beyond DMGs. We also agree with the Reviewer that a survey of pediatric cancers is important, and we thank them for this suggestion. We now include these analyses in our revised manuscript. We have also moved these analyses to the supplementary note to highlight more novel findings of our work in Figure 1.

“In addition to truncating *PPM1D* mutations, *PPM1D* has also been reported to be amplified in several cancer types, most notably breast, mesothelioma, and liver (TCGA; Supplementary Figure 1G). Among pediatric tumors, *PPM1D* was amplified in 1.7% of all tumors, mainly in breast, stomach, mesothelioma and uterine. Analysis of our copy-number data of pHGGs revealed broad low-level gains in the *PPM1D* containing region on 17q, but no focal amplification of *PPM1D* was detected (Supplementary Figure 1H). Extending our analysis to other adult cancers in the TCGA dataset, we observed *PPM1D* amplifications in 2% of 10,967 cancers comprising of 32 cancer types. These

amplifications were statistically recurrent in several cancer types including bladder urothelial carcinoma (q-value = 0.0452), breast adenocarcinoma (q-value = 3.93×10^{-5}), epithelial cancers (q-value = 2.04×10^{-31}), liver hepatocellular carcinoma (q-value = 2.68×10^{-4}), colorectal cancers (q-value = 0.0111), colon adenocarcinoma (q-value = 0.0704) and melanomas (q-value = 0.244). However, *PPM1D* was located in the consensus amplification peak predicted to contain oncogenic driver gene only in bladder urothelial carcinoma but not in any of the other cancer types.”

[REDACTED]

Line 170-172: Taken together, these data support truncating PPM1D mutations as being contributors to DMG gliomagenesis. Furthermore, the rate of copy-number gains and amplifications of the region containing PPM1D across various cancer types suggest a possible dosage effect of wild-type PPM1D.

The last part has only been analyzed in adult cancers and not in the pediatric gliomas. Also, to make any conclusions about a possible dosage effect, one need to integrate transcriptome data as well then, again focused on the pediatric gliomas.

Thank you for this suggestion. Our analysis reveals a positive correlation between *PPM1D* copy-number and expression in pediatric gliomas ($p < 0.0001$). We have included these analyses in the Supplemental Note and Supplemental Figure 10E.

“Moreover, we observed a significant correlation between *PPM1D* copy-number and expression level in a cohort of 116 pHGGs, including 9 *PPM1D* mutant tumors ($R = 0.41$, $P < 5.6e-06$) (Supplementary Figure 10E).”

Line 182-183: These findings suggest that PPM1D expression is enriched in progenitor cells in PPM1D-mutant DMGs.

As PPM1D expression is not enriched in progenitor cells of tumors that are wild type for PPM1D, this would speak against a possible dosage effect of wildtype PPM1D in these tumors. How is the PPM1D expression in general in DMGs vs HGGs (and vs other brain tumors?), and in PPM1D mutant vs wildtype tumors?

This is an interesting question. We have performed these analyses at the Reviewers suggestion. *PPM1D*-mutant gliomas have increased expression of *PPM1D* relative to wild-type tumors (p value < 0.05), however we do not observe differences between DMGs and other pHGGs. We have included these results in Supplementary Figure 10F-G and the Supplemental Note.

“*PPM1D* expression was also significantly higher in *PPM1D*-mutant gliomas compared to the wild-type ones (Supplementary Figure 10F), but no difference was observed between DMGs and hemispheric gliomas (Supplementary Figure 10G).”

Line 187-188: The spectrum of PPM1D alterations in DMGs nominate both full-length and truncated PPM1D as potential oncogenes in de novo gliomagenesis. What is the evidence for the full length as there were no amplifications found?

We have amended this sentence to focus on *PPM1D* mutations.

‘Taken together, these data support truncating *PPM1D* mutations as being contributors to DMG gliomagenesis.’

Lines 218-220: We therefore reasoned that PPM1D mutations may cooperate with other oncogenes to enhance gliomagenesis. We evaluated this with IUE, electroporating sgPpm1d exon6 with concurrent H3.3K27M and PdgfraD842V PiggyBac plasmids.

This makes sense indeed, and from other publications it is indeed known that PDGFRA plays a role in DMGs, but the rationale why including PDGFRA mutant is lacking from their own genomic data as the authors did not include the analyses for PDGFRA mutations / amplifications etc. in their own cohort. This should have done as well then.

We agree that this strengthens our manuscript. We now show *PDGFRA* amplifications and mutations in the comut plot in Supplemental Figure 9A. Curiously, while *PDGFRA*

amplifications/mutations do not tend to be enriched in *PPM1D*-mutant gliomas in our cohort, we find *PPM1D*-mutant DMGs to express similar levels of *PDGFRA* as *PPM1D*-WT tumors (Reviewer Figure 1 shown below). We hypothesize that these tumors may harbor other mechanisms through which *PDGFRA* is activated, for example through enhancer hijacking similar to mechanisms reported in a recent study (Chen et al., 2020).

Reviewer Figure 1

Reviewer Figure 1. Violin plot comparing *PDGFRA* expression levels between *PPM1D*-mutant (n=9) and wild-type (n=107) human pHGG.

Line 220-222: Control IUE conditions resulted in a partially penetrant phenotype, with only 50% (9/18) of mice developing neurological symptoms related to tumor, with a median survival of 85 days postnatal.

Unclear from the text what these control conditions are then. Becomes only clear from the Figure.

We apologize that this was not clear and have corrected this as below.

“Control IUE (guides targeting LacZ with concurrent *H3.3^{K27M}* and *Pdgfra^{D842V}* PiggyBac plasmids) conditions resulted in a partially penetrant phenotype, with only 50% (9/18) of mice developing neurological symptoms related to tumor, with a median survival of 85 days postnatal (Figure 1D).”

Line 222-223: In contrast, C-terminal truncation of Ppm1d by sgPpm1d_{exon6} was sufficient to generate fully penetrant brainstem gliomas,.....

Yes, but only in combination with H3.3K27M and PdgfraD842V. However, it remains unclear whether these are both needed. Authors should have at least tested also the

combination of sgPpm1d^{exon6} plus H3.3^{K27M}, but preferably also the combination of sgPpm1d^{exon6} plus Pdgfra^{D842V}. Unclear why this was not done.

The reviewer is correct that *PPM1D^{tr}* is not sufficient to induce glioma formation by itself, however, is sufficient to enhance glioma formation in the presence of *H3.3^{K27M}* and *Pdgfra^{D842V}*.

We have performed the additional experiments suggested by the Reviewer. Our results suggest that *PPM1D^{tr}* is not sufficient to enhance glioma formation in the context of histone mutations alone (which is also insufficient as a single driver). However, when paired with *Pdgfra^{D842V}* mutations, *Ppm1d* truncations are sufficient to promote glioma formation. This agrees with our prior work using DNp53, which when combined with *Pdgfra^{D842V}* also promotes gliomagenesis (Patel et al., 2020).

“Additional *sgPpm1d^{exon6}* IUE combinations tested recapitulated prior findings in DNp53 glioma models (Patel et al., 2020), where expression of *H3.3^{K27M}* and *sgPpm1d^{exon6}* was not sufficient to induce gliomas, while *Pdgfra^{D842V}* and *sgPpm1d^{exon6}* drove efficient tumorigenesis with an extended latency (Supplementary Figure 1C).”

Also surprised to see why the mouse models were not further molecularly characterized, for instance using transcriptomics, to see how well they represent the human tumors. This would also be helpful with respect to the downstream analyses of PPM1D mutants as this was all done now in cell line models only.

We agree with the Reviewer that it is important to validate our models as being representative of human DMGs. We have applied a number of analyses to address this.

1. Comparison of the morphology/histopathology of the mouse tumors to human tumors, performed by a Neuropathologist.
2. Immunohistochemistry to confirm expression of markers commonly expressed in DMGs.
3. Evaluation of the overlap of the most differential expressed genes in *PPM1D^{tr}* mouse tumors and human *PPM1D*-mutant DMGs.

The results are described as below.

“*sgPpm1d^{exon6}* IUE brainstem tumors exhibited features that are also observed in human DMGs. GFP-positive *sgPpm1d^{exon6}* IUE brainstem tumors harbored truncations in exon 6 of *Ppm1d* (Supplementary Figure 1D-F) and displayed histopathological traits of high-grade glioma (Figure 1E-F). IUE generated murine gliomas showed characteristic histological features seen in human diffuse midline gliomas of the pons (Figure 1E-F). Tumors exhibited diffuse single cell infiltration of the brainstem parenchyma with moderate atypia and mitoses consistent with a high-grade glioma and

equivalent to WHO 2016 Grade 3 or 4. Immunohistochemical analysis (Figure 1F) showed the expression of GFP was retained in tumor cells uniformly and at high levels and cells retained defining lineage markers of DMG including diffuse Olig2 and Gfap labeling. The proliferation rate by KI67 staining showed more than 50 % of glioma cells positive, similar to human DMGs.

We next compared mouse and human DMG transcriptomes. We leveraged RNA-sequencing of *sgPpm1d*^{exon6} IUE brainstem tumors to identify highly expressed genes and compared this profile with that of *PPM1D*-mutant human tumors. 209/14481 genes were found to be highly expressed (Z-score>2) in the *sgPpm1d*^{exon6} tumors (Figure 2A and Supplementary Table 1). 152 of these 209 genes were also highly *PPM1D*-mutant human DMGs (Figure 2B), representing a significant overlap between the two conditions (p<0.0001). Taken together with the histopathological comparisons of these tumors, *Ppm1d*-mutant IUE brainstem tumors are representative of the tumors observed in human DMG patients.”

Line 299-307: Our finding that PPM1D is necessary for PPM1D-mutant DMGs extended to other PPM1D-expressing cancers. We examined whether PPM1D dependency was linked to genetic activation of PPM1D using pooled CRISPR-cas9 assays across 558 cancer cell lines (Ghandi et al., 2019; Meyers et al., 2017). We hypothesized that TP53-WT cell lines would be more dependent on PPM1D than TP53-mutant cell lines because of PPM1D’s role in opposing p53 function. As expected, lines with wild-type TP53 were significantly more dependent on PPM1D than TP53-mutant lines (Figure 3D). Among TP53-WT lines, PPM1D copy-number gain was associated with significantly higher probability of dependency on PPM1D (Figure 3D). We conclude that PPM1D is required for proliferation of p53 wild-type cell lines, particularly those that harbor PPM1D activating genetic alterations.

Figure 3D shows that TP53 wt cell lines depend on the expression of PPM1D, even when PPM1D is not amplified and these cell lines do also not have a PPM1D mutation as these mostly occur in the lines with TP53 mutations in this cell line panel. However, how does this match then with the results of the DMG lines that shows that cells do NOT depend on PPM1D when not mutated?

We thank the Reviewer for making this interesting observation regarding the small number of *TP53* WT lines that do not harbor *PPM1D* amplifications or mutations but are dependent on *PPM1D* expression. We speculate that these lines may have cryptic features that also predict *PPM1D* dependency such as *MDM2* amplification and other alterations that confer dependency on reactivation of the p53 and DDR pathways.

Figure 3F: is this not just an additive effect?

It is indeed possible that the effect observed in Figure 3F is additive and will have amended the text as below.

“However, the combination of these two treatments further decreased the viability of these cells over time (BT869: mean fold change at day 5 1.4 ± 0.1 ($P < 0.0001$); SF7761 mean fold change at day 5 1.5 ± 0.0 ($P < 0.0001$)) (Figures 3E-F), exhibiting a possible additive effect.”

Using cell line models the authors further investigated the downstream effects of mutant PPM1D using transcriptomics and mass spec analyses to analyze the phosphoproteome. Several interesting candidates came out of these analyses, mostly concentrated around the TP53 and/or DDR pathways. It would have been helpful if some these targets could have been validated (by mass spec or IHC) also in the mouse tumors (by comparing the ones with and without PPM1Dtr), but also in human DMGs that are mutated or not for PPM1D to see to which extent these identified targets indeed have a role in the tumors.

Our phosphoproteomic data revealed p53 and H2AX to be amongst the most differentially altered phosphosites in the human cell lines. We have validated these in our mouse models and include these data in Figures 4C and Supplemental Figure 5B.

We also agree that validating these in human DMGs would have been interesting, however, we are challenged by the lack of true isogenic human tumor models as PPM1D WT tumors tend to have TP53 mutation or other alterations that are predicted to affect TP53 and DDR responses.

Discussion

Line 494-496: Our integrative analyses with an array of models and genomic, proteomic and functional assays (Figure 7), including the development of novel mouse models of PPM1D-mutant DMGs, nominate PPM1D as a tractable therapeutic target in DMG.

It was already known that PPM1D forms a therapeutic target in DMGs. Published in several papers, which are not cited here.

Thank you. We agree that previous publications have shown sensitivity of PPM1D mutant cells to the tool PPM1D inhibitor and shRNA knockdown and we have cited these papers in the Discussion. We have also amended our first paragraph of the Discussion to highlight our novel finding that PPM1D mutations are sufficient to enhance *de novo* DMG formation.

“Our integrative analyses with an array of models and genomic, proteomic and functional assays (Figure 7), including the development of novel mouse models of PPM1D-mutant DMGs, nominate PPM1D as an oncogenic driver in *de novo* DMG. We confirm previous reports suggesting truncating mutations in exon 6 of PPM1D are clonal driver events in DMG (Nikbakht et al., 2016; Vinci et al., 2018) and show that they are sufficient to enhance *de novo* glioma formation. Moreover, we find expression of PPM1Dtr to be necessary for proliferation of PPM1D-mutant DMG cells, and, applying orthogonal approaches, we find that these effects are largely mediated through PPM1Dtr’s role in regulating TP53, DDR and cell cycle pathways”

“The current results nominate *PPM1D* as a therapeutic target for DMGs, both in solo *PPM1D*-targeting therapy and by enhancing DDR activators such as γ -H2AX and p53 phosphorylation when combined with DNA-damaging agents including radiation and chemotherapy. These findings are consistent with previous reports that *PPM1D* is a dependency when suppressed by shRNA or tool *PPM1D* inhibitors (Akamandisa et al., 2019; Wang et al., 2020). While there are currently no clinically relevant *PPM1D* inhibitors, tool compounds that inhibit its phosphatase domain have been developed, providing optimism for future efforts to generate compounds with favorable pharmacokinetic and brain penetrant properties (Gilmartin et al., 2014; Rayter et al., 2008). Notably, *TP53*-mutant/*PPM1D*-WT DMGs would not be predicted to benefit from *PPM1D* inhibition since *PPM1D* acts upstream of p53. These findings imply that *PPM1D* and *TP53* mutation status should be considered as biomarkers for response in investigations of *PPM1D*-directed therapies in DMG.”

Line 496: We show that truncating mutations in exon 6 of PPM1D are clonal driver events in DMG....

Is also not new, and has been shown before in the paper by Nikbakht et al. (2015). Should have been cited then as well.

We have cited the Nikbakht paper as requested.

“Our integrative analyses with an array of models and genomic, proteomic and functional assays (Figure 7), including the development of novel mouse models of *PPM1D*-mutant DMGs, nominate *PPM1D* as an oncogenic driver in *de novo* DMG. We confirm previous reports suggesting truncating mutations in exon 6 of *PPM1D* are clonal driver events in DMG (Nikbakht et al., 2016; Vinci et al., 2018) and show that they are sufficient to enhance *de novo* glioma formation.”

Line 497: Moreover, we find expression of PPM1Dtr to be necessary for proliferation of PPM1D-mutant DMG cells...

Again, already demonstrated before.....

Thank you. We agree that previous publications have shown sensitivity of *PPM1D* mutant cells to the tool *PPM1D* inhibitor and shRNA knockdown and have cited this as described above.

References:

- Akamandisa, M. P., Nie, K., Nahta, R., Hambardzumyan, D., & Castellino, R. C. (2019). Inhibition of mutant PPM1D enhances DNA damage response and growth suppressive effects of ionizing radiation in diffuse intrinsic pontine glioma. *Neuro Oncol*, *21*(6), 786-799. doi:10.1093/neuonc/noz053
- Cha, H., Lowe, J. M., Li, H., Lee, J. S., Belova, G. I., Bulavin, D. V., & Fornace, A. J., Jr. (2010). Wip1 directly dephosphorylates gamma-H2AX and attenuates the DNA damage response. *Cancer Res*, *70*(10), 4112-4122. doi:10.1158/0008-5472.CAN-09-4244
- Chen, C. C. L., Deshmukh, S., Jessa, S., Hadjadj, D., Lisi, V., Andrade, A. F., . . . Jandolo, N. (2020). Histone H3.3G34-Mutant Interneuron Progenitors Co-opt PDGFRA for Gliomagenesis. *Cell*, *183*(6), 1617-1633 e1622. doi:10.1016/j.cell.2020.11.012
- Fong, A. P., & Tapscott, S. J. (2013). Skeletal muscle programming and re-programming. *Curr Opin Genet Dev*, *23*(5), 568-573. doi:10.1016/j.gde.2013.05.002
- Fons, N. R., Sundaram, R. K., Breuer, G. A., Peng, S., McLean, R. L., Kalathil, A. N., . . . Bindra, R. S. (2019). PPM1D mutations silence NAPRT gene expression and confer NAMPT inhibitor sensitivity in glioma. *Nat Commun*, *10*(1), 3790. doi:10.1038/s41467-019-11732-6
- Gilmartin, A. G., Faitg, T. H., Richter, M., Groy, A., Seefeld, M. A., Darcy, M. G., . . . Kumar, R. (2014). Allosteric Wip1 phosphatase inhibition through flap-subdomain interaction. *Nat Chem Biol*, *10*(3), 181-187. doi:10.1038/nchembio.1427
- Her, N. G., Oh, J. W., Oh, Y. J., Han, S., Cho, H. J., Lee, Y., . . . Nam, D. H. (2018). Potent effect of the MDM2 inhibitor AMG232 on suppression of glioblastoma stem cells. *Cell Death Dis*, *9*(8), 792. doi:10.1038/s41419-018-0825-1
- Howard, T. P., Arnoff, T. E., Song, M. R., Giacomelli, A. O., Wang, X., Hong, A. L., . . . Roberts, C. W. M. (2019). MDM2 and MDM4 Are Therapeutic Vulnerabilities in Malignant Rhabdoid Tumors. *Cancer Res*, *79*(9), 2404-2414. doi:10.1158/0008-5472.CAN-18-3066
- Hsu, J. I., Dayaram, T., Tovy, A., De Braekeleer, E., Jeong, M., Wang, F., . . . Goodell, M. A. (2018). PPM1D Mutations Drive Clonal Hematopoiesis in Response to Cytotoxic Chemotherapy. *Cell Stem Cell*, *23*(5), 700-713 e706. doi:10.1016/j.stem.2018.10.004
- Kahn, J. D., Miller, P. G., Silver, A. J., Sellar, R. S., Bhatt, S., Gibson, C., . . . Ebert, B. L. (2018). PPM1D-truncating mutations confer resistance to chemotherapy and sensitivity to PPM1D inhibition in hematopoietic cells. *Blood*, *132*(11), 1095-1105. doi:10.1182/blood-2018-05-850339
- Kleiblova, P., Shaltiel, I. A., Benada, J., Sevcik, J., Pechackova, S., Pohlreich, P., . . . Macurek, L. (2013). Gain-of-function mutations of PPM1D/Wip1 impair the p53-dependent G1 checkpoint. *J Cell Biol*, *201*(4), 511-521. doi:10.1083/jcb.201210031
- Kung, C. P., Khaku, S., Jennis, M., Zhou, Y., & Murphy, M. E. (2015). Identification of TRIML2, a novel p53 target, that enhances p53 SUMOylation and regulates the transactivation of proapoptotic genes. *Mol Cancer Res*, *13*(2), 250-262. doi:10.1158/1541-7786.MCR-14-0385
- Lu, X., Nannenga, B., & Donehower, L. A. (2005). PPM1D dephosphorylates Chk1 and p53 and abrogates cell cycle checkpoints. *Genes Dev*, *19*(10), 1162-1174. doi:10.1101/gad.1291305

- Lu, X., Nguyen, T. A., & Donehower, L. A. (2005). Reversal of the ATM/ATR-mediated DNA damage response by the oncogenic phosphatase PPM1D. *Cell Cycle*, 4(8), 1060-1064. Retrieved from <https://www.ncbi.nlm.nih.gov/pubmed/15970689>
- Mackay, A., Burford, A., Carvalho, D., Izquierdo, E., Fazal-Salom, J., Taylor, K. R., . . . Jones, C. (2017). Integrated Molecular Meta-Analysis of 1,000 Pediatric High-Grade and Diffuse Intrinsic Pontine Glioma. *Cancer Cell*, 32(4), 520-537 e525. doi:10.1016/j.ccell.2017.08.017
- Nikbakht, H., Panditharatna, E., Mikael, L. G., Li, R., Gayden, T., Osmond, M., . . . Nazarian, J. (2016). Spatial and temporal homogeneity of driver mutations in diffuse intrinsic pontine glioma. *Nat Commun*, 7, 11185. doi:10.1038/ncomms11185
- Patel, S. K., Hartley, R. M., Wei, X., Furnish, R., Escobar-Riquelme, F., Bear, H., . . . Phoenix, T. N. (2020). Generation of diffuse intrinsic pontine glioma mouse models by brainstem-targeted in utero electroporation. *Neuro Oncol*, 22(3), 381-392. doi:10.1093/neuonc/noz197
- Rayter, S., Elliott, R., Travers, J., Rowlands, M. G., Richardson, T. B., Boxall, K., . . . Ashworth, A. (2008). A chemical inhibitor of PPM1D that selectively kills cells overexpressing PPM1D. *Oncogene*, 27(8), 1036-1044. doi:10.1038/sj.onc.1210729
- Stolte, B., Iniguez, A. B., Dharia, N. V., Robichaud, A. L., Conway, A. S., Morgan, A. M., . . . Stegmaier, K. (2018). Genome-scale CRISPR-Cas9 screen identifies druggable dependencies in TP53 wild-type Ewing sarcoma. *J Exp Med*, 215(8), 2137-2155. doi:10.1084/jem.20171066
- Verreault, M., Schmitt, C., Goldwirt, L., Pelton, K., Haidar, S., Levasseur, C., . . . Idbaih, A. (2016). Preclinical Efficacy of the MDM2 Inhibitor RG7112 in MDM2-Amplified and TP53 Wild-type Glioblastomas. *Clin Cancer Res*, 22(5), 1185-1196. doi:10.1158/1078-0432.CCR-15-1015
- Vinci, M., Burford, A., Molinari, V., Kessler, K., Popov, S., Clarke, M., . . . Jones, C. (2018). Functional diversity and cooperativity between subclonal populations of pediatric glioblastoma and diffuse intrinsic pontine glioma cells. *Nat Med*, 24(8), 1204-1215. doi:10.1038/s41591-018-0086-7
- Wang, Z., Xu, C., Diplas, B. H., Moure, C. J., Chen, C. J., Chen, L. H., . . . Yan, H. (2020). Targeting Mutant PPM1D Sensitizes Diffuse Intrinsic Pontine Glioma Cells to the PARP Inhibitor Olaparib. *Mol Cancer Res*. doi:10.1158/1541-7786.MCR-19-0507
- Zhang, L., Chen, L. H., Wan, H., Yang, R., Wang, Z., Feng, J., . . . Yan, H. (2014). Exome sequencing identifies somatic gain-of-function PPM1D mutations in brainstem gliomas. *Nat Genet*, 46(7), 726-730. doi:10.1038/ng.2995

REVIEWER COMMENTS

Reviewer #1 (Remarks to the Author):

The authors have satisfactorily addressed the concerns raised, and the manuscript is sufficiently improved.

Reviewer #2 (Remarks to the Author):

As stated in my comments from the first round of review, the main weakness of this manuscript remains the incremental degree of novelty, a point similarly raised by another reviewer. Nevertheless, I appreciate the authors' efforts to highlight the new findings primarily concerning the development of an in utero electroporation method to induce tumors in the brainstem. The possibility to successfully target MDM2 is also potentially interesting. It is reassuring that the authors have discussed the limited novelty and other limitations of the study in the revised manuscript.

Reviewer #3 (Remarks to the Author):

The manuscript has improved a lot and I have no further major comments. Only a few minor comments:

Abstract

Should write DMG full out

Introduction:

Diffuse midline gliomas H3K27M-mutant (DMG)

In the updated 5th edition of the WHO classification of CNS tumors, the name of these tumors has changed to "Diffuse midline gliomas H3K27-altered as not all of them have H3K27 mutations but there are other mechanisms that lead to low H3K27me3 levels in these tumors.

we first performed a comprehensive analysis of whole genome sequences (WGS) of 131 pre-treatment pediatric gliomas.

Maybe indicate here that these were all high grade gliomas?

Reviewer #4 (Remarks to the Author): Expert in proteomics and phosphoproteomics

The manuscript by Khadka et al. describes a multi-omics approach to identify PPM1D mutations that are oncogenic drivers of de novo Diffuse Midline Glioma formation. I have been asked to provide a technical review of the proteomics and phosphoproteomics approaches that were employed in this manuscript, which are related to Figures 5 and 7 and Supplementary Figure 7. Briefly, the authors did WGS on 170 pediatric gliomas to find that the phosphatase PPM1D is a likely driver of the malignancy. They then performed a number of in-vitro and in-vivo experiments to test this hypothesis including transcriptomics and phosphoproteomics of glioma cell lines with inhibitors of PPM1D to find that p53 activity is increased. The phosphoproteomic analysis was repeated to compare mNSC expressing PPM1Dtr or control, with similar results.

The proteomics analyses were been conducted by Steve Carr's group and they are experts in the field. The MS data quality and raw data processing is state-of-the-art and the conclusions drawn based on this correct, but it looks to me like the authors could do a better job in the presentation and bioinformatics analysis of the proteomics data.

Specifically:

1. The experimental design for proteomics analysis is sound (although with a limited sample size),

but the authors do not use the resulting large-scale datasets to any useful extent. Surprisingly, the authors do not derive any new conclusions from the phosphoproteomics-screening. I will recommend them to perform additional bioinformatics analysis including signaling pathway reconstitutions to mine the phosphoproteomics datasets in more depth.

The authors should also add some phosphosite motif analysis on top of the PTM-SEA they already show, as a double check of what they conclude.

2. The authors use multiplexed TMT labelling for the phosphoproteomics experiments, but I do not understand the labelling scheme described in the methods: if they only have two conditions and three replicates in mNSC, and two condition and two replicates for the patient-derived cell lines, do they use all TMT reporter channels in the 10plex? This needs to be addressed and better described in the methods section.

In addition to this, a specific description of the correspondence of each TMT tag with each sample needs to be provided.

Moreover, can the authors explain why they only use two replicates for the analysis of patient-derived cell lines? The standard guidelines in proteomics dictates at least three biological replicates per condition. This needs to be addressed.

3. Figure 5A and Supplementary Figure 7: I do not understand the labeling and annotation for the individual phosphosites. For example, for "S1065s" what does the second "s" indicate?

Also, sites labelled as "N", does this mean Asparagine phosphorylation? My guess is that the authors are referring to other amino acid modifications (deamidation?), since there is also sites labelled as M255m that I assume is an oxidized Methionine. These modifications are most likely 'in-vitro'-artifacts that occurred during sample preparation and they do not inform about the biology of the samples. Consequently, these sites are not meaningful for the figure and can lead to confusion, and therefore need to be removed or explained in the legend.

RESPONSE TO THE REVIEWERS

Reviewer #1 (Remarks to the Author):

The authors have satisfactorily addressed the concerns raised, and the manuscript is sufficiently improved.

Reviewer #2 (Remarks to the Author):

As stated in my comments from the first round of review, the main weakness of this manuscript remains the incremental degree of novelty, a point similarly raised by another reviewer. Nevertheless, I appreciate the authors' efforts to highlight the new findings primarily concerning the development of an in utero electroporation method to induce tumors in the brainstem. The possibility to successfully target MDM2 is also potentially interesting. It is reassuring that the authors have discussed the limited novelty and other limitations of the study in the revised manuscript.

Reviewer #3 (Remarks to the Author):

The manuscript has improved a lot and I have no further major comments. Only a few minor comments:

We thank all of the Reviewers for their comments which helped us improve our manuscript during this process.

Abstract

Should write DMG full out

We thank the Reviewer for pointing this out. We now show the full name of DMG in the abstract as follows:

“Here we analyze whole genomes sequences of 170 pediatric high-grade gliomas and find that truncating mutations in *PPM1D* that increase the stability of its phosphatase are clonal driver events in 11% of **Diffuse Midline Gliomas (DMGs)** and are enriched in primary pontine tumors.”

Introduction:

Diffuse midline gliomas H3K27M-mutant (DMG)

In the updated 5th edition of the WHO classification of CNS tumors, the name of these tumors has changed to “Diffuse midline gliomas H3K27-altered as not all of them have H3K27 mutations but there are other mechanisms that lead to low H3K27me3 levels in these tumors.

We thank the Reviewer for clarifying this. We have now corrected the nomenclature in the introduction as follows:

“**Diffuse midline gliomas H3K27-altered (DMG)** are universally fatal pediatric brain tumors.”

we first performed a comprehensive analysis of whole genome sequences (WGS) of 131 pre-treatment pediatric gliomas.

Maybe indicate here that these were all high-grade gliomas?

We thank the Reviewer for this suggestion. We have indicated that these were all high-grade gliomas as follows:

“To further evaluate the role of *PPM1D* mutations in gliomagenesis, we first performed a comprehensive analysis of whole genome sequences (WGS) of 131 pre-treatment **pediatric high-grade gliomas (pHGGs)**.”

Reviewer #4 (Remarks to the Author): Expert in proteomics and phosphoproteomics

The manuscript by Khadka et al. describes a multi-omics approach to identify *PPM1D* mutations that are oncogenic drivers of de novo Diffuse Midline Glioma formation. I have been asked to provide a technical review of the proteomics and phosphoproteomics approaches that were employed in this manuscript, which are related to Figures 5 and 7 and Supplementary Figure 7. Briefly, the authors did WGS on 170 pediatric gliomas to find that the phosphatase *PPM1D* is a likely driver of the malignancy. They then performed a number of in-vitro and in-vivo experiments to test this hypothesis including transcriptomics and phosphoproteomics of glioma cell lines with inhibitors of *PPM1D* to find that p53 activity is increased. The phosphoproteomic analysis was repeated to compare mNSC expressing *PPM1D*tr or control, with similar results.

The proteomics analyses were been conducted by Steve Carr's group and they are experts in the field. The MS data quality and raw data processing is state-of-the-art and the conclusions drawn based on this correct, but it looks to me like the authors could do a better job in the presentation and bioinformatics analysis of the proteomics data.

Specifically:

1. The experimental design for proteomics analysis is sound (although with a limited sample size), but the authors do not use the resulting large-scale datasets to any useful extent. Surprisingly, the authors do not derive any new conclusions from the phosphoproteomics-screening. I will recommend them to perform additional bioinformatics analysis including signaling pathway reconstitutions to mine the phosphoproteomics datasets in more depth.

The authors should also add some phosphosite motif analysis on top of the PTM-SEA they already show, as a double check of what they conclude.

We thank the Reviewer for their comments and suggestion regarding further pathway analysis and phosphosite motif analysis. In addition to the biological pathway enrichment analysis using STRING and PTMSEA, we have now performed CausalPath analysis as well as motif analysis of our dataset as follows:

“We next applied a recently published method CausalPath (Babur et al., 2021) to integrate our phosphoproteomic data with literature knowledge to generate causal hypotheses from the data. This analysis also indicated several DNA repair genes (*ATM*, *ATR*, *MDC1*, *MRE11*) to be activated upon *PPM1D* inhibition (Supplementary Figure 7A). Some of the observed phosphorylations of *TP53* and *TRIM28* are identified to be a consequence of *PPM1D* inhibition (Supplementary Figure 7A). We also observed some inhibitory sites on *PPM1D* (*S54* and *S85*) to be downregulated, possibly due to some negative feedback on *PPM1D* (Supplementary Figure 7A). Furthermore, sequence motif analysis identified a conserved SQ motif among putative *PPM1D*-dependent phosphorylation events ($P = 1.6e-49$) (Figure 5E), further supporting the role of *PPM1D* in response to DNA damage (Matsuoka et al., 2007). Consistent with previous reports (Kahn et al., 2018), we also observed an overrepresentation of glutamic acid at position +2 ($P = 4.2e-19$).”

2. The authors use multiplexed TMT labelling for the phosphoproteomics experiments, but I do not understand the labelling scheme described in the methods: if they only have two conditions and three replicates in mNSC, and two condition and two replicates for the patient-derived cell lines, do they use all TMT reporter channels in the 10plex? This needs to be addressed and better described in the methods section.

In addition to this, a specific description of the correspondence of each TMT tag with each sample needs

to be provided.

Moreover, can the authors explain why they only use two replicates for the analysis of patient-derived cell lines? The standard guidelines in proteomics dictates at least three biological replicates per condition. This needs to be addressed.

We apologize for the confusion that has been caused by the mismatch between the number of replicates and TMT plex strategy used in the patient-derived cell line experiment. This was because the original TMT 10-plex included additional samples representing another treatment group that were not included in this manuscript.

Therefore, to address reviewer's reasonable concern about the number of replicates, we designed and carried out an additional experiment with TMT6 reagent and included 3 replicates of each drug and DMSO treatment groups in the same patient-derived cell line. The methods section as well as the results and associated figure panels have been updated to reflect the new experiments. Supplementary Table 5 includes the dataset and column headers describe the channel to sample assignment. In addition, MassIVE data repository upload includes all the raw data files as well as detailed description of sample to channel assignments, reagent correction factors, databases searched, etc.

We thank the Reviewer for this suggestion as the increased number of replicates robustly confirmed our initial findings. We have updated the Results section, replacing the prior results with this new experiment that includes three replicates per condition.

3. Figure 5A and Supplementary Figure 7: I do not understand the labeling and annotation for the individual phosphosites. For example, for "S1065s" what does the second "s" indicate? Also, sites labelled as "N", does this mean Asparagine phosphorylation? My guess is that the authors are referring to other amino acid modifications (deamidation?), since there is also sites labelled as M255m that I assume is an oxidized Methionine. These modifications are most likely 'in-vitro'-artifacts that occurred during sample preparation and they do not inform about the biology of the samples. Consequently, these sites are not meaningful for the figure and can lead to confusion, and therefore need to be removed or explained in the legend.

Thank you for pointing out this error in labelling our figures. The labeling scheme for individual phosphosites was carried through from Spectrum Mill, the proteomics data analysis platform that we use in our lab, but it is indeed confusing in the figures. We have now corrected this labeling scheme and removed the last smaller case letter from the phosphosites in all of our figures as it is redundant and does not add any information. We have also removed all amino acid modifications that are most likely to be technical artifacts during sample preparation as suggested by the reviewer and therefore does not add any meaningful information about PPM1D's biology. This information for the phosphosites detected and quantified can still be found in the Supplementary Tables 5 and 7.

REVIEWERS' COMMENTS

Reviewer #4 (Remarks to the Author):

The authors have satisfactorily addressed all concerns and implemented my suggestions. The manuscript is significantly improved and I now recommend publication.